# Hyper Evidential Deep Learning to Quantify Composite Classification Uncertainty

**Changbin Li**[1], **Kangshuo Li**[1], **Yuzhe Ou**[1], **Lance M. Kaplan**[2], **Audun Jøsang**[3],
**Jin-Hee Cho**[4], **Dong Hyun Jeong**[5], **Feng Chen**[1]
Department of Computer Science, The University of Texas at Dallas[1],
US Army Research Laboratory[2], University of Oslo[3], Virginia Tech[4],
University of the District of Columbia[5]
{changbin.li,kangshuo.li,yuzhe.ou,feng.chen}@utdallas.edu
lance.m.kaplan.civ@army.mil, audun.josang@mn.uio.no,
djeong@udc.edu, jicho@vt.edu

## Abstract

Deep neural networks (DNNs) have been shown to perform well on exclusive, multi-class classification tasks. However, when different classes have similar visual features, it becomes challenging for human annotators to differentiate them. This scenario necessitates the use of composite class labels. In this paper, we propose a novel framework called *Hyper-Evidential Neural Network* (HENN) that explicitly models predictive uncertainty due to composite class labels in training data in the context of the belief theory called *Subjective Logic* (SL). By placing a grouped Dirichlet distribution on the class probabilities, we treat predictions of a neural network as parameters of hyper-subjective opinions and learn the network that collects both single and composite evidence leading to these hyper-opinions by a deterministic DNN from data. We introduce a new uncertainty type called *vagueness* originally designed for hyper-opinions in SL to quantify composite classification uncertainty for DNNs. Our results demonstrate that HENN outperforms its state-of-the-art counterparts based on four image datasets. The code and datasets are available at: https://github.com/Hugo101/HyperEvidentialNN.

## 1 Introduction

In various applications, particularly those dependent on data from low-quality sensors or high-quality data with insufficiently distinct features to separate some individual classes, the resulting data often exhibits significant vagueness and ambiguity (Allison, 2001; Ng et al., 2011). For example, in security surveillance, grainy images from store cameras may not provide clear enough resolution to accurately distinguish between different individuals or activities, necessitating the use of composite class labels to address this uncertainty (Allison, 2001). Similarly, in the field of medical imaging, a radiograph displaying features suggestive of multiple possible diagnoses may require composite labels to capture this uncertainty (Allison, 2001) effectively. When different classes have similar visual features in image datasets, it becomes challenging for human annotators to differentiate them. An ambiguous image, such as a blurry one where an annotator cannot distinguish between a husky and a wolf, may be labeled with both classes: {husky, wolf}. The composite label implies that the image belongs to husky or wolf, but not both. When training data consists of composite class labels, existing DNN methods face the following challenges: (a) how to train a DNN model based on a training set with composite labels; (b) how to train a DNN to predict the composite labels that human annotators could provide; and (c) how to quantify the predictive uncertainty of a DNN due to the evidence of composite labels collected from the training set.

In current literature, partial label learning (Feng et al., 2020; Hong et al., 2023) has been proposed to address the first challenge. It aims to train a DNN that can disambiguate the partially-labeled training samples and predict singleton class labels for testing data. To address the second challenge, conformal prediction (Vovk et al., 2005; Romano et al., 2020; Angelopoulos et al., 2021) is typically considered

in safety-critical applications (e.g., computer vision based medical diagnostics) and aims to provide a composite set that covers the true class label (e.g., the true diagnosis) with high probability (e.g., 90%). A composite set generated by a conformal prediction method is due to high entropy in the predicted class probabilities rather than composite class labels in the training set.

To the best of our knowledge, limited work has been conducted to address the third challenge. For predictive uncertainty quantification, several types/sources of predictive uncertainty have been studied in deep learning: model uncertainty (mutual information between model parameters and the predicted class probabilities (Depeweg et al., 2018; Malinin & Gales, 2018)), data uncertainty (entropy of the predicted class probabilities (Gal, 2016)), confidence (the largest predicted class probability (Hendrycks & Gimpel, 2017)), vacuity (uncertainty due to lack of evidence (Jøsang, 2016; Shi et al., 2020)), and dissonance (due to conflicting evidence (Zhao et al., 2020)). However, it lacks an effective uncertainty measure (here we name *vagueness*) that can quantify the uncertainty associated with predictions of a DNN due to composite class labels in the training set. For example, if the prediction (e.g., a singleton class or a composite class) of a DNN for a given input sample is based on evidence collected from training samples mostly labeled to composite sets, the vagueness should be high. If the collected evidence is from training samples mostly labeled to singleton classes, the vagueness should be low.

We propose a new framework called *Hyper Evidential Neural Network* (HENN) that explicitly models the predictive uncertainty of a DNN due to composite class labels in the training set. HENN is designed based on the theory of Subjective Logic (Jøsang, 2016) and aims to predict the evidence parameters of a hyper-opinion regarding the classification of the input sample. A hyper-opinion defines a belief mass distribution on the composite sets of singleton classes and an uncertainty mass and can be equivalently represented by a grouped Dirichlet distribution (GDD). We introduce a new uncertainty measure based on hyper-opinions, originally designed in SL (Jøsang, 2016; Jøsang et al., 2018), to quantify the *vagueness* of a DNN. **Our contributions** are three-fold: (1) We propose a novel framework (HENN) that can quantify *vagueness*, a new uncertainty type for measuring the predictive uncertainty of a DNN due to composite class labels in the training set. (2) We propose a new loss function, uncertainty partial cross entropy (UPCE), for HENN training. UPCE is a generalization of the well-known uncertainty cross-entropy (UCE)(Sensoy et al., 2018; Biloš et al., 2019) designed for singleton class labels. We provide a theoretical analysis of UPCE and propose a regularization term to future improve the effectiveness of UPCE for HENN learning. (3) We conduct extensive empirical analyses on four image datasets to demonstrate the effectiveness of the HENN in comparison with five competitive baselines.

## 2 RELATED WORK

**Evidential Neural Networks** (ENNs) (Sensoy et al., 2018; 2020; Ulmer et al., 2023) are deterministic neural networks that predict subjective opinions (Dirichlet distributions, equivalently) about the classifications of the input samples. The predicted subjective opinions can be used to quantify predictive uncertainties, such as vacuity (due to lack of evidence) and dissonance (due to conflicting evidence). PriorNet (Malinin & Gales, 2018; 2019) and PosteriorNet (Charpentier et al., 2020) are in this category. While Bengs et al. (2022) investigated the flaw of second-order uncertainty estimation of ENNs because of the lack of ground truth of target distribution, many applications (Xie et al., 2023; Sun et al., 2023; Park et al., 2023; Sapkota & Yu, 2023) show the usefulness of ENNs in recent years.

**Partial label learning** aims to train a DNN that can disambiguate the partially-labeled training samples and predict singleton class labels for testing data. Average-based methods (Cour et al., 2011) treat each candidate label as equally important during training. Conversely, identification-based approaches (Feng et al., 2020; Xu et al., 2021; Wang et al., 2022a; Qiao et al., 2023; Yan & Guo, 2023a;b; Hong et al., 2023) aim to disambiguate the effect of noisy labels and maximize outputs based on the most likely "ground-truth" label. **Soft label learning** aims to aggregate labels collected from multiple annotators to create probabilistic or "soft" labels for training data and learn a DNN for singleton class predictions based on the soft labels in the training data (Peterson et al., 2019; Collins et al., 2022). However, both partial and soft-label learning methods are limited to singleton-class predictions but not composite set predictions. Their learned models do not provide uncertainties associated with singleton-class predictions due to composite class labels in the training set.

**Composite set prediction.** E-CNN (Tong et al., 2021) could do set prediction for any possible combinations among all singleton classes based on Dempster-Shafer theory. RAPS (Angelopoulos

et al., 2021) is a state-of-the-art conformal prediction method that gives more stable predictive sets by regularizing the small scores of unlikely classes after Platt scaling. However, these methods predict composite sets for data instances with large probabilities for multiple classes. This means that their locations in the representation space are near the decision boundary of the DNN. In contrast, HENN predicts composite sets for data instances near the training instances with composite labels. These two methods are considered baselines in our empirical study.

## 3 HYPER-OPINIONS AND EVIDENTIAL UNCERTAINTY MEASURES

### 3.1 HYPER-OPINIONS IN SUBJECTIVE LOGIC AND GDD

In the Dempster–Shafer Theory of Evidence (DST) (Shafer, 1976), class probabilities in the Bayesian theory are generalized to belief masses in subjective opinions. It assigns belief masses to subsets of a ground set of exclusive possible states or classes (called "*domain*"). One can then express '*I do not know*' by assigning all belief masses to the whole domain as an opinion for the truth over possible classes. SL formalizes the DST's notion of belief assignments using a *hyper-opinion*. More specifically, let $\mathbb{Y} = \{1, ..., K\}$ denote the class domain with the cardinality $K$. Let $\mathscr{R}(\mathbb{Y})$ denote the reduced power set of $\mathbb{Y}$ (called "*hyper-domain*"), which is the set of the power set of $\mathbb{Y}$ that excludes $\{\mathbb{Y}\}$ and $\{\emptyset\}$. Let $\mathscr{C}(\mathbb{Y})$ denote the set of composite sets: $\mathscr{C}(\mathbb{Y}) = \mathscr{R}(\mathbb{Y}) \setminus \{\{1\}, \cdots, \{K\}\}$. A hyper-opinion $\omega = (\mathbf{b}, u)$ assigns a belief mass $b_S$ to each element (singleton class or composite set) $S \in \mathscr{R}(\mathbb{Y})$ and provides an uncertainty mass of $u$ called *vacuity*. These mass values are all non-negative and sum up to one, i.e.,

$$u + \sum\nolimits_{S \in \mathscr{R}(\mathbb{Y})} b_S = 1. \tag{1}$$

A belief mass $b_S$ is computed using the evidence for each element $S \in \mathscr{R}(\mathbb{Y})$. $S$ represents a singleton class if it has a single class element (e.g., $S = \{1\}$); otherwise, it represents a composite set (e.g., $S = \{1, 2\}$). Let $e_S \geq 0$ be the evidence derived for $S$, then the belief $b_S$ and the uncertainty mass $u$ are computed as:

$$b_S = \frac{e_S}{T}, \quad \text{and} \quad u = \frac{K}{T}, \tag{2}$$

where $T = \sum_{S \in \mathscr{R}(\mathbb{Y})} e_S + K$. The uncertainty mass $u$ is inversely proportional to the total evidence: $\sum_{S \in \mathscr{R}(\mathbb{Y})} e_S$. When the total evidence is 0, the belief mass for each $S$ needs to be 0, and the uncertainty mass $u$ is 1. In contrast to Bayesian modeling terms, we define "*evidence*" as a measure of the accumulated support from training samples, indicating that the input sample should be categorized into a particular singleton class or composite set. The accumulated support can be interpreted as the weighted aggregated number of training samples that support this class or composite set. Unlike a simple count of samples, evidence is typically weighted. This means that not all samples contribute equally to the evidence. For instance, some samples might be more informative or reliable than others, and the network learns to weigh their contribution to the evidence accordingly. Fig. 1 shows examples of high uncertainties for different types and their corresponding probability density plots for 3-class classification.

A hyper-opinion can be equivalently represented by a hyper-Dirichlet distribution of the class-probability vector $\mathbf{p} \in \Delta_K$, where $\Delta_K = \{\mathbf{p} | \sum_{k=1}^{K} p_k = 1 \text{ and } p_k \in [0,1]\}$ is the $K$-dimensional simplex. It is characterized by class-specific concentration parameters $\boldsymbol{\alpha} = [\alpha_1, \cdots, \alpha_K]$ and set-specific concentration parameters $\mathbf{c} = [c_S]_{S \in \mathscr{C}(\mathbb{Y})}$. The probability density function (pdf) for possible values of the class-probability vector $\mathbf{p}$ is given by

$$\texttt{HyperDir}(\mathbf{p}|\boldsymbol{\alpha}, \mathbf{c}) = Z_h^{-1} \prod_{k=1}^{K} p_k^{\alpha_k - 1} \prod_{S \in \mathscr{C}(\mathbb{Y})} \left( \sum_{k \in S} p_k \right)^{c_S}, \text{ for } \mathbf{p} \in \Delta_K, \tag{3}$$

where $Z_h$ is the normalization constant that has no analytical form. The concentration parameters of a hyper-Dirichlet distribution $\texttt{HyperDir}(\mathbf{p}|\boldsymbol{\alpha}, \mathbf{c})$ can be mapped to the evidence parameters of a hyper-opinion as follows: $\alpha_k = e_k + 1$ for $k = 1, \cdots, K$ and $c_S = e_S, \forall S \in \mathscr{C}(\mathbb{Y})$.

This paper considers an important special instance of Hyper-Dirichlet distribution: the grouped Dirichlet distribution (GDD), as it offers the practical appeal of an analytical normalization factor that can be easily calculated. GDD assumes that the composite sets in $\mathscr{C}(\mathbb{Y})$ represent a partition of the ground set of singleton classes, i.e., $\boldsymbol{\mathcal{S}} = \{\mathcal{S}_1, ..., \mathcal{S}_\eta\}$, where $\cup_{j=1}^{\eta} \mathcal{S}_j = \mathbb{Y}$ and $\mathcal{S}_i \cap \mathcal{S}_j = \emptyset$, $\forall i, j \in \{1, ..., \eta\}$, and $i \neq j$. Let $c_j$ denote $c_{\mathcal{S}_j}$. The pdf of GDD has the following form:

$$\texttt{GDD}(\mathbf{p}|\boldsymbol{\alpha}, \mathbf{c}) = Z^{-1} \prod_{k=1}^{K} p_k^{\alpha_k - 1} \prod_{j=1}^{\eta} \left( \sum_{l \in \mathcal{S}_j} p_l \right)^{c_j}, \text{ for } \mathbf{p} \in \Delta_K, \tag{4}$$

where $Z = \left[ \prod_{j=1}^{\eta} B\left( \{\alpha_l\}_{l \in \mathcal{S}_j} \right) \right] B\left( \{\beta_j\}_{j=1}^{\eta} \right)$, $\beta_j = \sum_{l \in \mathcal{S}_j} \alpha_l + c_j$, and $B(\cdot)$ is *beta* function. We aim to design and train evidential neural networks that can effectively predict the hyper-opinions about the uncertainty-aware classification of the input sample. As discussed in the following subsection, the predicted hyper-opinions can quantify *vagueness* and other uncertainty types.

**Relations with multinomial opinions and Dirichlet distribution.** A hyper-opinion is a generalized version of the *multinomial opinion* that assigns belief masses to singleton classes but not to composite sets (Jøsang et al., 2018). In particular, if there is no evidence for the composite sets in $\mathscr{C}(\mathbb{Y})$, then $e_S = 0, \forall S \in \mathscr{C}(\mathbb{Y})$. It follows that $\mathbf{c} = \mathbf{0}$ and the resulting hyper-opinion only assigns belief masses to singleton classes, and the corresponding Hyper Dirichlet distribution $\texttt{HyperDir}(\boldsymbol{\alpha}, \mathbf{0})$ is equivalent to the Dirichlet distribution $\texttt{Dir}(\boldsymbol{\alpha})$.

## 3.2 Vagueness and Other Evidential Uncertainty Measures by Hyper-Opinions

SL explicitly represents second-order probabilistic uncertainty through a hyper-opinion consisting of a belief mass distribution on $\mathscr{R}(\mathbb{Y})$ and uncertainty mass. A hyper-opinion can be used to quantify different types of uncertainty, such as vagueness (due to composite evidence), vacuity (due to lack of evidence), and dissonance (due to conflicting evidence). The *vagueness* uncertainty measure (also named total vague belief mass) of a hyper-opinion can be estimated as:

$$vag(\omega) = \sum_{S \in \mathscr{C}(\mathbb{Y})} b_S. \tag{5}$$

An opinion is totally vague when $vag(\omega) = 1$, and is partially vague when $0 < vag(\omega) < 1$. An opinion has mono-vagueness when only a single composite set has (vague) belief mass assigned to it. On the other hand, an opinion has pluri-vagueness when several composite sets have (vague) belief masses assigned to them.

The *vacuity* uncertainty corresponds to the uncertainty mass $u$ in a hyper-opinion and is calculated as $vac(\omega) = K/T$ in Eq. 2. The *dissonance* of a hyper-opinion can be derived from the same amount of conflicting evidence for different singleton classes or composite sets (see Eq.54 in App.) for its estimation based on the hyper-opinion). The vagueness $vag(\omega)$ is different from the vacuity $vac(\omega)$ in that vagueness results from existing evidence of composite sets that fail to discriminate between specific singleton classes, but vacuity reflects the lack of evidence for any singleton classes and composite sets. A totally vacuous opinion does not contain any vagueness by definition. The vagueness $vag(\omega)$ is different from the dissonance $diss(\omega)$ in that vagueness is due to evidence on composite sets, whereas dissonance reflects conflicting evidence collected from different singleton classes or composite sets. It is possible that an opinion has a high vagueness (e.g., $vag(\omega) = 1$) but a low dissonance (e.g., $diss(\omega) = 0$). *Hyper-opinions can contain vagueness, whereas multinomial opinions never contain vagueness. The ability to express vagueness is thus the main aspect that makes hyper-opinions different from multinomial opinions.*

Table 1: An example of hyper-opinion with low vacuity and dissonance but high vagueness.

| $\omega$ | $S \in \mathscr{R}(\mathbb{Y})$ | {1} | {2} | {3} | {1,2} | {1,3} | {2,3} | $u$ | $vac(\omega)$ | $diss(\omega)$ | $vag(\omega)$ |
|---|---|---|---|---|---|---|---|---|---|---|---|
| 1 | Evidence $e_S$ | 3 | 0 | 0 | 0 | 0 | 24 | 0.1 | 0.1 | 0.2 | 0.8 |
|  | Belief Mass $b_S$ | 0.1 | 0 | 0 | 0 | 0 | 0.8 | | | | |
| 2 | Evidence $e_S$ | 3 | 12 | 12 | 0 | 0 | 0 | 0.1 | 0.1 | 0.744 | 0 |
|  | Belief Mass $b_S$ | 0.1 | 0.4 | 0.4 | 0 | 0 | 0 | | | | |

Tab. 1 provides two examples ($\mathbb{Y} = \{1, 2, 3\}$, $K = 3$). The first example of hyper-opinion reflects high vagueness. There is high evidence observed on the composite set $\{2, 3\}$, and it causes high vagueness. There is also conflicting evidence between $\{1\}$ and $\{2, 3\}$ that contributes to dissonance. However, the evidence on $\{2, 3\}$ dominates the evidence on $\{1\}$, the dissonance is low. The total evidence is large for $K = 3$, and it results in low vacuity. The second example is a hyper-opinion with low vagueness and high dissonance, where the evidence in classes 2 and 3 is equally distributed on singletons instead of a composite set and becomes the conflicting evidence between the two classes.

## 4 Hyper Evidential Neural Network

In this section, we will present a novel hyper-evidential neural network (HENN) that predicts a hyper-opinion about the classification of the input feature vector $\mathbf{x}$. The predicted hyper-opinion can be used to quantify different types of predictive uncertainty, such as vagueness, vacuity, and

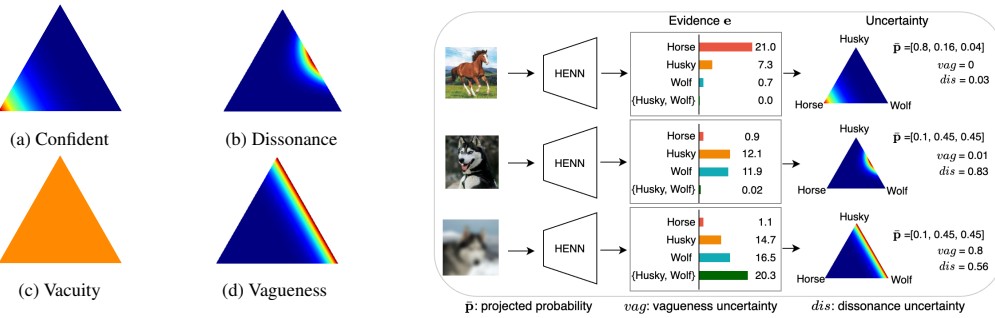

Examples of different uncertainties.      Different predictive uncertainties from HENN.

Figure 1: *Left*: Different probability densities corresponding to specific uncertainty type for 3-class classification (Brighter colors mean higher density). Each corner represents a class. (a) A confident prediction. (b) Conflicting evidence exists for two classes (*dissonance* or *data* uncertainty). (c) Uniform Dirichlet distribution with no evidence for known classes (i.e., OOD inputs) (*vacuity* uncertainty). (d) There is enough evidence to exclude one class but still fail to determine the final prediction from the rest of the classes. *Right*: The first example shows a confident prediction w/o *vagueness* and low *dissonance*. The other two examples have the same projected probabilities but different sources of uncertainties. One is caused by conflicting evidence (*dissonance*), and the other one is caused by vague evidence only for the final decision from the set {Husky, Wolf} (*vagueness*). Fig.(d) is drawn by grouped Dirichlet distribution, not ordinary Dirichlet distribution.

dissonance, as discussed in Section 3.2. We consider the scenario where the composite sets form a partition of the ground set of singleton classes, $\mathcal{S} = \{\mathcal{S}_1, ..., \mathcal{S}_\eta\}$, and the hyper-opinion can be equivalently represented by a grouped Dirichlet distribution. Formally, the HENN is defined as a function $f(\cdot, \boldsymbol{\theta}) : \mathbb{R}^D \to \mathbb{R}_+^{K+\eta}$, mapping an input $\mathbf{x} \in \mathbb{R}^D$ to the evidence vector $\mathbf{e} \in \mathbb{R}^{K+\eta}$, where $\boldsymbol{\theta}$ are the network parameters, $\mathbf{e} = [e_1, \cdots, e_K, e_{\mathcal{S}_1}, \cdots, e_{\mathcal{S}_\eta}]$ and $e_k$ and $e_{\mathcal{S}_i}$ refer to the predicted evidence values of the singleton class $k$ and the composite set $\mathcal{S}_i$, respectively. The architecture of HENNs for classification is similar to classical neural networks. The only difference is that the softmax layer is replaced with an activation layer (e.g., Softplus or ReLU) to ascertain non-negative and unbounded output that is considered as the evidence vector for the predicted hyper-opinion (or grouped Dirichlet distribution, equivalently). Based on the predicted evidence vector, we can then predict the singleton class or composite set that has the largest evidence:

$$m = \arg\max_{i \in \{1, 2, ..., K+\eta\}} e_i \qquad (6)$$

If $m \in \{1, \cdots, K\}$, then the prediction is a singleton class; otherwise, it is the composite set $\mathcal{S}_{m-K} \in \boldsymbol{\mathcal{S}}$. We can also transform the evidence vector $\mathbf{e}$ to a grouped Dirichlet distribution $\text{GDD}(\mathbf{p}|\boldsymbol{\alpha}, \mathbf{c})$ based on the mapping between the parameters $(\boldsymbol{\alpha}, \mathbf{c})$ and $\mathbf{e}$: $\boldsymbol{\alpha} = [e_1 + 1, \cdots, e_K + 1]$ and $\mathbf{c} = [e_{\mathcal{S}_1}, \cdots, e_{\mathcal{S}_\eta}]$. The relations between this distribution $\text{GDD}(\boldsymbol{\alpha}, \mathbf{c})$, the class probability vector $\mathbf{p}$, and the class label $y$ have the form:

$$y \sim \text{Cat}(\mathbf{p}), \quad \mathbf{p} \sim \text{GDD}(\boldsymbol{\alpha}, \boldsymbol{c}), \quad \mathbf{e} = f(\mathbf{x}; \boldsymbol{\theta}), \qquad (7)$$

where $\text{Cat}(\mathbf{p})$ is a categorical distribution on the class variable $y$. The expectation of the class-probability vector $\mathbf{p}$ has the form:

$$\bar{\mathbf{p}} := \mathbb{E}_{\mathbf{p} \sim \text{GDD}(\mathbf{p}|\boldsymbol{\alpha}, \mathbf{c})}[\mathbf{p}], \quad \bar{p}_k = \mathbb{E}[p_k] = \frac{\alpha_k}{\beta_0} \left( \sum_{j=1}^{\eta} \frac{\beta_j}{\alpha_{\mathcal{S}_j}} \cdot \mathbb{1}(k \in \mathcal{S}_j) \right) \text{ for } k \in \{1, \cdots, K\}, \qquad (8)$$

where $\beta_0 = \sum_{j=1}^{\eta} \beta_j$, $\alpha_{\mathcal{S}_j} = \sum_{l \in \mathcal{S}_j} \alpha_l$, $\beta_j = \alpha_{\mathcal{S}_j} + c_j$. Then, use $\bar{\mathbf{p}}$ as the projected class probability vector, we can also predict the singleton class with the largest projected class probability:

$$y = \arg\max_{k \in \{1, 2, ..., K\}} \bar{p}_k. \qquad (9)$$

**Relations with ENNs:** ENNs (a.k.a prior networks (Malinin & Gales, 2018)) and their variants (e.g., posterior networks (Charpentier et al., 2020)) are deterministic neural networks that predict the multinomial opinion (Dirichlet distribution, equivalently) about the singleton class label of the input sample. In comparison, HENNs are deterministic neural networks that predict the hyper-opinion (GDD, equivalently) about the classification of the input sample into a singleton class or composite class label. As discussed in Section 3.2, hyper-opinion has the ability to express the vagueness uncertainty (due to evidence collected from composite labels in the training data), but multinomial opinions never contain vagueness. HENNs are designed to handle composite labels in the training set and can quantify the composite classification uncertainty using the vagueness measure, whereas ENNs can not. In addition, as multinomial opinion is a special instance of hyper-opinion, by setting the evidence on composite sets to zero, HENNs include ENNs as a special instance.

## 4.1 THE LOSS FUNCTION AND REGULARIZATION FOR HENN LEARNING

Let $\mathcal{D} = \{(\mathbf{x}^{(i)}, \tilde{\mathbf{y}}^{(i)})\}_{i=1}^{N}$ denote a training set, where $\mathbf{x}$ refers to the input and $\tilde{\mathbf{y}} \in \{0, 1\}^K$ refers to the binary vector representation of a singleton class label or composite set label. For instance, $\tilde{\mathbf{y}}^{(i)} = [0, 1, 1, 0]^{\top}$ represents a composite set label $\{2, 3\}$ for the sample $i$ and $\tilde{\mathbf{y}}^{(i)} = [0, 0, 1, 0]^{\top}$ represents a singleton class label 3. In the related task of partial label learning (Cour et al., 2011), the partial cross-entropy (PCE) is used as a loss function for learning a softmax-based NN based on composite set (or called partial) labels:

$$\text{PCE}(\mathbf{p}, \tilde{\mathbf{y}}) = -\log\Big(\sum_{k=1}^{K} \tilde{y}_k p_k\Big), \tag{10}$$

where $\mathbf{p}$ refers to the class probability vector predicted by the softmax layer of the NN. When $\tilde{\mathbf{y}}$ is a singleton class label, the PCE loss becomes equivalent to the standard cross-entropy (CE) loss: $\text{CE}(\mathbf{p}, \tilde{\mathbf{y}}) = -\sum_{k=1}^{K} \tilde{y}_k \log p_k$. As the output of a HENN is a GDD of $\mathbf{p}$, we propose a new loss function, namely, Uncertainty Partial Cross Entropy (UPCE), to learn the parameters of a HENN:

$$\text{UPCE}(\mathbf{x}, \tilde{\mathbf{y}}; \boldsymbol{\theta}) = \mathbb{E}_{\mathbf{p} \sim \text{GDD}(\mathbf{p}|\boldsymbol{\alpha}, \mathbf{c})}[\text{PCE}(\mathbf{p}, \tilde{\mathbf{y}})], \tag{11}$$

where $\boldsymbol{\theta}$ refers to the network parameters of the HENN. We note that if $\tilde{\mathbf{y}}$ is a singleton class label, and we replace GDD with the Dirichlet distribution, then the UPCE loss becomes equivalent to the default UCE loss used in learning ENNs: $\text{UCE}(\mathbf{x}, \tilde{\mathbf{y}}) = \mathbb{E}_{\mathbf{p} \sim \text{Dir}(\mathbf{p}|\boldsymbol{\alpha})}[\text{CE}(\mathbf{p}, \tilde{\mathbf{y}})]$. Our proposed UPCE loss has the following analytical form (see our Proposition A1 in App. B.2 for derivations):

$$\text{UPCE}(\mathbf{x}, \tilde{\mathbf{y}}; \boldsymbol{\theta}) = \Big[\psi(\beta_0^{(i)}) - \psi(\beta_{\text{IC}}^{(i)})\Big] \mathbb{1}(\|\tilde{\mathbf{y}}^{(i)}\|_1 > 1) +$$

$$\Big[\Big(\psi(\beta_0^{(i)}) - \psi(\alpha_{\text{IS}}^{(i)})\Big) - \sum_{j=1}^{\eta} \Big(\psi(\beta_j^{(i)}) - \psi(\alpha_{\mathcal{S}_j}^{(i)})\Big) \mathbb{1}(y^{(i)} \in \mathcal{S}_j)\Big] \mathbb{1}(\|\tilde{\mathbf{y}}^{(i)}\|_1 = 1), \tag{12}$$

where the first term corresponds to composite example and the second term refers to singleton example respectively. In particular, $\psi(\cdot)$ is *digamma* function, $\beta_0^{(i)} = \sum_{j=1}^{\eta} \beta_j^{(i)} = \|\boldsymbol{\alpha}^{(i)}\| + \|\boldsymbol{c}^{(i)}\|$ denotes the sum of all positive strength parameters for the $i$-th sample, $\alpha_{\mathcal{S}_j}^{(i)} = \sum_{l \in \mathcal{S}_j} \alpha_l^{(i)}$ is the sum of strength parameters corresponding all singleton classes in the partition $\mathcal{S}_j$. For simplicity, we let $\beta_{\text{IC}}$ represent IC-th $\beta$ corresponding to one of a composite label in the list $\{\mathcal{S}_1, ..., \mathcal{S}_\eta\}$ which contains the singleton ground truth, and $\alpha_{\text{IS}}$ denote IS-th $\alpha$ corresponding to the singleton target.

Our proposed UPCE loss function has the lower bound (see Proposition A2 in App. C.1):

$$\text{UPCE}(\mathbf{x}, \tilde{\mathbf{y}}; \boldsymbol{\theta}) \geq \text{PCE}(\mathbb{E}_{\mathbf{p} \sim \text{GDD}(\mathbf{p}|\boldsymbol{\alpha}, \mathbf{c})}[\mathbf{p}], \tilde{\mathbf{y}}; \boldsymbol{\theta}). \tag{13}$$

It follows that the minimization of the UPCE loss ensures the minimization of the PCE loss between the projected class-probability vector $\mathbb{E}_{\text{GDD}(\mathbf{p}|\boldsymbol{\alpha}, \mathbf{c})}[\mathbf{p}]$ and the composite set label $\tilde{\mathbf{y}}$. This result indicates a favorable property of UPCE: The HENN with the UPCE loss function is optimized to output high projected probabilities for the classes belonging to the composite set label but low projected probabilities for the other classes.

However, as shown in Proposition 1, we observed an issue with the UPCE loss function in differentiating the evidence values of singleton classes and composite sets. In particular, when $\tilde{\mathbf{y}}$ is a composite set label, the learned HENN based on UPCE tends to predict large evidence values for both the composite set label $\tilde{\mathbf{y}}$ and for all the singleton classes belonging to the composite set. Similarly, when $\tilde{\mathbf{y}}$ is a singleton class label, the learned HENN based on UPCE tends to predict large evidence values for this singleton class and all the composite set that contains this singleton class as an element.

**Proposition 1** (Properties of the empirical UPCE risk function). *Assume that the universal approximation property (UAP) holds for a HENN, i.e., the network can learn an arbitrary mapping function from the input feature vector $\mathbf{x}$ to the evidence vector $\mathbf{e}$. Then, the empirical UPCE risk function $R(f) = \frac{1}{N} \sum_{i=1}^{N} \text{UPCE}(\mathbf{x}^{(i)}, \tilde{\mathbf{y}}^{(i)}; \boldsymbol{\theta})$ approaches the infimum 0 if the solution $\boldsymbol{\theta}^{\star}$ satisfies the following properties, with $\mathbf{e} = f(\mathbf{x}; \boldsymbol{\theta}^{\star})$: (1) $\forall(\mathbf{x}, \tilde{\mathbf{y}}) \in \mathcal{D}$, where $\tilde{\mathbf{y}}$ denotes a singleton class label, $k \in [K]$, the predicted evidence values $e_k \to +\infty$ and $e_{\mathcal{S}_i} \to +\infty, \forall \mathcal{S}_i \in \boldsymbol{\mathcal{S}}$, such that $k \in \mathcal{S}_i$; and (2) $\forall(\mathbf{x}, \tilde{\mathbf{y}}) \in \mathcal{D}$, where $\tilde{\mathbf{y}}$ denotes a composite set label $\mathcal{S}_i$, the predicted evidence values $e_{\mathcal{S}_i} \to +\infty$ and $e_k \to +\infty, \forall k \in \mathcal{S}_i$.*

To address the previous issue, we propose the following KL-divergence regularization term (see App. B.3 for derivations) to make the evidence output more flat:

$$\text{Reg}(\mathbf{x}, \tilde{\mathbf{y}}; \boldsymbol{\theta}) = \text{KL}\big[\text{GDD}(\mathbf{p}|\bar{\boldsymbol{\alpha}}, \bar{\boldsymbol{c}}) \| \text{GDD}(\mathbf{p}|\mathbf{1}^K, \mathbf{0}^\eta)\big], \tag{14}$$

where $\text{KL}(\cdot)$ is KL-divergence, and $\bar{\boldsymbol{\alpha}} = \tilde{\mathbf{y}} + (1 - \tilde{\mathbf{y}}) \odot \boldsymbol{\alpha}$ and $\bar{\mathbf{c}} = (1 - \tilde{\mathbf{y}}) \odot \mathbf{c}$ are the GDD parameters after the removal of ground-truth parameters from the predicted parameters $\boldsymbol{\alpha}$ and $\mathbf{c}$. This regularization term is designed to enforce misleading evidence from the false single and composite classes in $\boldsymbol{\alpha}$ and $\mathbf{c}$ to be as small as possible. The regularized UPCE loss function has the form:

$$\mathcal{L}(\boldsymbol{\theta}) = \frac{1}{N} \sum_{i=1}^{N} \left( \text{UPCE}(\mathbf{x}^{(i)}, \tilde{\mathbf{y}}^{(i)}; \boldsymbol{\theta}) + \lambda \cdot \text{Reg}(\mathbf{x}^{(i)}, \tilde{\mathbf{y}}^{(i)}; \boldsymbol{\theta}) \right), \qquad (15)$$

where $\lambda$ is the tradeoff coefficient. As indicated in Proposition 2 below, the HENN, when trained using the aforementioned regularized UPCE loss, tends to predict high evidence for the ground-truth singleton class/composite set while predicting low evidence for other elements. Stochastic gradient descent (i.e., Adam) is adopted to optimize the regularized loss function. The pseudocode is shown in App.(Algo. 1).

**Proposition 2** (Effectiveness of the regularization term $\text{Reg}(\mathbf{x}, \tilde{\mathbf{y}}; \boldsymbol{\theta})$)**.** *Following the UAP assumption, the regularized empirical UPCE risk defined in Eq. (15) approaches the infimum 0 if the solution $\boldsymbol{\theta}^{\star}$ satisfies the following properties: 1) $\forall (\mathbf{x}, \tilde{\mathbf{y}}) \in \mathcal{D}$, where $\tilde{\mathbf{y}}$ is a singleton class label $k \in [K]$, the predicted evidence values $e_k \to +\infty$ and $e_t \to 0, \forall t \in \boldsymbol{\mathcal{S}} \cup [K] \setminus k$; and 2) $\forall (\mathbf{x}, \tilde{\mathbf{y}}) \in \mathcal{D}$, where $\tilde{\mathbf{y}}$ denotes a composite set label $\mathcal{S}_i$, the predicted evidence values $e_{\mathcal{S}_i} \to +\infty$ and $e_t \to 0, \forall t \in \boldsymbol{\mathcal{S}} \cup [K] \setminus \mathcal{S}_i$.*

The proofs of the Propositions are shown in App. C.2 and C.3, respectively. This is consistent with the fact that the Dirichlet distribution is a special instance of GDD. As a generalized framework of ENN, the HENN learned based on UPCE will perform similarly to the ENN learned based on UCE when trained based on the same dataset with only singleton class labels.

**Limitations and discussions.** In essence, the proposed HENN is the GDD extension of evidential deep learning (Ulmer et al., 2023) that is based upon Dirichlet distributions. The propositions demonstrate the need for the KL regularization term in the cost function so that only evidence for the corresponding ground truth class can grow large. While the assumption of UAP in the above propositions may not hold in practice, the analysis does demonstrate how UPCE requires the KL regularization term to moderate the evidence. An ablation study is discussed in Section 5.2 to empirically demonstrate the need for the regularization term.

## 5 EXPERIMENTS

### 5.1 EXPERIMENTAL SETUP

**Datasets & Preprocessing**. **TinyImageNet** (Fei-Fei et al., 2015), **Living17** (Santurkar et al., 2021), **Nonliving26** (Santurkar et al., 2021), and **CIFAR100** (Krizhevsky & Hinton, 2009) are used in the experiments. Each dataset has class hierarchy relation. For example, CIFAR100 has 100 subclasses which are grouped into 20 disjointing superclasses. Superclasses are utilized to generate composite class labels because of their semantic and visual similarities. We first select a fixed number of composite class labels, denoted as $M$. Then several random subclasses for each selected superclass will be chosen. A subset of the selected images will be blurred by the Gaussian Blurring operation (RichardWebster et al., 2018) to generate vague images, and the corresponding set of categories/subclasses of these vague images will be the new label (composite instead of singleton) of these vague images to build the dataset. Detail is presented in App. E.

**Baselines**. **DNN** is the traditional deep neural network model. **ENN** (Sensoy et al., 2018) is the evidential network that only deals with traditional singleton domains as DNN does. We also use UCE loss and KL regularizer for a fair comparison for ENN. In practice, it is necessary to set a threshold value of predicted conditional class probabilities to generate set predictions for DNNs and ENNs. (see App. E.3). **E-CNN** (Tong et al., 2021) could do set prediction for any possible combinations among all singleton classes based on DST. **RAPS** (Angelopoulos et al., 2021) leverages conformal prediction to generate a prediction set to ensure the size of the predicted set is as small as possible. **PiCO** (Wang et al., 2022b) applies contrastive learning into partial label learning problem.

**Implementation**. Both HENN and ENN use Softplus as the activation layer. Since HENN is model-agnostic, we consider three pre-trained backbones: EfficientNet-b3 (Tan & Le, 2019), ResNet50 (He et al., 2015) and VGG16 (Simonyan & Zisserman, 2015) for HENN model and all other baselines for a fair comparison. Model agnostic property experiments are represented in App. F.2 due to the space limit. To generate composite examples for baselines, we create duplicate training examples with

Table 2: Results (%) based on Gaussian kernel size: $3\times3$ on CIFAR100 and tinyImageNet. (The average of three runs is provided, and the confidence interval is included in the App. due to space limitations.)

| $M$ | Methods | tinyImageNet | | | living17 | | | nonliving26 | | |
|---|---|---|---|---|---|---|---|---|---|---|
| | | OverJS | CompJS | Acc | OverJS | CompJS | Acc | OverJS | CompJS | Acc |
| 10 | DNN (Tan & Le, 2019) | 83.4 | 66.9 | 79.8 | 88.1 | 81.0 | 83.3 | 85.6 | 62.0 | 82.9 |
| | ENN (Sensoy et al., 2018) | 75.9 | 63.5 | 80.7 | 88.0 | 72.3 | 84.5 | 85.0 | 52.9 | 84.5 |
| | E-CNN (Tong et al., 2021) | 33.4 | 31.1 | 68.2 | 30.5 | 36.8 | 65.7 | 28.3 | 35.8 | 60.6 |
| | RAPS (Angelopoulos et al., 2021) | 73.1 | 43.6 | 79.8 | 86.4 | 61.3 | 83.3 | 82.7 | 46.3 | 82.9 |
| | PiCO (Wang et al., 2022b) | 57.2 | 35.6 | 64.3 | 62.5 | 43.7 | 65.2 | 61.8 | 42.6 | 64.8 |
| | HENN (ours) | **84.4** | **93.4** | **82.5** | **88.8** | **96.5** | **85.6** | **86.9** | **96.8** | **85.4** |
| 15 | DNN (Tan & Le, 2019) | 84.3 | 67.3 | 79.5 | 88.1 | 84.8 | 80.2 | 85.6 | 68.9 | 81.5 |
| | ENN (Sensoy et al., 2018) | 83.5 | 60.7 | 81.2 | 88.0 | 78.3 | 82.4 | 85.4 | 62.6 | 82.9 |
| | E-CNN (Tong et al., 2021) | 32.5 | 33.3 | 68.4 | 31.6 | 37.3 | 65.5 | 29.8 | 35.1 | 60.1 |
| | RAPS (Angelopoulos et al., 2021) | 68.1 | 45.6 | 79.5 | 85.5 | 66.5 | 80.2 | 83.8 | 56.1 | 81.5 |
| | PiCO (Wang et al., 2022b) | 56.8 | 35.3 | 64.6 | 61.4 | 43.1 | 64.8 | 61.5 | 42.5 | 64.6 |
| | HENN (ours) | **84.6** | **90.6** | **81.6** | **88.8** | **96.6** | **85.7** | **86.9** | **96.2** | **84.1** |

different singleton labels in the composite set. We adopt grid search based on a held-out validation set to select the best hyperparameters for each competitive method. Please refer to App. E.4 for details.

**Evaluation Metric**. Jaccard Similarity (JS) (Zaffalon et al., 2012) is used to evaluate a model's performance in predicting a set of classes: $\mathtt{JS}(y,\hat{y}) = \frac{|y \cap \hat{y}|}{|y \cup \hat{y}|}$, where $\hat{y}$ is the predicted set of classes and $y$ is ground-truth set of classes. Either $\hat{y}$, $y$ or both can be a single class or a set of two or more classes. A model identifies a datapoint as composite if two or more classes are predicted and singleton otherwise. We compare HENN's performance with baselines in terms of different average JS. OverJS: averaged JSs of *all* test samples, $\frac{1}{N_t}\sum_{i=1}^{N_t} \mathtt{JS}(y^{(i)}, \hat{y}^{(i)})$. CompJS: averaged JSs of composite samples the model identifies, $\frac{1}{N_c}\sum_{i=1}^{N_c} \mathtt{JS}(y^{(i)}, \hat{y}^{(i)})$, where $\mathtt{len}(\hat{y}^{(i)}) > 1$. Here $N_c = \sum_{i=1}^{N_t} \mathbb{1}(\mathtt{len}(\hat{y}^{(i)}) > 1)$ denotes the number of examples which are predicted as composite sets. Accuracy is used to evaluate the projected singleton label prediction (Acc). The Area Under the Receiver Operating Characteristic (AUROC, the larger the better) is to measure the different uncertainties in discriminating between true composite and true singleton samples.

## 5.2 EXPERIMENTAL RESULTS

For each dataset, we consider different numbers of composite class labels during training: $M$= 10, 15, 20, and multiple Gaussian kernel sizes of blurring operation: $3\times3$, $5\times5$, $7\times7$. Due to the space limit, some additional experiments including CIFAR100 are presented in App. F.

**Classification.** Tab. 2 shows the results of composite predictions on tinyImageNet, living17 and nonliving26 in terms of OverJS, CompJS (for composite prediction) and accuracy (for singleton prediction) based on Gaussian kernel size $3\times3$. HENN outperforms other baselines in terms of OverJS (over 1% for tinyImageNet) and CompJS. In particular, the improvement of CompJS is significant (over 15-20% for three datasets). This validates that HENN is not only able to recognize vague images, but also differentiate different vague images. In particular, both RAPS and E-CNN underperform HENN in terms of compJS. This is because RAPS and E-CNN are inclined to make set predictions if they are unsure about the final prediction. In addition to the vague images, there might be other difficult (not vague) images in which E-CNN and RAPS cannot make a single decision. Therefore, compared to DNN, more examples will be wrongly predicted as composite sets. On the other hand, Tab. 2 also demonstrate the efficacy of HENN in singleton prediction (Eq. 9) in terms of Acc. The improvement is over 2-5% for three datasets. PiCO performs worse than HENN because the composite labels in its setting are randomly flipped, and it might not be able to deal with blurring images during training. In summary, HENN can generate high-quality composite prediction and accurate singleton label classification. It is practical to consider a limited number of composite sets due to the majority of clearly labeled data. So our experiments do not encounter the combinatorial complexity issue ($2^K$).

**Analysis of confusion between multiple classes.** The ROC curves depicted in Fig. 2 showcase performances of various uncertainty indicators, namely *vagueness* of HENN, *vacuity* and *dissonance* of ENN, and *entropy* of DNN, in identifying confusion between multiple classes when some samples have composite labels, and some have singleton labels (see more in Fig. 4 and Fig. 5 in App.). For both datasets, HENN's *vagueness* outperforms the other uncertainties, as indicated by its larger AUC score and smallest error region, making it a highly effective dis-

criminator between composite and singleton samples and a successful indicator of confusion between a set of classes when there is composite evidence. RAPS and E-CNN, however, do not provide any measurement to evaluate these uncertainties. ENN's *vacuity*, similar to *epistemic* uncertainty, which is more useful for OOD detection (Fig. 1) and is not suitable to our case.

While *dissonance* and *entropy* are better than *vacuity*, they are still inferior to *vagueness*. A data point with high *dissonance* is usually located in the decision boundary, and a point with high *vagueness* can also be close to the decision boundary. However, the verse does not apply for *vagueness*, because *vagueness* could also be decided by the labeling bias of the annotators, but not purely by their closeness to the decision boundary between the associated singleton classes. For instance, an annotator who has extensive knowledge about different cat breeds (i.e., Tabby, Egyptian, Persian), will still annotate them as singletons, even if they are near decision boundaries. However, this annotator may give composite labels for other animal breeds, such as dog breeds (i.e., Husky, Malamute, Samoyed), that he may not be knowledgeable about. For this reason, a data point with high dissonance may likely have low vagueness.

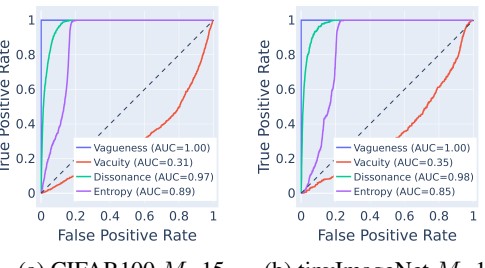

(a) CIFAR100 $M$=15    (b) tinyImageNet $M$=15

Figure 2: ROC curves of separating composite and singleton examples among different measurements: *vagueness* of HENN, *vacuity* and *dissonance* of ENN, and *entropy* of DNN on based on kernel size 7×7.

**Effect of regularization.** To show the effectiveness of KL divergence regularization, we compare different regularizations and UPCE loss without any regularization in Tab. 3. HENN-KL refers to the HENN with the proposed regularization (Eq. 14). HENN-Ent refers to the HENN with the entropy of GDD as the regularization $\text{Reg} = -\text{H}\big[\text{GDD}(\mathbf{p}|\boldsymbol{\alpha}, \boldsymbol{c})\big]$ (see App. B.4). HENN-KL-Dir refers to the HENN using KL-divergence only for singleton classes $\text{Reg} = \text{KL}\big[\text{Dir}(\mathbf{p}|\boldsymbol{\alpha})\big]$. Generally, the comparison of their performances is HENN-KL≈HENN-Ent > HENN-KL-Dir > HENN-only-UPCE. App. F.4 illustrates the coefficient effect of the KL regularizer.

Table 3: Model performance of different regularization on $M$=15, and kernel size: 7×7 on nonliving26.

| Methods | OverJS | CompJS | Acc |
|---|---|---|---|
| HENN-only-UPCE | 78.25 | 62.89 | 84.96 |
| HENN-KL-Dir | 86.66 | 94.15 | 82.13 |
| HENN-Ent | 86.68 | 94.44 | 86.33 |
| HENN-KL | 86.93 | 94.78 | 85.19 |

**Effect of varying numbers of composite labels.** To investigate the effect of ratio of composite class labels during training, we vary $M = \{10, 15, 20\}$ in experiments. Fig. 3 shows OverallJS and Accuracy regarding the number of composite sets. Regularized HENN outperforms other baselines for these two metrics. In particular, with the increase of number of composite sets, the gap between HENN and baselines is enlarging in terms of accuracy (Fig. 3b), which demonstrates the advantage of HENN.

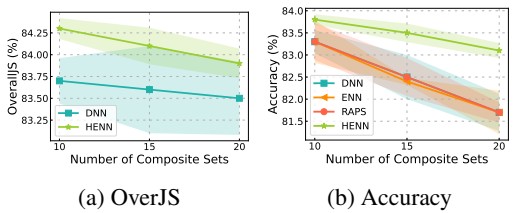

(a) OverJS    (b) Accuracy

Figure 3: OverJS and Accuracy trends vs. the number of composite labels in tinyImageNet.

# 6 CONCLUSION

In this work, we propose a novel hyper-evidential network framework (HENN) designed to predict hyper-opinions and quantify predictive classification uncertainty caused by composite class labels (introduced as *vagueness*) by utilizing composite training examples. This framework is capable of identifying either a singleton class or a composite set with the highest belief, and it can predict the singleton class with the greatest projected class probability. Extensive empirical findings show that HENN outperforms other competitive methods, demonstrating its effectiveness and potentiality.

ACKNOWLEDGMENTS

We thank Raisaat Atifa Rashid for her contribution to the initial dataset preprocessing. We thank the anonymous reviewers for the stimulating discussion and for helping improve the paper. This work is supported by the National Science Foundation (NSF) under Grant No DMS-2220574, FAI-2147375, IIS-2107450, IIS-2107451, and IIS-2107449.

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

# A  NOTATIONS

For clear interpretation, we list main notations used in this paper and their corresponding explanation, as shown in Table 4.

Table 4: Important Notations and Descriptions

| Notation | Description |
|---|---|
| $\mathbb{Y}$ | Domain of singleton elements or classes |
| $\mathscr{R}(\mathbb{Y})$ | Hyper-domain of $\mathbb{Y}$ |
| $\mathscr{C}(\mathbb{Y})$ | Domain of composite classes |
| $\mathcal{D} = \{(\mathbf{x}^{(i)}, \mathbf{y}^{(i)})\}_{i=1}^{N}$ | Training data with size $N$ |
| $B(\cdot), \Gamma(\cdot), \psi(\cdot), \psi_1(\cdot)$ | Beta function, Gamma function, Digamma function, trigamma function |
| $\mathtt{Dir}(\boldsymbol{\alpha})$ | Dirichlet distribution with strength $\boldsymbol{\alpha}$ |
| $\mathtt{GDD}(\boldsymbol{\alpha}, \mathbf{c})$ | Grouped Dirichlet distribution with strength $\boldsymbol{\alpha}$ and composite evidence $\mathbf{c}$ |
| $K, M$ | Total number of singleton (composite) classes |
| $Z$ | Normalizing constant of the Grouped Dirichlet distribution |
| $\Delta_K$ | $K$-dimensional simplex, *i.e.*, $\Delta_K := \{\mathbf{p} \mid \mathbf{p} = [p_1, \cdots, p_K] \in [0,1]^K$ and $\|\mathbf{p}\|_1 = 1\}$ |
| $y$ | Singleton ground truth label |
| $b_S$ | Vague belief mass of value $S$ in $\mathscr{R}(\mathbb{Y})$ |
| $u$ | Vacuity of evidence in a hyper-opinion |
| $\eta$ | Total number of partitions |
| $\kappa$ | Total number of elements in $\mathscr{R}(\mathbb{Y})$, *i.e.*, the total no. of singleton and composite classes |
| $\epsilon$ | A small error |
| $\omega$ | Hyper-opinion of a random hyper-variable $y \in \mathscr{R}(\mathbb{Y})$ |
| $\mathbf{x}^{(i)}$ | The feature vector of the $i$-th sample |
| $\tilde{\mathbf{y}}$ | Binary vector over $\{0,1\}^K$ |
| $\boldsymbol{b} = [b_1, .., b_K, b_{\mathcal{S}_1}, .., b_{\mathcal{S}_\eta}]^\intercal$ | Belief mass distribution over $\mathscr{R}(\mathbb{Y})$ |
| $\mathbf{e} = [e_1, ..., e_\kappa]^\intercal$ | Observed evidence vector over $\mathscr{R}(\mathbb{Y})$, $\mathbf{e} = [e_1, \cdots, e_K, e_{K+1}, \cdots, e_\kappa]^\intercal$ |
| $\mathbf{p} = [p_1, ..., p_K]^\intercal$ | Class probability vector over $\mathbb{Y}$ |
| $\boldsymbol{\alpha} = [\alpha_1, ..., \alpha_K]^\intercal$ | Strength vector of a Dirichlet distribution or the singleton part in grouped Dirichlet distribution |
| $S$ | An element as a set in hyper-domain (singleton or composite) |
| $\boldsymbol{\mathcal{S}} = \{\mathcal{S}_1, ..., \mathcal{S}_\eta\}$ | The set of partitions |
| $\mathcal{S}_j$ | $j$-th composite set in GDD |
| $\mathbf{c} = [c_1, ..., c_\eta]^\intercal$ | Evidence vector for the partitions in $\boldsymbol{\mathcal{S}}$ |
| $f(\mathbf{x}^{(i)}; \boldsymbol{\theta})$ | HENN parameterized by $\boldsymbol{\theta}$ that takes $\mathbf{x}^{(i)}$ as input |
| IS | Singleton ground-truth index |
| IC | Composite ground-truth index |
| $\mathcal{L}(\boldsymbol{\theta})$ | Uncertainty loss function w.r.t. parameters $\boldsymbol{\theta}$ |
| $\mathtt{UPCE}(\boldsymbol{\theta})$ | UPCE loss |
| $\mathtt{Reg}(\boldsymbol{\theta})$ | KL-divergence regularizer |
| $\boldsymbol{\rho}(\boldsymbol{\alpha})$ | Natural parameter based on $\boldsymbol{\alpha}$ (only in Appendix) |
| $\boldsymbol{\gamma}(\mathbf{c})$ | Natural parameter based on $\mathbf{c}$ (only in Appendix) |
| $\boldsymbol{u}(\mathbf{p})$ | Sufficient statistic of natural parameter $\boldsymbol{\rho}(\boldsymbol{\alpha})$ (only in Appendix) |
| $\boldsymbol{v}(\mathbf{p})$ | Sufficient statistic of natural parameter $\boldsymbol{\gamma}(\mathbf{c})$ (only in Appendix) |

## B DERIVATIVES OF LOSS FUNCTION

### B.1 EXPECTATION OF GDD

**Theorem.** *Let $\mathbf{x} \sim \text{GDD}_{n,2}(\boldsymbol{\alpha}, \mathbf{c})$ with 2 partitions, $\mathbf{x} \in \Delta_n$, where $\Delta_n$ denotes the n-dimensional simplex, $\boldsymbol{\alpha} = (\alpha_1, \cdots, \alpha_n)^\intercal$ is the strength parameter, and $\mathbf{c} = (c_1, c_2)^\intercal$ is the composite evidence parameter. Let $\mathcal{S}_1, \mathcal{S}_2$ denote the 2 partitions. The moment of $x_i$ is given by*

$$\mathbb{E}(x_i) = \frac{\alpha_i}{\beta_{12}} \left( \frac{\beta_1}{\alpha_{\mathcal{S}_1}} \cdot \mathbb{1}(i \in \mathcal{S}_1) + \frac{\beta_2}{\alpha_{\mathcal{S}_2}} \cdot \mathbb{1}(i \in \mathcal{S}_2) \right) \tag{16}$$

*where $\alpha_{\mathcal{S}_1} = \sum_{l \in \mathcal{S}_1} \alpha_l$, $\alpha_{\mathcal{S}_2} = \sum_{l \in \mathcal{S}_2} \alpha_l$, $\beta_1, \beta_2$, and $\beta_{12}$ are defined as $\beta_1 = \alpha_{\mathcal{S}_1} + c_1, \beta_2 = \alpha_{\mathcal{S}_2} + c_2, \beta_{12} = \beta_1 + \beta_2$, and $\mathbb{1}(\cdot)$ denotes the indicator function.*

According to the above Theorem which is from the book of Dirichlet and Related Distributions (Ng et al., 2011), analogy from two partitions to multiple partitions, we can get Eq. 8 in the main paper:

$$\mathbb{E}[p_k] = \frac{\alpha_k}{\beta_0} \left( \sum_{j=1}^{\eta} \frac{\beta_j}{\alpha_{\mathcal{S}_j}} \cdot \mathbb{1}(k \in \mathcal{S}_j) \right)$$

where $\beta_0 = \sum_{j=1}^{\eta} \beta_j$, $\alpha_{\mathcal{S}_j} = \sum_{l \in \mathcal{S}_j} \alpha_l$, and $\beta_j = \alpha_{\mathcal{S}_j} + c_j$.

### B.2 UPCE LOSS OF GDD

**Proposition A1** (Analytical form of UPCE). *Given the i-th sample $(\mathbf{x}^{(i)}, \tilde{\mathbf{y}}^{(i)}) \in \mathcal{D}$, and a HENN $f(\cdot; \boldsymbol{\theta})$, the Uncertainty Partial Cross Entropy (UPCE) loss for this sample can be formulated as the following analytical form:*

$$UPCE(\mathbf{x}^{(i)}, \tilde{\mathbf{y}}^{(i)}; \boldsymbol{\theta}) = \mathbb{E}_{\mathbf{p} \sim \text{GDD}(\mathbf{p}|\boldsymbol{\alpha}^{(i)}, \mathbf{c}^{(i)})} (-\log \sum_{k=1}^{K} \tilde{y}_k p_k)$$

$$= \left[ \psi(\sum_{j=1}^{\eta} \beta_j^{(i)}) - \psi(\beta_{IC}^{(i)}) \right] \mathbb{1}(\|\tilde{\mathbf{y}}^{(i)}\|_1 > 1) +$$

$$\left[ \left( \psi(\sum_{j=1}^{\eta} \beta_j^{(i)}) - \psi(\alpha_{IS}^{(i)}) \right) - \sum_{j=1}^{\eta} \left( \psi(\beta_j^{(i)}) - \psi(\sum_{l \in \mathcal{S}_j} \alpha_l^{(i)}) \right) \cdot \mathbb{1}(\tilde{\mathbf{y}}^{(i)} \in \mathcal{S}_j) \right] \mathbb{1}(\|\tilde{\mathbf{y}}^{(i)}\|_1 = 1)$$

*where $\beta_j = \sum_{l \in \mathcal{S}_j} \alpha_l + c_j$.*

*Proof.* The formal formulation of UPCE loss is formulated as follows.

$$\text{UPCE}(\boldsymbol{\theta}) = \mathbb{E}_{\mathbf{p} \sim \text{GDD}(\mathbf{p}|\boldsymbol{\alpha}^{(i)}, \mathbf{c}^{(i)})} (-\log \sum_{k=1}^{K} \tilde{y}_k p_k)$$

$$= \underbrace{\mathbb{E}\left[ -\log \sum_{l:\tilde{y}_l^{(i)}=1} p_l^{(i)} \right] \mathbb{1}(\|\tilde{\mathbf{y}}^{(i)}\|_1 > 1)}_{\text{Term 1}} + \underbrace{\mathbb{E}\left[ -\log p_{IS}^{(i)} \right] \mathbb{1}(\|\tilde{\mathbf{y}}^{(i)}\|_1 = 1)}_{\text{Term 2}} \tag{17}$$

We need to get the log expectations Term 1 and Term 2 above to calculate the UPCE loss. The following is to explain how we can derive these two terms.

Given the PDF of GDD $\text{GDD}(\mathbf{p}|\boldsymbol{\alpha}, \mathbf{c})$ (Eq. 4) we can rewrite it in the form of exponential family:

$$p(\mathbf{x}; \boldsymbol{\gamma}) = h(\mathbf{x}) \exp\left( \boldsymbol{\gamma}^\intercal u(\mathbf{x}) - A(\boldsymbol{\gamma}) \right) \tag{18}$$

with natural parameters $\boldsymbol{\gamma}$, sufficient statistic $u(\mathbf{x})$, and log-partition $A(\boldsymbol{\gamma})$.

Construct the pdf of GDD as exponential family:

$$p(\mathbf{x}; \boldsymbol{\gamma}) = \exp\left( \log \text{GDD}(\mathbf{p}|\boldsymbol{\alpha}, \mathbf{c}) \right). \tag{19}$$

The logarithm term can be constructed as:

$$
\begin{aligned}
\log \text{GDD}(\mathbf{p}|\boldsymbol{\alpha}, \mathbf{c}) &= \log \prod_{k=1}^{K} p_k^{\alpha_k - 1} \prod_{j=1}^{\eta} \Big( \sum_{l \in \mathcal{S}_j} p_l \Big)^{c_j} - \log Z \\
&= \sum_{k=1}^{K} \log p_k^{\alpha_k - 1} + \sum_{j=1}^{\eta} \log \Big( \sum_{l \in \mathcal{S}_j} p_l \Big)^{c_j} - \log Z \\
&= \sum_{k=1}^{K} (\alpha_k - 1) \log p_k + \sum_{j=1}^{\eta} c_j \log \Big( \sum_{l \in \mathcal{S}_j} p_l \Big) - \log Z.
\end{aligned}
\tag{20}
$$

Note that Gamma function has the transition relation between Beta function:

$$
\Gamma(a)\Gamma(b) = B(a, b)\Gamma(a + b),
\tag{21}
$$

which can be generalized to multiple variables (Murphy, 2022) as follows,

$$
B(a_1, a_2, ..., a_K) = \frac{\prod_{k=1}^{K} \Gamma(a_k)}{\Gamma(\sum_{k=1}^{K} a_k)}
\tag{22}
$$

Therefore, the normalizing constant:

$$
\begin{aligned}
Z &= \Big[ \prod_{j=1}^{\eta} B\Big( \{\alpha_l\}_{l \in \mathcal{S}_j} \Big) \Big] \cdot B\Big( \{\beta_j\}_{j=1}^{\eta} \Big) \\
&= \Big[ \prod_{j=1}^{\eta} \frac{\prod_{l \in \mathcal{S}_j} \Gamma(\alpha_l)}{\Gamma(\sum_{l \in \mathcal{S}_j} \alpha_l)} \Big] \cdot \frac{\prod_{j=1}^{\eta} \Gamma(\beta_j)}{\Gamma(\sum_{j=1}^{\eta} \beta_j)},
\end{aligned}
\tag{23}
$$

Now define the log-partition $A(\boldsymbol{\alpha}, \mathbf{c})$ as follows:

$$
\begin{aligned}
\log Z &= \sum_{j=1}^{\eta} \log \Big[ \frac{\prod_{l \in \mathcal{S}_j} \Gamma(\alpha_l)}{\Gamma(\sum_{l \in \mathcal{S}_j} \alpha_l)} \Big] + \log \frac{\prod_{j=1}^{\eta} \Gamma(\beta_j)}{\Gamma(\sum_{j=1}^{\eta} \beta_j)} \\
&= \sum_{j=1}^{\eta} \log \prod_{l \in \mathcal{S}_j} \Gamma(\alpha_l) - \sum_{j=1}^{\eta} \log \Gamma(\sum_{l \in \mathcal{S}_j} \alpha_l) + \log \prod_{j=1}^{\eta} \Gamma(\beta_j) - \log \Gamma(\sum_{j=1}^{\eta} \beta_j) \\
&= \sum_{j=1}^{\eta} \sum_{l \in \mathcal{S}_j} \log \Gamma(\alpha_l) - \sum_{j=1}^{\eta} \log \Gamma(\sum_{l \in \mathcal{S}_j} \alpha_l) + \sum_{j=1}^{\eta} \log \Gamma(\beta_j) - \log \Gamma(\sum_{j=1}^{\eta} \beta_j) \\
&= A(\boldsymbol{\alpha}, \mathbf{c}).
\end{aligned}
\tag{24}
$$

Suppose $\rho_k = \alpha_k - 1$, $\boldsymbol{\rho} = \boldsymbol{\alpha} - 1 = [\alpha_1 - 1, ..., \alpha_K - 1]^\intercal$, $\boldsymbol{u}(\mathbf{p}) = [\log p_1, \log p_2, ..., \log p_K]^\intercal$ and $\gamma_j = c_j$, $\boldsymbol{\gamma} = \mathbf{c} = [c_1, ..., c_\eta]^\intercal$, $\boldsymbol{v}(\mathbf{p}) = [\log \sum_{l \in \mathcal{S}_1} p_l, \log \sum_{l \in \mathcal{S}_2} p_l, ..., \log \sum_{l \in \mathcal{S}_\eta} p_l]^\intercal$, then the PDF of GDD would be in the form of exponential family as follows:

$$
\begin{aligned}
\text{GDD}(\mathbf{p}|\boldsymbol{\alpha}, \mathbf{c}) &= \exp \Big[ \sum_{k=1}^{K} (\alpha_k - 1) \log p_k + \sum_{j=1}^{\eta} c_j \log(\sum_{l \in \mathcal{S}_j} p_l) - A(\boldsymbol{\alpha}, \mathbf{c}) \Big] \\
&= \exp \Big[ \boldsymbol{\rho}(\boldsymbol{\alpha})^\intercal \cdot \boldsymbol{u}(\mathbf{p}) + \boldsymbol{\gamma}(\mathbf{c})^\intercal \cdot \boldsymbol{v}(\mathbf{p}) - A(\boldsymbol{\alpha}, \mathbf{c}) \Big].
\end{aligned}
\tag{25}
$$

We can identify that $\{\boldsymbol{\rho}(\boldsymbol{\alpha}), \boldsymbol{\gamma}(\mathbf{c})\}$ are natural parameters, $\{\boldsymbol{u}(\mathbf{p}), \boldsymbol{v}(\mathbf{p})\}$ are corresponding sufficient statistics, respectively.

According to the property with respect to the exponential family, we can state that

$$
\mathbb{E}[\boldsymbol{u}(\mathbf{p})_k] = \frac{dA(\boldsymbol{\alpha}, \mathbf{c})}{d\rho_k} = \frac{dA(\boldsymbol{\alpha}, \mathbf{c})}{d\alpha_k}, \qquad \mathbb{E}[\boldsymbol{v}(\mathbf{p})_j] = \frac{dA(\boldsymbol{\alpha}, \mathbf{c})}{d\gamma_j} = \frac{dA(\boldsymbol{\alpha}, \mathbf{c})}{dc_j}.
\tag{26}
$$

Since $\beta_j = \sum_{l \in \mathcal{S}_j} \alpha_l + c_j$, so $\frac{\partial \beta_j}{\partial \alpha_k} = \mathbb{1}(k \in \mathcal{S}_j)$. In addition, since $\psi(x) = \frac{d}{dx} \log \Gamma(x)$, the log expectation $\mathbb{E}[\boldsymbol{u}(\mathbf{p})_k]$ in Eq. 26 would be:

$$
\begin{aligned}
\mathbb{E}[\log(p_k)] &= \frac{\partial A(\boldsymbol{\alpha}, \mathbf{c})}{\partial \alpha_k} \\
&= \sum_{j=1}^{\eta} \frac{\partial \sum_{l \in \mathcal{S}_j} \log \Gamma(\alpha_l)}{\partial \alpha_k} - \sum_{j=1}^{\eta} \frac{\partial \log \Gamma(\sum_{l \in \mathcal{S}_j} \alpha_l)}{\partial \alpha_k} + \sum_{j=1}^{\eta} \frac{\partial \log(\Gamma(\beta_j))}{\partial \alpha_k} - \frac{\partial \log(\Gamma(\sum_{j=1}^{\eta} \beta_j))}{\partial \alpha_k} \\
&= \sum_{j=1}^{\eta} \frac{\sum_{l \in \mathcal{S}_j} \partial \log \Gamma(\alpha_l)}{\partial \alpha_k} - \sum_{j=1}^{\eta} \psi(\sum_{l \in \mathcal{S}_j} \alpha_l) \frac{\sum_{l \in \mathcal{S}_j} \partial \alpha_l}{\partial \alpha_k} + \sum_{j=1}^{\eta} \psi(\beta_j) \frac{\partial \beta_j}{\partial \alpha_k} - \psi(\sum_{j=1}^{\eta} \beta_j) \sum_{j=1}^{\eta} \frac{\partial \beta_j}{\partial \alpha_k} \\
&= \sum_{j=1}^{\eta} \psi(\alpha_k) \cdot \mathbb{1}(k \in \mathcal{S}_j) - \sum_{j=1}^{\eta} \psi(\sum_{l \in \mathcal{S}_j} \alpha_l) \cdot \mathbb{1}(k \in \mathcal{S}_j) + \sum_{j=1}^{\eta} \psi(\beta_j) \cdot \mathbb{1}(k \in \mathcal{S}_j) - \psi(\sum_{j=1}^{\eta} \beta_j) \sum_{j=1}^{\eta} \mathbb{1}(k \in \mathcal{S}_j) \\
&= \psi(\alpha_k) - \sum_{j=1}^{\eta} \psi(\sum_{l \in \mathcal{S}_j} \alpha_l) \cdot \mathbb{1}(k \in \mathcal{S}_j) + \sum_{j=1}^{\eta} \psi(\beta_j) \cdot \mathbb{1}(k \in \mathcal{S}_j) - \psi(\sum_{j=1}^{\eta} \beta_j) \\
&= \Big(\psi(\alpha_k) - \psi(\sum_{j=1}^{\eta} \beta_j)\Big) + \sum_{j=1}^{\eta} \Big(\psi(\beta_j) - \psi(\sum_{l \in \mathcal{S}_j} \alpha_l)\Big) \cdot \mathbb{1}(k \in \mathcal{S}_j).
\end{aligned}
\tag{27}
$$

Similarly, with the leverage of $\mathbb{E}[\boldsymbol{v}(\mathbf{p})_j]$ in Eq. 26,

$$
\begin{aligned}
\mathbb{E}[\log(\sum_{l \in \mathcal{S}_j} p_l)] &= \frac{\partial A(\boldsymbol{\alpha}, \mathbf{c})}{\partial c_j} \\
&= \sum_{j=1}^{\eta} \frac{\partial \sum_{l \in \mathcal{S}_j} \log \Gamma(\alpha_l)}{\partial c_j} - \sum_{j=1}^{\eta} \frac{\partial \log \Gamma(\sum_{l \in \mathcal{S}_j} \alpha_l)}{\partial c_j} + \sum_{j=1}^{\eta} \frac{\partial \log(\Gamma(\beta_j))}{\partial c_j} - \frac{\partial \log(\Gamma(\sum_{j=1}^{\eta} \beta_j))}{\partial c_j} \\
&= \sum_{j=1}^{\eta} \frac{\partial \log(\Gamma(\beta_j))}{\partial c_j} - \frac{\partial \log(\Gamma(\sum_{j=1}^{\eta} \beta_j))}{\partial c_j} \\
&= \frac{\partial \log(\Gamma(\beta_j))}{\partial c_j} - \psi(\sum_{j=1}^{\eta} \beta_j) \frac{\partial \sum_{j=1}^{\eta} \beta_j}{\partial c_j} \\
&= \psi(\beta_j) \frac{\partial \beta_j}{\partial c_j} - \psi(\sum_{j=1}^{\eta} \beta_j) \frac{\partial \beta_j}{\partial c_j} \\
&= \psi(\beta_j) - \psi(\sum_{j=1}^{\eta} \beta_j).
\end{aligned}
\tag{28}
$$

Thus, we successfully derive the essential component Term 1 (Eq. 28) and Term 2 (Eq. 27), which can be used to calculate UPCE loss as follows,

$$
\begin{aligned}
\text{UPCE}(\boldsymbol{\theta}) &= \mathbb{E}_{\mathbf{p} \sim \text{GDD}(\mathbf{p}|\boldsymbol{\alpha}^{(i)}, \mathbf{c}^{(i)})} (-\log \sum_{k=1}^{K} \tilde{y}_k p_k) \\
&= \mathbb{E}\Big[-\log \sum_{l: \tilde{y}_l^{(i)}=1} p_l^{(i)}\Big] \mathbb{1}(\|\tilde{\mathbf{y}}^{(i)}\|_1 > 1) + \mathbb{E}\Big[-\log p_{\text{IS}}^{(i)}\Big] \mathbb{1}(\|\tilde{\mathbf{y}}^{(i)}\|_1 = 1) \\
&= \Big[\psi(\sum_{j=1}^{\eta} \beta_j^{(i)}) - \psi(\beta_{\text{IC}}^{(i)})\Big] \mathbb{1}(\|\tilde{\mathbf{y}}^{(i)}\|_1 > 1) + \\
&\quad \Big[\Big(\psi(\sum_{j=1}^{\eta} \beta_j^{(i)}) - \psi(\alpha_{\text{IS}}^{(i)})\Big) - \sum_{j=1}^{\eta} \Big(\psi(\beta_j^{(i)}) - \psi(\sum_{l \in \mathcal{S}_j} \alpha_l^{(i)})\Big) \cdot \mathbb{1}(\tilde{\mathbf{y}}^{(i)} \in \mathcal{S}_j)\Big] \mathbb{1}(\|\tilde{\mathbf{y}}^{(i)}\|_1 = 1)
\end{aligned}
$$

where $\beta_j = \sum_{l \in \mathcal{S}_j} \alpha_l + c_j$. $\qquad \square$

### B.3 KL Divergence as Regularization

Let $\mathtt{KL}(\cdot)$ denote the KL-divergence of two distributions. According to the Appendix C.3 in Ulmer et al. (2023), the KL-divergence of two GDD distributions can be written as:

$$
\begin{aligned}
\mathtt{KL}\Big(\mathtt{GDD}(\mathbf{p}|\bar{\boldsymbol{\alpha}},\bar{\mathbf{c}})||\mathtt{GDD}(\mathbf{p}|\boldsymbol{\alpha},\mathbf{c})\Big) &= \mathbb{E}\bigg[\log\frac{\mathtt{GDD}(\mathbf{p}|\bar{\boldsymbol{\alpha}},\bar{\mathbf{c}})}{\mathtt{GDD}(\mathbf{p}|\boldsymbol{\alpha},\mathbf{c})}\bigg] \\
&= \mathbb{E}\bigg[\log\mathtt{GDD}(\mathbf{p}|\bar{\boldsymbol{\alpha}},\bar{\mathbf{c}})\bigg] - \mathbb{E}\bigg[\log\mathtt{GDD}(\mathbf{p}|\boldsymbol{\alpha},\mathbf{c})\bigg].
\end{aligned}
\tag{29}
$$

Since we derived the entropy of GDD distribution in Eq. 33, we have

$$
-\mathbb{E}[\log\mathtt{GDD}(\mathbf{p}|\boldsymbol{\alpha},\mathbf{c})] = \log Z(\boldsymbol{\alpha},\mathbf{c}) - \sum_{k=1}^{K}(\alpha_k-1)\mathbb{E}\Big[\log p_k\Big] - \sum_{j=1}^{\eta}c_j\mathbb{E}\Big[\log\sum_{l\in\mathcal{S}_j}p_l\Big],
\tag{30}
$$

By putting the above term into the Eq. 29, we now have:

$$
\begin{aligned}
&\mathtt{KL}\Big(\mathtt{GDD}(\mathbf{p}|\bar{\boldsymbol{\alpha}},\bar{\mathbf{c}})||\mathtt{GDD}(\mathbf{p}|\boldsymbol{\alpha},\mathbf{c})\Big) \\
&= -\log Z(\bar{\boldsymbol{\alpha}},\bar{\mathbf{c}}) + \sum_{k=1}^{K}(\bar{\alpha}_k-1)\mathbb{E}\Big[\log p_k\Big] + \sum_{j=1}^{\eta}\bar{c}_j\mathbb{E}\Big[\log\sum_{l\in\mathcal{S}_j}p_l\Big] \\
&\quad - \bigg[-\log Z(\boldsymbol{\alpha},\mathbf{c}) + \sum_{k=1}^{K}(\alpha_k-1)\mathbb{E}\Big[\log p_k\Big] + \sum_{j=1}^{\eta}c_j\mathbb{E}\Big[\log\sum_{l\in\mathcal{S}_j}p_l\Big]\bigg] \\
&= \log\frac{Z(\boldsymbol{\alpha},\mathbf{c})}{Z(\bar{\boldsymbol{\alpha}},\bar{\mathbf{c}})} + \sum_{k=1}^{K}(\bar{\alpha}_k-\alpha_k)\mathbb{E}\Big[\log p_k\Big] + \sum_{j=1}^{\eta}(\bar{c}_j-c_j)\mathbb{E}\Big[\log\sum_{l\in\mathcal{S}_j}p_l\Big].
\end{aligned}
\tag{31}
$$

Therefore, we derive the following regularization based on $\mathtt{GDD}(\mathbf{p}|\mathbf{1}^K,\mathbf{0}^\eta)$,

$$
\begin{aligned}
&\mathtt{KL}\Big(\mathtt{GDD}(\mathbf{p}|\bar{\boldsymbol{\alpha}},\bar{\mathbf{c}})||\mathtt{GDD}(\mathbf{p}|\mathbf{1}^K,\mathbf{0}^\eta)\Big) \\
&= \log Z(\mathbf{1}^K,\mathbf{0}^\eta) - \log Z(\bar{\boldsymbol{\alpha}},\bar{\mathbf{c}}) + \sum_{k=1}^{K}(\bar{\alpha}_k-1)\mathbb{E}\Big[\log p_k\Big] + \sum_{j=1}^{\eta}\bar{c}_j\mathbb{E}\Big[\log\sum_{l\in\mathcal{S}_j}p_l\Big],
\end{aligned}
\tag{32}
$$

where $\mathbb{E}\left[\log p_k\right]$ and $\mathbb{E}\left[\log\sum_{l\in\mathcal{S}_j}p_l\right]$ are derived in Eq. 27 and Eq. 28 respectively, $\bar{\alpha}_k = \tilde{y}_k + (1-\tilde{y}_k)\odot\alpha_k$ is the Dirichlet parameter after removal of the non-misleading evidence from the predicted parameters $\boldsymbol{\alpha}$, specifically, we skip the comparison of $\alpha_k$ with $\mathbf{1}_k$ given $y=k$ for $k\in[K]$. $\bar{\mathbf{c}}_j = (1-\tilde{y}_j)\odot c_j$ as composite evidence parameter with the target class setting to be 0, for $j\in[\eta]$.

### B.4 Entropy of GDD

We can derive the entropy of a GDD distribution from its definition, and by using the component $\mathbb{E}\left[\log p_k\right]$ and $\mathbb{E}\left[\log\sum_{l\in\mathcal{S}_j}p_l\right]$ which are derived in Eq. 27 and Eq. 28 respectively, the full analytical

form can be derived:

$$
\begin{aligned}
H[\mathbf{p}] &= -\mathbb{E}\big[\log \texttt{GDD}(\mathbf{p}|\boldsymbol{\alpha},\mathbf{c})\big] \\
&= -\mathbb{E}\Big[\log Z^{-1} + \log \prod_{k=1}^{K} p_k^{\alpha_k-1} + \log \prod_{j=1}^{\eta}\Big(\sum_{l\in\mathcal{S}_j} p_l\Big)^{c_j}\Big] \\
&= \log Z - \mathbb{E}\Big[\sum_{k=1}^{K}\log p_k^{\alpha_k-1}\Big] - \mathbb{E}\Big[\sum_{j=1}^{\eta}\log\Big(\sum_{l\in\mathcal{S}_j} p_l\Big)^{c_j}\Big] \\
&= \log Z - \sum_{k=1}^{K}(\alpha_k-1)\mathbb{E}\Big[\log p_k\Big] - \sum_{j=1}^{\eta} c_j\mathbb{E}\Big[\log\sum_{l\in\mathcal{S}_j} p_l\Big] \\
&= \log Z - \sum_{k=1}^{K}(\alpha_k-1)\Big(\Big(\psi(\alpha_k)-\psi(\sum_{j=1}^{\eta}\beta_j)\Big) + \sum_{j=1}^{\eta}\Big(\psi(\beta_j)-\psi(\sum_{l\in\mathcal{S}_j}\alpha_l)\Big)\cdot\mathbb{1}\big(k\in\mathcal{S}_j\big)\Big) \\
&\quad - \sum_{j=1}^{\eta} c_j\mathbb{E}\Big[\log\sum_{l\in\mathcal{S}_j} p_l\Big] \\
&= \log Z - \sum_{k=1}^{K}(\alpha_k-1)\psi(\alpha_k) + \sum_{k=1}^{K}(\alpha_k-1)\psi(\sum_{j=1}^{\eta}\beta_j) - \sum_{k=1}^{K}(\alpha_k-1)\sum_{j=1}^{\eta}\Big(\psi(\beta_j)-\psi(\sum_{l\in\mathcal{S}_j}\alpha_l)\Big)\cdot\mathbb{1}(k\in\mathcal{S}_j) \\
&\quad - \sum_{j=1}^{\eta} c_j\Big(\psi(\beta_j)-\psi(\sum_{j=1}^{\eta}\beta_j)\Big)
\end{aligned}
\tag{33}
$$

## C  THEORETICAL ANALYSIS OF LOSS FUNCTION

### C.1  CONVEXITY OF CE & PCE

To prove the convexity of CE and PCE loss with respect to class probabilities, we only need to show that the second-order derivative of both losses is non-negative. For CE loss $\texttt{CE}(\mathbf{p},\tilde{\mathbf{y}}) = -\sum_{k=1}^{K}\tilde{y}_k\log p_k$, since $p_k \geq 0$ and $\tilde{y}_k \geq 0$ for any $k\in\{1,2,...,K\}$:

$$
\begin{aligned}
\texttt{CE}_k' &= \frac{d}{dp_k}\texttt{CE} = \frac{d}{dp_k}[-\tilde{y}_k\log p_k] = -\frac{\tilde{y}_k}{p_k}, \\
\texttt{CE}_k'' &= \frac{d}{dp_k}\texttt{CE}_k' = \frac{d}{dp_k}[-\frac{\tilde{y}_k}{p_k}] = \frac{\tilde{y}_k}{p_k^2} \geq 0.
\end{aligned}
\tag{34}
$$

By Eq. 34, we can know that the Hessian matrix is diagonal and positive semi-definite. Hence, the CE loss is convex.

For PCE loss $\texttt{PCE}(\mathbf{p},\tilde{\mathbf{y}}) = -\log(\sum_{k=1}^{K}\tilde{y}_k p_k)$, we have:

$$
\begin{aligned}
\texttt{PCE}_k' &= \frac{d}{dp_k}\texttt{PCE} = -\frac{\tilde{y}_k}{\sum_{j=1}^{K}\tilde{y}_j p_j}, \\
\texttt{PCE}_k'' &= \frac{d}{dp_k}\texttt{PCE}_k' = -\tilde{y}_k\Big[-(\sum_{j=1}^{K}\tilde{y}_j p_j)^{-2}\Big]\tilde{y}_k = \Big(\frac{\tilde{y}_k}{\sum_{j=1}^{K}\tilde{y}_j p_j}\Big)^2 \geq 0,
\end{aligned}
\tag{35}
$$

where $\tilde{y}_k$ is the $k$-th element in the binary vector $\tilde{\mathbf{y}}$ representing classes in $\mathscr{R}(\mathbb{Y})$. Analogously, PCE loss is convex and thus follows Jensen's inequality.

**Proposition A2** (Lower Bound of UPCE). *Given any instance* $(\mathbf{x},\tilde{\mathbf{y}})$, *and a HENN* $f(\cdot;\boldsymbol{\theta})$, *the Uncertainty Partial Cross Entropy (UPCE) for this sample* $UPCE(\mathbf{x},\tilde{\mathbf{y}};\boldsymbol{\theta})$ *has the following lower bound:*

$$
UPCE(\mathbf{x},\tilde{\mathbf{y}};\boldsymbol{\theta}) \geq PCE(\mathbb{E}_{\mathbf{p}\sim\texttt{GDD}(\mathbf{p}|\boldsymbol{\alpha},\mathbf{c})}[\mathbf{p}],\tilde{\mathbf{y}};\boldsymbol{\theta}).
\tag{36}
$$

*Proof.* Since $\mathrm{PCE}(\mathbf{p}, \tilde{\mathbf{y}}; \boldsymbol{\theta})$ is convex (proved earlier), it is straightforward to get the following inequality through Jensen's inequality:

$$
\begin{aligned}
\mathrm{UPCE}(\mathbf{x}, \tilde{\mathbf{y}}; \boldsymbol{\theta}) &= \mathbb{E}_{\mathbf{p} \sim \mathrm{GDD}(\mathbf{p}|\boldsymbol{\alpha}, \mathbf{c})} \Big[ \mathrm{PCE}(\mathbf{p}, \tilde{\mathbf{y}}; \boldsymbol{\theta}) \Big] \\
&\geq \mathrm{PCE}(\mathbb{E}_{\mathbf{p} \sim \mathrm{GDD}(\mathbf{p}|\boldsymbol{\alpha}, \mathbf{c})}[\mathbf{p}], \tilde{\mathbf{y}}; \boldsymbol{\theta}).
\end{aligned}
\tag{37}
$$

$\square$

### C.2 PROOF OF PROPOSITION 1

**Proposition 1** (Properties of the empirical UPCE risk function). *Assume that the universal approximation property (UAP) holds for a HENN, i.e., the network can learn an arbitrary mapping function from the input feature vector $\mathbf{x}$ to the evidence vector $\mathbf{e}$. Then, the empirical UPCE risk function $R(f) = \frac{1}{N} \sum_{i=1}^{N} UPCE(\mathbf{x}^{(i)}, \tilde{\mathbf{y}}^{(i)}; \boldsymbol{\theta})$ approaches the infimum $0$ if the solution $\boldsymbol{\theta}^{\star}$ satisfies the following properties, with $\mathbf{e} = f(\mathbf{x}; \boldsymbol{\theta}^{\star})$: (1) $\forall (\mathbf{x}, \tilde{\mathbf{y}}) \in \mathcal{D}$, where $\tilde{\mathbf{y}}$ denotes a singleton class label, $k \in [K]$, the predicted evidence values $e_k \to +\infty$ and $e_{\mathcal{S}_i} \to +\infty, \forall \mathcal{S}_i \in \boldsymbol{\mathcal{S}}$, such that $k \in \mathcal{S}_i$; and (2) $\forall (\mathbf{x}, \tilde{\mathbf{y}}) \in \mathcal{D}$, where $\tilde{\mathbf{y}}$ denotes a composite set label $\mathcal{S}_i$, the predicted evidence values $e_{\mathcal{S}_i} \to +\infty$ and $e_k \to +\infty, \forall k \in \mathcal{S}_i$.*

*Proof.* Given the HENN with empirical risk as $R(f) = \frac{1}{N} \sum_{i=1}^{N} \Big[ \mathrm{UPCE}(\mathbf{x}^{(i)}, \tilde{\mathbf{y}}^{(i)}; \boldsymbol{\theta}) \Big]$, we can show that one of the optimal risk minimizers can always predict non-confident evidence while still maintain the property regarding the loss minimizer for arbitrary examples in the training set.

First, we show the properties hold for a UPCE loss minimizer. Since opinions in Subjective Logic rely on estimating evidence to form subjective opinions and reflect structural knowledge, it is necessary to have accurate and consistent evidence output that supports a subset of the hypothesis space. Based on this definition, the composite evidence should not be too large given only singleton training examples, and vice versa. Nonetheless, Proposition 1 states that for a given data point $(\mathbf{x}, \tilde{\mathbf{y}})$, different minimizers trained with the same UPCE objective can end up with different evidence predictions.

As UPCE loss is convex in terms of evidence, we consider analyzing the impact of output evidence on UPCE loss. We start with partial derivatives because the UPCE loss is multivariate differentiable. For clarity, we've organized our proof into two parts: one dealing with a single ground-truth scenario, and the other with a composite ground-truth.

**Case 1**: Under singleton ground-truth assumption.

Ideally, under the singleton ground-truth assumption, we anticipate that, as the singleton ground-truth evidence increases, the UPCE loss should decrease. When composite evidence increases, the UPCE loss should increase, i.e., $\frac{\partial}{\partial \alpha_\nu} \mathrm{UPCE} < 0, \forall \nu \in [K]$ and $\frac{\partial}{\partial c_\nu} \mathrm{UPCE} \geq 0, \forall \nu \in [\eta]$. This expectation is rooted in the fact that the UPCE loss is multivariate differentiable. If we explicitly write the partial derivative for composite evidence $c_\nu$ ($\nu \in [\eta]$) with singleton ground-truth, we will have

$$
\begin{aligned}
\frac{\partial}{\partial c_\nu} \mathrm{UPCE} &= \frac{\partial}{\partial c_\nu} \Bigg[ \psi(\sum_{j=1}^{\eta} \beta_j) - \psi(\alpha_{\mathrm{IS}}) + \sum_{j=1}^{\eta} \Big( \psi(\sum_{l \in \mathcal{S}_j} \alpha_l) - \psi(\sum_{l \in \mathcal{S}_j} \alpha_l + c_j) \Big) \mathbb{1}(y \in \mathcal{S}_j) \Bigg] \\
&= \frac{\partial}{\partial c_\nu} \psi(\sum_{k=1}^{K} \alpha_k + \sum_{j=1}^{\eta} c_j) - \sum_{j=1}^{\eta} \mathbb{1}(y \in \mathcal{S}_j) \frac{\partial}{\partial c_\nu} \psi(\sum_{l \in \mathcal{S}_j} \alpha_l + c_j) \\
&= \psi_1(\sum_{k=1}^{K} \alpha_k + \sum_{j=1}^{\eta} c_j) - \sum_{j=1}^{\eta} \mathbb{1}(y \in \mathcal{S}_j) \psi_1(\sum_{l \in \mathcal{S}_j} \alpha_l + c_j) \mathbb{1}(\nu = j).
\end{aligned}
\tag{38}
$$

where $\psi_1(\cdot)$ is $\psi_1(x) = \frac{d\psi(x)}{dx}$, known as the trigamma function, which is positive and monotonically decreasing on $(0, +\infty)$ (Qi & Berg, 2013). Next, we will go through different composite set labels to simplify the partial derivative.

If the partial derivative taken is not for the composite class label including the singleton ground-truth, then $\frac{\partial}{\partial c_\nu} \mathrm{UPCE} = \psi_1(\sum_{k=1}^{K} \alpha_k + \sum_{j=1}^{\eta} c_j)$. Since $\alpha_k \geq 1$, $c_j \geq 0$, and $\psi_1(\cdot)$ is positive on $(0, +\infty)$,

it follows that $\frac{\partial}{\partial c_\nu}\text{UPCE} > 0$. However, if the partial derivative taken is exactly for the composite set label including the singleton ground-truth, then $\frac{\partial}{\partial c_\nu}\text{UPCE} = \psi_1(\sum_{k=1}^{K}\alpha_k + \sum_{j=1}^{\eta} c_j) - \sum_{j=1}^{\eta}\mathbb{1}(y \in \mathcal{S}_j)\psi_1(\sum_{l\in\mathcal{S}_j}\alpha_l + c_j)\mathbb{1}(\nu = j)$. Since $\alpha_k \geq 1$, $c_j \geq 0$, and $\psi_1(\cdot)$ is monotonically decreasing on $(0, +\infty)$, therefore $\psi_1(\sum_{k=1}^{K}\alpha_k + \sum_{j=1}^{\eta} c_j) < \sum_{j=1}^{\eta}\mathbb{1}(y \in \mathcal{S}_j)\psi_1(\sum_{l\in\mathcal{S}_j}\alpha_l + c_j)\mathbb{1}(\nu = j)$ It follows that $\frac{\partial}{\partial c_\nu}\text{UPCE} < 0$. This outcome, which is not desirable, reveals that the optimal HENN has the potential to increase the composite evidence output, even in cases where the singleton ground-truth does not apply. Remember that in Eq.2, Subjective Logic determines the subjective opinion based on evidence. Therefore, this approach to prediction can negatively impact the quantification of uncertainty and further affect the classification accuracy that relies on evidence-related projected class probabilities.

Under the same singleton ground-truth assumption, the partial derivative for singleton evidence $\alpha_\nu(\nu \in [K])$ is:

$$
\begin{aligned}
\frac{\partial}{\partial\alpha_\nu}\text{UPCE} &= \frac{\partial}{\partial\alpha_\nu}\Big[\psi(\sum_{j=1}^{\eta}\beta_j) - \psi(\alpha_{\text{IS}}) + \sum_{j=1}^{\eta}\Big(\psi(\sum_{l\in\mathcal{S}_j}\alpha_l) - \psi(\sum_{l\in\mathcal{S}_j}\alpha_l + c_j)\Big)\mathbb{1}(y \in \mathcal{S}_j)\Big] \\
&= \psi_1(\sum_{j=1}^{\eta}\beta_j) - \psi_1(\alpha_{\text{IS}})\mathbb{1}(\nu = \text{IS}) + \sum_{j=1}^{\eta}\mathbb{1}(y \in \mathcal{S}_j)\mathbb{1}(\nu \in \mathcal{S}_j)(\psi_1(\sum_{l\in\mathcal{S}_j}\alpha_l) - \psi_1(\beta_j)).
\end{aligned}
\tag{39}
$$

Following the same strategy, if the partial derivative taken is not for the singleton ground-truth class, then $\frac{\partial\text{UPCE}}{\partial\alpha_\nu} = \psi_1(\sum_{j=1}^{\eta}\beta_j) + \sum_{j=1}^{\eta}\mathbb{1}(y \in \mathcal{S}_j)\mathbb{1}(\nu \in \mathcal{S}_j)(\psi_1(\sum_{l\in\mathcal{S}_j}\alpha_l) - \psi_1(\beta_j))$. Since $\beta_j \geq \sum_{l\in\mathcal{S}_j}\alpha_l$, the dereasing monotonicity of trigamma function gives $\psi_1(\sum_{l\in\mathcal{S}_j}\alpha_l) - \psi_1(\beta_j) > 0$. So the partial derivative $\frac{\partial\text{UPCE}}{\partial\alpha_\nu} > 0$. If the partial derivative is for the singleton ground-truth, we can rewrite the equation as $\frac{\partial\text{UPCE}}{\partial\alpha_\nu} = \big[\psi_1(\sum_{j=1}^{\eta}\beta_j) - \sum_{j=1}^{\eta}\mathbb{1}(y \in \mathcal{S}_j)\mathbb{1}(\nu \in \mathcal{S}_j)\psi_1(\beta_j)\big] + \big[\sum_{j=1}^{\eta}\mathbb{1}(y \in \mathcal{S}_j)\mathbb{1}(\nu \in \mathcal{S}_j)\psi_1(\sum_{l\in\mathcal{S}_j}\alpha_l) - \psi_1(\alpha_{\text{IS}})\mathbb{1}(\nu = \text{IS})\big]$. Noting that $\beta_j <= \sum_{j=1}^{\eta}\beta_j$ and $\alpha_{\text{IS}} < \sum_{l\in\mathcal{S}_j}\alpha_l$, so we know that $\frac{\partial\text{UPCE}}{\partial\alpha_\nu} < 0$ by decreasing monotonicity of trigamma function.

Hence, for $\forall(\mathbf{x}, \tilde{\mathbf{y}}) \in \mathcal{D}$, with fixed finite values of $\mathbf{c}$ and $\boldsymbol{\alpha}$ except for either ground-truth singleton evidence or for both the singleton ground-truth evidence and the composite evidence including the singleton ground-truth, the limits

$$
\begin{aligned}
\lim_{\alpha_{\text{IS}}\to+\infty}\text{UPCE} &= \lim_{\alpha_{\text{IS}}\to+\infty}\Big[\psi(\beta_0) - \psi(\alpha_{\text{IS}}) + \sum_{j=1}^{\eta}(\psi(\sum_{l\in\mathcal{S}_j}\alpha_l) - \psi(\beta_j))\mathbb{1}(\text{IS} \in \mathcal{S}_j)\Big] \to 0, \\
\lim_{\substack{\alpha_{\text{IS}}\to+\infty, \\ c_j\to+\infty, \\ \text{IS}\in\mathcal{S}_j}}\text{UPCE} &= \lim_{\substack{\alpha_{\text{IS}}\to+\infty, \\ c_j\to+\infty, \\ \text{IS}\in\mathcal{S}_j}}\Big[\psi(\beta_0) - \psi(\alpha_{\text{IS}}) + \sum_{j=1}^{\eta}(\psi(\sum_{l\in\mathcal{S}_j}\alpha_l) - \psi(\beta_j))\mathbb{1}(\text{IS} \in \mathcal{S}_j)\Big] \to 0.
\end{aligned}
\tag{40}
$$

hold.

Recall that $\alpha_k = e_k + 1$, and trigamma function $\psi_1(\cdot)$ is also strictly convex. Therefore, rewrite the concentration parameters as evidence, for $\forall(\mathbf{x}, \tilde{\mathbf{y}}) \in \mathcal{D}$, where $\tilde{\mathbf{y}}$ is a singleton class label $k \in [K]$, when $e_k \to +\infty$ and $e_{\mathcal{S}_i} \to +\infty$, $\forall\mathcal{S}_i \in \mathcal{S}$, such that $k \in \mathcal{S}_i$, we will have $\text{UPCE}(\mathbf{x}^{(i)}, \tilde{\mathbf{y}}^{(i)}; \boldsymbol{\theta})$ approaches the infimum 0. It is worth noting that the infimum can also be approached when solely maximizing the singleton ground-truth evidence. Hence, with different evidence predictions causing the same loss for the same learning objective, hyper-opinions derived from the evidence will also become inconsistent.

**Case 2**: Under composite ground-truth assumption.

If we assume the ground-truth is a composite class label, we expect that as the composite ground-truth evidence increases, the UPCE loss decreases, in contrast, if any singleton evidence increases, the UPCE loss should increase. Mathmatically, our goal is $\frac{\partial}{\partial c_\nu}\text{UPCE} < 0, \forall\nu \in [\eta]$, and $\frac{\partial}{\partial\alpha_\nu}\text{UPCE} \geq 0, \forall\nu \in [K]$. For composite ground-truth, the partial derivative with respect to $c_\nu, \nu \in [\eta]$ is known as:

$$\frac{\partial}{\partial c_\nu} \text{UPCE} = \frac{\partial}{\partial c_\nu} \left[ \psi(\sum_{j=1}^{\eta} \beta_j) - \psi(\beta_{\text{IC}}) \right]$$

$$= \psi_1(\sum_{j=1}^{\eta} \beta_j) - \psi_1(\beta_{\text{IC}}) \mathbb{1}(\nu = \text{IC}) \tag{41}$$

If the partial derivative taken is for a composite class label that is not the ground-truth, $\frac{\partial}{\partial c_\nu}\text{UPCE} = \psi_1(\sum_{j=1}^{\eta} \beta_j)$. Since $\beta_j > 0$, and $\psi_1(\cdot)$ is positive on $(0, +\infty)$, it follows that $\frac{\partial}{\partial c_\nu}\text{UPCE} > 0$. This means the HENN will compress non-related composite evidence. Nonetheless, the partial derivative for the composite ground-truth is $\frac{\partial}{\partial c_\nu}\text{UPCE} = \psi_1(\sum_{j=1}^{\eta} \beta_j) - \psi_1(\beta_{\text{IC}})$. Since $\sum_{j=1}^{\eta} \beta_j > \beta_{\text{IC}}$, and $\psi_1(\cdot)$ is monotonically decreasing on $(0, +\infty)$, it follows that $\psi_1(\sum_{j=1}^{\eta} \beta_j) < \psi_1(\beta_{\text{IC}}) \mathbb{1}(\nu = \text{IC})$, $\frac{\partial}{\partial c_\nu}\text{UPCE} < 0$, indicating HENN will only enlarge the evidence of ground-truth among all composite set classes during training.

Similarly, the partial derivative for singleton evidence $\alpha_\nu$ ($\nu \in [K]$) under composite ground-truth is:

$$\frac{\partial}{\partial \alpha_\nu} \text{UPCE} = \frac{\partial}{\partial \alpha_\nu} \left[ \psi(\sum_{j=1}^{\eta} \beta_j) - \psi(\beta_{\text{IC}}) \right]$$

$$= \psi_1(\sum_{j=1}^{\eta} \beta_j) - \mathbb{1}(\nu \in \mathcal{S}_{\text{IC}}) \psi_1(\beta_{\text{IC}}), \tag{42}$$

If the partial derivative taken is not for the singleton class included in the composite ground-truth, then $\frac{\partial}{\partial \alpha_\nu}\text{UPCE} = \frac{\partial}{\partial \alpha_\nu}\left[\psi(\sum_{j=1}^{\eta} \beta_j)\right] > 0$. In contrast, if the partial derivative is with respect to the singleton class included in composite ground-truth, then $\frac{\partial}{\partial \alpha_\nu}\text{UPCE} = \frac{\partial}{\partial \alpha_\nu}\left[\psi(\sum_{j=1}^{\eta} \beta_j) - \psi_1(\beta_{\text{IC}})\right]$ Since $\sum_{j=1}^{\eta} \beta_j > \beta_{\text{IC}}$, then we have $\psi_1(\sum_{j=1}^{\eta} \beta_j) < \psi_1(\beta_{\text{IC}})$, $\frac{\partial}{\partial c_\nu}\text{UPCE} < 0$, which causes the confusion. In other words, the UPCE loss guides HENN to enlarge the evidence of composite ground-turth and the singleton classes included in the ground-truth.

Now given finite fixed values of $\boldsymbol{\alpha}$ and $\mathbf{c}$ except for either the $c_{\text{IC}}$ or $c_{\text{IC}}$ with several other $\alpha_k$ included in composite ground-truth, we have the limits:

$$\lim_{c_{\text{IC}} \to +\infty} \text{UPCE} = \lim_{c_{\text{IC}} \to +\infty} \left[ \psi(\beta_0) - \sum_{j=1}^{\eta} \psi(\beta_j) \mathbb{1}(\text{IC} = j) \right] \to 0,$$

$$\lim_{\substack{c_{\text{IC}} \to +\infty, \\ \alpha_k \to +\infty, \\ k \in \mathcal{S}_{\text{IC}}}} \text{UPCE} = \lim_{\substack{c_{\text{IC}} \to +\infty, \\ \alpha_k \to +\infty, \\ k \in \mathcal{S}_{\text{IC}}}} \left[ \psi(\beta_0) - \sum_{j=1}^{\eta} \psi(\beta_j) \mathbb{1}(\text{IC} = j) \right] \to 0. \tag{43}$$

If we convert the parameters back to evidence space, then the limits show that for $\forall (\mathbf{x}, \tilde{\mathbf{y}}) \in \mathcal{D}$, where $\tilde{\mathbf{y}}$ is a composite class label $\mathcal{S}_i$, the $\text{UPCE}(\mathbf{x}^{(i)}, \tilde{\mathbf{y}}^{(i)}; \boldsymbol{\theta})$ approaches the infimum 0 as the predicted evidence values $e_{\mathcal{S}_i} \to +\infty$ and $e_k \to +\infty, \forall k \in S_i$. The infimum value can also be approached by only maximizing the composite evidence for the ground truth. Again, with different evidence, predictions correspond to the same empirical loss for the same composite learning objective, causing unreliable issues in subjective opinion modeling.

After proving 2 cases of inconsistent evidence predictions regarding the optimal loss minimizer, we prove that the risk minimizer can be approximated by a UPCE loss minimizer for each training data point. This step is crucial for connecting the empirical risk minimizer with the loss minimizer and for highlighting the inconsistency in evidence predictions made by the empirical risk minimizer HENN. Under the assumption of universal approximation property (UAP) (Cybenko, 1989; Leshno et al., 1993a), suppose the HENN has the capability to produce sufficient non-linearity to estimate any functions in the evidential space, we can have at least one optimal HENN $f(\cdot; \boldsymbol{\theta}^\star)$ (or $f^\star$ for short) such that

$$R(f^\star) = \inf_{\boldsymbol{\theta} \in \Theta} R(f) = \inf_{\boldsymbol{\theta} \in \Theta} \left[ \frac{1}{N} \sum_{i=1}^{N} \left[ \text{UPCE}(\mathbf{x}^{(i)}, \tilde{\mathbf{y}}^{(i)}; \boldsymbol{\theta}) \right] \right] = \frac{1}{N} \sum_{i=1}^{N} \left[ \inf_{\boldsymbol{\theta} \in \Theta} \text{UPCE}(\mathbf{x}^{(i)}, \tilde{\mathbf{y}}^{(i)}; \boldsymbol{\theta}) \right]. \tag{44}$$

This equation connects the empirical risk minimizer and the loss minimizer on each training data. To prove Eq.(44), we apply the assumed UAP. Note that $\text{UPCE}(\mathbf{x}^{(i)}, \tilde{\mathbf{y}}^{(i)}; \boldsymbol{\theta}) = \text{UPCE}(f(\mathbf{x}^{(i)}; \boldsymbol{\theta}), \tilde{\mathbf{y}}^{(i)})$. We abbreviate them by $\text{UPCE}^{(i)}$ for simplicity. Since $\text{UPCE}(\mathbf{x}, \tilde{\mathbf{y}}; \boldsymbol{\theta}) \geq \inf \text{UPCE}(\mathbf{x}, \tilde{\mathbf{y}}; \boldsymbol{\theta})$, we have $\frac{1}{N} \sum_{i=1}^{N} \left[ \text{UPCE}^{(i)} \right] \geq \frac{1}{N} \sum_{i=1}^{N} \left[ \inf \text{UPCE}^{(i)} \right]$, and a trivial conclusion is $\left[ \inf \frac{1}{N} \sum_{i=1}^{N} \left[ \text{UPCE}^{(i)} \right] \right] \geq \frac{1}{N} \sum_{i=1}^{N} \left[ \inf \text{UPCE}^{(i)} \right]$.

Now recall the universal approximation property demonstrates the existence of a function that can approximate any function within the same function space. Applying assumed UAP to our setting, it states for any $(\mathbf{x}, \tilde{\mathbf{y}}) \in \mathcal{D}$, and arbitrary function $g(\mathbf{x}, \tilde{\mathbf{y}}; \boldsymbol{\theta}) = \inf \text{UPCE}(f(\mathbf{x}; \boldsymbol{\theta}), \tilde{\mathbf{y}})$, there exists an optimal HENN can approximate $g(\cdot)$ by mapping input features to evidence $f(\mathbf{x}; \boldsymbol{\theta}^{\star})$. s.t.

$$\sup_{\mathbf{x}, \tilde{\mathbf{y}}} \| g(\mathbf{x}, \tilde{\mathbf{y}}; \boldsymbol{\theta}) - \text{UPCE}(f(\mathbf{x}; \boldsymbol{\theta}^{\star}), \tilde{\mathbf{y}}) \| < \epsilon, \quad \forall \epsilon > 0. \tag{45}$$

Because of the relation $\beta_0 > \beta_{\text{IC}}$ and $\alpha_{\mathcal{S}_j} > \alpha_{\text{IS}}$, the form of UPCE loss based on digamma functions in Eq.(12) determines its positive value, according to the increasing monotonicity of digamma function on $(0, +\infty)$.

Given the limits shown in Eq.(40) and Eq.(43), we know that the infimum of UPCE is 0, Eq.(45) can be rewritten as:

$$\sup_{\mathbf{x}, \tilde{\mathbf{y}}} \text{UPCE}(f(\mathbf{x}; \boldsymbol{\theta}^{\star}), \tilde{\mathbf{y}}) < \epsilon, \quad \forall \epsilon > 0. \tag{46}$$

Based on the inequality

$$0 < \frac{1}{N} \sum_{i=1}^{N} \left[ \inf \text{UPCE}^{(i)} \right] \leq \left[ \inf \frac{1}{N} \sum_{i=1}^{N} \left[ \text{UPCE}^{(i)} \right] \right] < \frac{1}{N} \sum_{i=1}^{N} \epsilon = \epsilon, \tag{47}$$

with both lower bound and upper bound as 0, according to the squeeze theorem, the exchangeability of inf operators $\frac{1}{N} \sum_{i=1}^{N} \left[ \inf \text{UPCE}(\mathbf{x}, \tilde{\mathbf{y}}; \boldsymbol{\theta}) \right] = \inf \frac{1}{N} \sum_{i=1}^{N} \text{UPCE}(\mathbf{x}, \tilde{\mathbf{y}}; \boldsymbol{\theta}) = 0$ holds for each training data point, and the Eq.(44) is proved.

Knowing the existence of an empirical minimizer for all observations also works as the loss minimizer on each training data point, the HENN should always predict evidence $f(\mathbf{x}; \boldsymbol{\theta}^{\star}) = (\tilde{\boldsymbol{\alpha}}, \tilde{\mathbf{c}}), \forall (\mathbf{x}, \tilde{\mathbf{y}}) \in \mathcal{D}$, s.t.

$$\text{UPCE}(f(\mathbf{x}; \boldsymbol{\theta}^{\star}), \tilde{\mathbf{y}}) \to \inf \text{UPCE} = 0, \quad \forall (\mathbf{x}, \tilde{\mathbf{y}}) \in \mathcal{D} \tag{48}$$

We can conclude that the properties derived from the analysis of the UPCE loss $\text{UPCE}(\mathbf{x}, \tilde{\mathbf{y}}; \boldsymbol{\theta})$ for arbitrary $(\mathbf{x}, \tilde{\mathbf{y}}) \sim \mathcal{D}$ also holds for the HENN with empirical risk $R(f)$ under the assumption of UAP.

$\square$

## C.3 PROOF OF PROPOSITION 2

**Proposition 2** (Effectiveness of the regularization term $\text{Reg}(\mathbf{x}, \tilde{\mathbf{y}}; \boldsymbol{\theta})$). *Following the UAP assumption, the regularized empirical UPCE risk defined in Eq. (15) approaches the infimum 0 if the solution $\boldsymbol{\theta}^{\star}$ satisfies the following properties: 1) $\forall (\mathbf{x}, \tilde{\mathbf{y}}) \in \mathcal{D}$, where $\tilde{\mathbf{y}}$ is a singleton class label $k \in [K]$, the predicted evidence values $e_k \to +\infty$ and $e_t \to 0, \forall t \in \boldsymbol{\mathcal{S}} \cup [K] \setminus k$; and 2) $\forall (\mathbf{x}, \tilde{\mathbf{y}}) \in \mathcal{D}$, where $\tilde{\mathbf{y}}$ denotes a composite set label $\mathcal{S}_i$, the predicted evidence values $e_{\mathcal{S}_i} \to +\infty$ and $e_t \to 0, \forall t \in \boldsymbol{\mathcal{S}} \cup [K] \setminus \mathcal{S}_i$.*

*Proof.* To address the inconsistent prediction issue of our evidence output, the KL-divergence between the predicted GDD and a flat GDD is introduced as a regularizer. All 0 evidence for each element in the hyper-domain composes a flat GDD as $\text{GDD}(\mathbf{p}|\mathbf{1}^K, \mathbf{0}^{\bar{\eta}})$. In following section, for simplicity, we abbreviate $\text{KL}\left[ \text{GDD}(\mathbf{p}|\bar{\boldsymbol{\alpha}}^{(i)}, \bar{\mathbf{c}}^{(i)}) \| \text{GDD}(\mathbf{p}|\mathbf{1}^K, \mathbf{0}^\eta) \right]$ by $\text{KL}(\bar{\boldsymbol{\alpha}}^{(i)}, \bar{\mathbf{c}}^{(i)})$, $\text{UPCE}(\mathbf{x}^{(i)}, \tilde{\mathbf{y}}^{(i)}; \boldsymbol{\theta})$ by $\text{UPCE}^{(i)}$, and let $[K]$ denote $\{1, ..., K\}$, $[\eta]$ denote $\{1, ..., \eta\}$. Now the optimal regularized generalization risk is

$$R(f^*) \to \inf \left[ \frac{1}{N} \sum_{i=1}^{N} \left[ \text{UPCE}(\mathbf{x}^{(i)}, \tilde{\mathbf{y}}^{(i)}; \boldsymbol{\theta}) + \lambda \cdot \text{KL}(\bar{\boldsymbol{\alpha}}^{(i)}, \bar{\mathbf{c}}^{(i)}) \right] \right]$$

$$= \inf \left[ \frac{1}{N} \sum_{i=1}^{N} \left[ \text{UPCE}^{(i)} + \lambda \cdot \text{KL}(\bar{\boldsymbol{\alpha}}^{(i)}, \bar{\mathbf{c}}^{(i)}) \right] \right] \tag{49}$$

$$= \inf \left[ \frac{1}{N} \sum_{i=1}^{N} \text{UPCE}^{(i)} + \lambda \cdot \left[ \frac{1}{N} \sum_{i=1}^{N} \text{KL}(\bar{\boldsymbol{\alpha}}^{(i)}, \bar{\mathbf{c}}^{(i)}) \right] \right].$$

According to the partial derivatives and the convexity of UPCE loss proved in section C.2, we already have two special limits for singleton ground truth as shown in Eq.(40). Correspondingly, to make UPCE loss approach its infimum value with composite ground-truth, there are also two special limits mentioned in Eq.(43). Multiple choices to minimize the UPCE loss imply different combinations of $\boldsymbol{\alpha}$ and $\mathbf{c}$ can become the loss minimizer and output by HENN.

Consider the KL-divergence between predicted GDD and flat GDD

$$\text{KL}(\bar{\boldsymbol{\alpha}}^{(i)}, \bar{\mathbf{c}}^{(i)}) = \int_{\Delta_K} \text{GDD}(\mathbf{p}|\bar{\boldsymbol{\alpha}}^{(i)}, \bar{\mathbf{c}}^{(i)}) \log \frac{\text{GDD}(\mathbf{p}|\bar{\boldsymbol{\alpha}}^{(i)}, \bar{\mathbf{c}}^{(i)})}{\text{GDD}(\mathbf{p}|\mathbf{1}^K, \mathbf{0}^\eta)} d\mathbf{p}. \tag{50}$$

It is straightforward to have its minimizer when the value of $\log \frac{\text{GDD}(\mathbf{p}|\bar{\boldsymbol{\alpha}}^{(i)}, \bar{\mathbf{c}}^{(i)})}{\text{GDD}(\mathbf{p}|\mathbf{1}^K, \mathbf{0}^\eta)}$ is 0. This indicates that we aim to minimize the difference between $\bar{\boldsymbol{\alpha}}^{(i)}$ and $\mathbf{1}^K$, as well as between $\bar{\mathbf{c}}^{(i)}$ and $\mathbf{0}^\eta$. Predicting flat GDD except for the evidence of the ground-truth to make the KL-divergence reach the minimum value of 0. Therefore,

$$\arg\min \text{KL}(\bar{\boldsymbol{\alpha}}^{(i)}, \bar{\mathbf{c}}^{(i)})$$
$$= \left\{ (\boldsymbol{\alpha}, \mathbf{c}) : \alpha_k = 1, k \neq \text{IS}, k \in [K], \mathbf{c} = \mathbf{0}^\eta \right\} \cup \left\{ (\boldsymbol{\alpha}, \mathbf{c}) : \boldsymbol{\alpha} = \mathbf{1}^K, c_j = 0, j \neq \text{IC}, j \in [\eta] \right\}. \tag{51}$$

Note that the feasible region of the output evidence for minimizers of UPCE loss and the KL-divergence overlaps, which illustrates that both infimums can be approached simultaneously. Specifically, for singleton ground-truth, the intersection of feasible evidence between KL-divergence minimizer and UPCE loss minimizer is $\{(\boldsymbol{\alpha}, \mathbf{c}) : \alpha_k = 1, \alpha_{\text{IS}} \to +\infty, k \neq \text{IS}, k \in [K], \mathbf{c} = \mathbf{0}^\eta\}$. In contrast, the intersection for composite ground-truth can be written as $\{(\boldsymbol{\alpha}, \mathbf{c}) : \boldsymbol{\alpha} = \mathbf{1}^K, c_j = 0, c_{\text{IC}} \to +\infty, j \neq \text{IC}, j \in [\eta]\}$.

The overlap of feasible evidence towards the lower bound of the UPCE loss, along with its regularizer, also enables the application of the Uniform Approximation Property (UAP) to the regularizer, with $\inf \left[ \frac{1}{N} \sum_{i=1}^{N} \text{Reg} \right] = \frac{1}{N} \sum_{i=1}^{N} \left[ \inf \text{Reg} \right]$. Based on the assumed UAP, there exists configuration $\boldsymbol{\theta}'$ such that HENN is an empirical UPCE loss minimizer for each training data point. Within the feasible evidence region for minimizing the UPCE loss with $\boldsymbol{\theta}'$, the learning objective is improved when considering the overlap in evidence outputs. This suggests the presence of an optimal configuration $\boldsymbol{\theta}^\star$ within the feasible range of $\boldsymbol{\theta}'$ that can attain the minimal KL-divergence at every training data point without hurting the optimality for empirical UPCE risk. By focusing on learning $\boldsymbol{\theta}^\star$, we finally can get the optimal regularized HENN given UAP assumption holds.

Therefore, we can move the infimum operator into the empirical risk,

$$R(f^*) \to \inf \left[ \frac{1}{N} \sum_{i=1}^{N} \text{UPCE}^{(i)} + \lambda \cdot \left[ \frac{1}{N} \sum_{i=1}^{N} \text{KL}(\bar{\boldsymbol{\alpha}}^{(i)}, \bar{\mathbf{c}}^{(i)}) \right] \right]$$

$$= \frac{1}{N} \sum_{i=1}^{N} \left[ \inf \text{UPCE}^{(i)} + \lambda \cdot \inf \text{KL}(\bar{\boldsymbol{\alpha}}^{(i)}, \bar{\mathbf{c}}^{(i)}) \right], \tag{52}$$

As proved in section C.2, based on the assumption of UAP, there exists a regularized loss minimizer for each data point, which also works as the empirical regularized minimizer for $\mathcal{D} = \{(\mathbf{x}^{(i)}, \tilde{\mathbf{y}}^{(i)})\}_{i=1}^{N}$.

Table 5: Dataset Statistic.

| Dataset | CIFAR100 | tinyImageNet | Living17 | Nonliving26 |
|---|---|---|---|---|
| Image Resolution | $32\times32$ | $64\times64$ | $224\times224$ | $224\times224$ |
| # superclasses | 20 | 29 | 17 | 26 |
| # subclasses | 100 | 200 | 68 | 104 |
| Training set size | 45k | 90k | 79.56k | 119.5k |
| Validation set size | 5k | 10k | 8.84k | 13.3k |
| Test set size | 10k | 10k | 3.4k | 5.2k |
| # SELECTED composite classes | {20,15,10} | {20,15,10} | {15,10} | {20,15,10} |

---

**Algorithm 1** Pseudo-code of HENN (one epoch)

---

**Require:** Training dataset $\mathcal{D} = \{(\mathbf{x}^{(i)}, \tilde{\mathbf{y}}^{(i)})\}|_{i=1}^{N}$; HENN model $f(\cdot, \boldsymbol{\theta})$; tradeoff coefficient $\lambda$; learning rate $\gamma$; the number of sampling data $N$; Batch size: $|B|$;

1: Initialize model parameters $\boldsymbol{\theta}$.
2: **for** $iter$ = 1, 2, ... , **do**
3:    Sample a mini-batch $B$ from $\mathcal{D}$
4:    Generate the evidence vector $\mathbf{e}^{(i)}|_{i=1}^{|B|}$ ($\mathbf{e} \in \mathbb{R}^{|B|\times\kappa}$): $\mathbf{e}^{(i)} = f(\mathbf{x}^{(i)}, \boldsymbol{\theta})$
5:    **for** each $(\mathbf{x}^{(i)}, \tilde{\mathbf{y}}^{(i)}) \in B$ **do**
5:       `//based on Grouped Dirichlet Distribution (GDD)`
6:       Get the UPCE loss for this example $\text{UPCE}^{(i)}(\boldsymbol{\theta})$ via Eq. 12
7:       Get the entropy regularization for this example $\text{Reg}^{(i)}(\boldsymbol{\theta})$ via Eq. 14
8:       Get the loss for this example: $\mathcal{L}^{(i)}(\boldsymbol{\theta}) = \text{UPCE}^{(i)}(\theta) + \lambda\text{Reg}^{(i)}(\boldsymbol{\theta})$
9:    **end for**
10:   Get the loss $\mathcal{L}$ for all examples in this batch B: $\mathcal{L}(\boldsymbol{\theta}) = \frac{1}{|B|}\sum_{i=1}^{|B|}\mathcal{L}^{(i)}(\boldsymbol{\theta})$.
11:   Update model parameters $\boldsymbol{\theta}$ via gradient descent $\boldsymbol{\theta}' = \boldsymbol{\theta} - \gamma\nabla\mathcal{L}(\boldsymbol{\theta})$
12: **end for**

---

By replacing the empirical risk minimizer with the regularized loss minimizer. We focus on loss minimizer that produces:

$$\text{UPCE} + \lambda \cdot \text{Reg} \to \inf\left[\text{UPCE} + \lambda \cdot \text{Reg}\right] = \inf \text{UPCE} + \lambda \cdot \inf \text{Reg} \tag{53}$$

Clearly, optimal regularized HENN $f(\mathbf{x}; \boldsymbol{\theta}) = (\tilde{\boldsymbol{\alpha}}, \tilde{\mathbf{c}})$ will take the intersection of the feasible space for approaching minimal UPCE loss and regularizer, that is, $(\tilde{\boldsymbol{\alpha}}, \tilde{\mathbf{c}}) = \{(\boldsymbol{\alpha}, \mathbf{c}) : \alpha_k = 1, k \neq \text{IS}, k \in [K], \alpha_{\text{IS}} \to +\infty, \mathbf{c} = \mathbf{0}^\eta\} \cup \{(\boldsymbol{\alpha}, \mathbf{c}) : \boldsymbol{\alpha} = \mathbf{1}^K, c_j = 0, j \neq \text{IC}, j \in [\eta], c_{\text{IC}} \to +\infty\}$. Convert parameter space back to evidence space, then we can say for $\forall(\mathbf{x}, \tilde{\mathbf{y}}) \in \mathcal{D}$ where $\tilde{\mathbf{y}}$ is a singleton class label $k \in [K]$, the predicted evidence has the form of $e_k \to +\infty, e_t \to 0, \forall t \in \boldsymbol{\mathcal{S}} \cup [K] \setminus k$. For $\forall(\mathbf{x}, \tilde{\mathbf{y}}) \in \mathcal{D}$, where $\tilde{\mathbf{y}}$ denotes a composite class label $\mathcal{S}_i$, the predicted evidence should be $e_{\mathcal{S}_i} \to +\infty$ and $e_t \to 0, \forall t \in \boldsymbol{\mathcal{S}} \cup [K] \setminus \mathcal{S}_i$.

$\square$

## D    RELATIONS WITH ALEATORIC AND EPISTEMIC UNCERTAINTIES

Epistemic and aleatoric uncertainties are two broad categories used to classify existing predictive uncertainty measures. Epistemic uncertainty is due to a lack of evidence or knowledge in the training data – it is a *known unknown*. It is *reducible* by collecting more data. In comparison, aleatoric uncertainty is due to the inherent complexity of the data (e.g., wrong labels, incomplete or partial labels, and other data randomness) – it is a *unknown unknown*. It is *irreducible* by collecting more data (e.g., the stochasticity of a dice roll cannot be reduced by observing more rolls), assuming the same measurement precision in the collected data (Gal, 2016). The aforementioned evidential uncertainties, including vacuity, vagueness, and dissonance, and other uncertainty measures, such as model uncertainty (mutual information between model parameters and the predicted class probabilities), data uncertainty (entropy of the predicted class probabilities), and confidence (the largest predicted class probability) can be classified to epistemic and aleatoric uncertainties based

on whether they can be reduced by collecting more data. In particular, the vacuity and model uncertainty fall into the category of epistemic uncertainty, and the dissonance and vagueness belong to the category of aleatoric uncertainty. The dissonance is irreducible by collecting more conflicting evidence. The vagueness is irreducible when we use the same measurement precision (sensor and annotator) to collect extra training data due to the invariant underlying distribution for getting composite labels. A recent work (Shi et al., 2020) demonstrates that the entropy of the predicted class probabilities can be decomposed into two distinct sources of uncertainty: vacuity and dissonance. As confidence is correlated with this entropy, both data uncertainty and confidence may involve a mixture of epistemic and aleatoric uncertainties.

### D.1 EXAMPLE ABOUT EVIDENCE

In medical diagnostics, the presence of 24 pieces of composite evidence could suggest that there are approximately 24 similar cases resulting in diseases $2, 3$ based on the current observation. This implies that the cases are identified as having either disease 2 or 3, but without specific information to distinguish between them. Conversely, 3 instances of class 1 evidence indicate that 3 similar cases have been identified as disease 1. In such scenarios, doctors might not have a clear preference between diseases 2 and 3, while maintaining a conflicting opinion between disease 1 and $\{2, 3\}$ for this observation.

### D.2 DISSONANCE IN HYPER-OPINION

Given a hyper-opinion with non-zero belief masses, the dissonance measure can be estimated as:

$$diss(\omega) = \sum_{\mathcal{S} \in \mathscr{R}(\mathbb{Y})} \left( \frac{b_{\mathcal{S}} \sum_{\mathcal{S}' \in \mathscr{R}(\mathbb{Y}), \mathcal{S}' \neq \mathcal{S}} \mathrm{d}(\mathcal{S} \triangle \mathcal{S}') b_{\mathcal{S}'} \mathrm{Bal}\left(b_{\mathcal{S}'}, b_{\mathcal{S}}\right)}{\sum_{\mathcal{S}' \in \mathscr{R}(\mathbb{Y}), \mathcal{S}' \neq \mathcal{S}} \mathrm{d}(\mathcal{S} \triangle \mathcal{S}') b_{\mathcal{S}'}} \right) \tag{54}$$

where $\mathrm{Bal}(\mathcal{S}', \mathcal{S}) = 1 - |b_{\mathcal{S}'} - b_{\mathcal{S}}|/(b_{\mathcal{S}'} + b_{\mathcal{S}})$, and $\mathrm{d}(\mathcal{S} \triangle \mathcal{S}')$ is the size of the symmetric difference between $\mathcal{S}$ and $\mathcal{S}'$ (Jøsang et al., 2018).

## E REPRODUCIBILITY

### E.1 DATASET

Table 5 shows detailed statistics for four datasets we used. In particular, tinyImageNet has 29 superclasses because we keep all superclasses which have 2-3 subclasses only.

We use **CIFAR100** (Krizhevsky & Hinton, 2009), **tinyImageNet** (Fei-Fei et al., 2015), **Living17** (Santurkar et al., 2021), and **Nonliving26** (Santurkar et al., 2021) in our experiments. CIFAR100 has 100 classes containing 600 images each (500 for training and 100 for testing, and the image size is $32 \times 32$). The 100 classes in this dataset are divided into 20 disjoint superclasses, each with 5 unique subclasses. Note that we compose composite class labels within the same superclass. Dataset tinyImageNet has 200 classes containing 550 images each (500 for training and 50 for testing, and the image size is $64 \times 64$). We generate the hierarchy information of tinyImageNet and generate superclasses according to the existing ImageNet class hierarchy - WordNet (Miller, 1995). In addition, it usually can be challenging to distinguish between different classes due to their similar visual features. While WordNet is a hierarchy based on semantic relationships between words, rather than visual similarities. Therefore, Living17 and Nonliving26 are considered because their class hierarchy is generated based on visual and semantic similarities. Both of them are subsets of ImageNet dataset (Deng et al., 2009) with an image size $224 \times 224$. Refer to Table 5 and 6 in their paper for more information.

We split the original training set into a training and a validation set according to the ratio 9:1. Therefore, the number of images per class will be: 450/50/50 for training/validation/test set for CIFAR100, similarly for other datasets.

### E.2 DATASET PREPROCESSING

For each dataset, the first step is to select vague images. To achieve that, first, we select $M$ superclasses randomly from all superclass candidates as SELECTED composite classes in our experiments. For

each SELECTED composite class, 2 or more subclasses belonging to this superclass will be selected randomly as components of the composite class label. Given the designed composite class labels, we can further select a fraction of images under each of the singleton classes included in the domain of composite classes $\mathscr{C}(\mathbb{Y})$. The selected examples are therefore expected to be converted to composite examples by applying Gaussian-blurring and label replacement to introduce vagueness. When selecting images to blur, for each selected singleton class, we balanced the number of singleton images remaining and the number of composite examples converted. Please check our code for implementation [1].

The selected vague examples will be blurred by Gaussian Blurring operation. To apply the Gaussian blur operation, there are two parameters to set: `kernel_size` and variance `sigma`. We use three different `kernel_size`s ($3 \times 3, 5 \times 5, 7 \times 7$), and `sigma` is determined by the default relation between them in PyTorch: `sigma` $= 0.3 * ((\text{kernel\_size} - 1) * 0.5 - 1) + 0.8$ [2].

We used 2 methods for data augmentation following a typical computer vision setting. First, each image is applied to a random horizontal flip with the flipping probability of $0.5$. After that, a random corp is introduced for each image with a size of $32 \times 32$ and padding of 4. Then, resize images to $224 \times 224$ because the pretrained model is trained by ImageNet (Deng et al., 2009) which is $224 \times 224$, we need to match the input size for model predictions. We apply regular data augmentation approaches and normalization to the data. Data augmentation approaches are only applied to the training set. For validation and test sets, we only use resize and normalization.

### E.3 IMPLEMENTATION

**Baselines.** DNN and ENN cannot predict set directly. In practice, it is necessary to set a threshold to make set prediction for DNN and ENN. The prediction set should consist of all classes with softmax probabilities larger than or equal to the pre-defined threshold.

In addition, DNN and ENN are only able to deal with singleton class labeled examples and cannot deal with composite class label during training. Note that there are vague images with composite class labels during training. To make baselines can handle these examples, and to avoid removing training examples, we duplicate composite examples and provide them singleton class labels which are from the subclasses of composite class labels. This ensures that all classes remain exclusive. For example, assuming there is an image $x$ with the composite class label A,B during training, we duplicate $x$ and take image $x$ with the singleton class label A and the same image with the singleton class label B as input for model training.

**HENN:** Pseudo-code of HENN is shown in Algorithm 1.

### E.4 HYPERPARAMETERS TUNING

We list all related methods and their corresponding hyperparameter settings below. For our method and all other baselines, we adopt Adam (Kingma & Ba, 2014) as optimizer with parameters $\beta_1 = 0.9$, $\beta_2 = 0.999$, weight decay is 0, $\epsilon = 1e - 8$ provided in (Kingma & Ba, 2014). The number of epochs for all experiments is set to 100. Other hyperparameters used in this paper mainly are learning rate and weight of entropy regularizer. Grid search is leveraged to determine the best hyperparameters based on a held-out validation set for each specific experiment. Specifically, (1) **DNN**. the learning rate is chosen from {1e-5, 1e-4, 1e-3}; the cutoff is chosen from {0, 0.05, 0.1, 0.15, 0.2, 0.25, 0.3, 0.35, 0.4, 0.45, 0.5}. (2) **ENN**. the learning rate is chosen from {1e-5, 1e-4, 1e-3}; the weight of entropy regularizer $\lambda$ is chosen from {1, 1e-1, 1e-2, 1e-3, 1e-4, 1e-5}. the cutoff is chosen from `[i for i in range(0, 0.02, 0.001)]`. (3) **E-CNN**. the learning rate is chosen from {1e-5, 1e-4, 1e-3}; the optimizer is Nadam. (4) **RAPS**. $k_{reg}$ is chosen from {1, 2, 5, 10, 50}; $\lambda$ is chosen from {0, 1e-4, 1e-3, 0.01, 0.02, 0.05, 0.2, 0.5, 0.7, 1}; $\alpha$ is chosen from {0.1, 0.2, 0.3, 0.4}. (5) **PiCO**. learning rate is chosen from {1e-5, 1e-4, 1e-3}. (6) **HENN**. the learning rate is chosen from {1e-5, 1e-4, 1e-3} and the weight of regularizer $\lambda$ is chosen from {1, 1e-1, 1e-2, 1e-3, 1e-4, 1e-5}.

---

[1]Our code: https://github.com/Hugo101/HyperEvidentialNN
[2]https://pytorch.org/vision/main/generated/torchvision.transforms.functional.gaussian_blur.html

A fixed number of epochs is given, and the highest validation accuracy is used to determine the best epoch.

For HENN, validation accuracy means the classification accuracy including the additional composite class labels on hyperdomain, such as 215-class classification in tinyImageNet dataset. Even though we report multiple metrics, such as OverJS, CompJS, and Acc, we use validation accuracy to select the model. We use the Best validation accuracy to evaluate and determine which combination of hyperparameters to use.

For DNN, there are two sets of hyperparameters. The first set includes the hyperparameter of general DNN: learning rate (we only tune this hyperparmeter for now). The second set includes the hyperparameter used to generate set prediction: the cutoff on class probability. For example, if there are only three classes and the prediction of the DNN for one test image is: $\{0.6, 0.3, 0.1\}$. If the cutoff is 0.3, then the set is: $\{$class 1, class 2$\}$.

For the first set, use the accuracy on the validation set to tune. Note that it is always 100-class classification after using duplicates for vague examples. Each duplicated image has its own class label. For example, for one training/validation image that has a set label: class 1, class 3. We will create two duplicates of this image labeled class 1 and class 3, respectively. For the second set, use the overJS on the validation set to tune. Here, we will replace duplicates with the images with vague labels in the validation set, in order to calculate the overJS.

## F ADDITIONAL EXPERIMENTAL RESULTS

### F.1 ADDITIONAL RESULTS

Table 6, 11, 12 show composite and singleton prediction results for different Gaussian kernel size $3\times3$, $5\times5$, $7\times7$ for CIFAR100 and tinyImageNet dataset, and Table 8, 9, 10 show composite and singleton prediction results for living17 and nonliving26 dataset, which represents consistent observation as in main paper.

Table 6: Results (%) based on Gaussian kernel size: $3\times3$ on CIFAR100 and tinyImageNet. (The average and 95% confidence interval of three runs are provided.)

| $M$ | Methods | CIFAR100 | | | tinyImageNet | | |
|---|---|---|---|---|---|---|---|
| | | OverJS | CompJS | Acc | OverJS | CompJS | Acc |
| 10 | DNN (Tan & Le, 2019) | **86.8**±0.36 | 68.6±1.42 | 84.3±0.51 | 83.4±0.38 | 66.9±0.93 | 79.8±0.32 |
| | ENN (Sensoy et al., 2018) | 84.4±0.28 | 42.3±1.23 | 84.8±0.22 | 75.9±0.31 | 63.5±1.26 | 80.7±0.27 |
| | E-CNN (Tong et al., 2021) | 38.5±0.74 | 34.2±2.63 | 73.2±0.92 | 33.4±0.83 | 31.1±2.38 | 68.2±0.92 |
| | RAPS (Angelopoulos et al., 2021) | 81.5±0.33 | 51.1±1.41 | 84.3±0.51 | 73.1±0.37 | 43.6±0.96 | 79.8±0.32 |
| | PiCO (Wang et al., 2022b) | 59.6±0.38 | 28.3±4.41 | 63.6±0.48 | 57.2±0.39 | 35.6±3.53 | 64.3±0.63 |
| | HENN (ours) | 86.5±0.47 | **90.4**±3.63 | **86.5**±0.53 | **84.4**±0.44 | **93.4**±2.57 | **82.5**±0.72 |
| 15 | DNN (Tan & Le, 2019) | 86.6±0.35 | 71.6±1.43 | 82.2±0.39 | 84.3±0.43 | 67.3±1.43 | 79.5±0.35 |
| | ENN (Sensoy et al., 2018) | 84.2±0.27 | 47.8±1.25 | 83.8±0.37 | 83.5±0.20 | 60.7±1.14 | 81.2±0.26 |
| | E-CNN (Tong et al., 2021) | 33.2±0.74 | 31.3±3.43 | 68.6±0.93 | 32.5±0.83 | 33.3±3.52 | 68.4±0.95 |
| | RAPS (Angelopoulos et al., 2021) | 81.5±0.36 | 54.1±1.44 | 82.2±0.39 | 68.1±0.44 | 45.6±1.52 | 79.5±0.35 |
| | PiCO (Wang et al., 2022b) | 58.4±0.74 | 25.5±4.32 | 61.3±0.50 | 56.8±0.38 | 35.3±3.53 | 64.6±0.64 |
| | HENN (ours) | **86.8**±0.28 | **90.1**±4.36 | **85.8**±0.19 | **84.6**±0.45 | **90.6**±2.61 | **81.6**±0.71 |
| 20 | DNN (Tan & Le, 2019) | **86.8**±0.35 | 75.4±1.65 | 80.3±0.35 | 84.0±0.33 | 57.9±1.06 | 81.5±0.36 |
| | ENN (Sensoy et al., 2018) | 83.3±0.23 | 53.7±1.14 | 81.9±0.19 | 57.4±0.29 | 41.9±1.11 | 58.9±0.36 |
| | E-CNN (Tong et al., 2021) | 28.6±0.78 | 23.7±3.25 | 73.6±0.87 | 23.3±0.86 | 22.4±2.51 | 67.8±0.97 |
| | RAPS (Angelopoulos et al., 2021) | 80.5±0.35 | 56.7±1.54 | 80.3±0.35 | 76.1±0.42 | 41.1±1.47 | 81.5±0.36 |
| | PiCO (Wang et al., 2022b) | 57.5±0.71 | 29.1±4.45 | 61.9±0.56 | 57.5±0.41 | 39.6±3.66 | 65.3±0.71 |
| | HENN (ours) | **86.7**±0.17 | **90.2**±1.36 | **86.3**±0.34 | **84.9**±0.40 | **90.7**±2.87 | **81.7**±0.69 |

### F.2 MODEL-AGNOSTIC PROPERTY

Table 7 shows model agnostic performance (%) on $M$=10, and kernel size: $5\times5$ on CIFAR100, including confidence interval for three different runs. It demonstrates different methods' performance based on ResNet50 and VGG16 on CIFAR100. HENN outperforms other approaches, for example, the Acc of HENN surpasses that of DNN by 2% for CIFAR100. The consistent observation is demonstrated based on different backbones, which validates the model agnostic property of our proposed approach.

Table 7: Model agnoistic performance (%) on $M$=10, and kernel size: 5×5 on CIFAR100. (The average and 95% confidence interval of three runs are provided.)

| Methods | ResNet50 (He et al., 2015) | | | VGG16 (Simonyan & Zisserman, 2015) | | |
|---|---|---|---|---|---|---|
| | OverallJS | CompJS | Acc | OverallJS | CompJS | Acc |
| DNN | 82.0±0.26 | 56.7±1.29 | 80.6±0.21 | 77.6±0.32 | 53.3±1.35 | 75.2±0.38 |
| ENN (Sensoy et al., 2018) | 80.1±0.28 | 46.7±1.32 | 80.9±0.25 | 74.6±0.33 | 42.7±1.41 | 76.2±0.43 |
| RAPS (Angelopoulos et al., 2021) | 71.8±0.26 | 40.1±1.31 | 80.6±0.21 | 66.4±0.34 | 35.5±1.38 | 75.2±0.38 |
| HENN (ours) | **82.9**±0.34 | **85.7**±2.41 | **81.1**±0.32 | **78.4**±0.37 | **78.5**±2.83 | **77.7**±0.47 |

Table 8: Results (%) of BREEDS-living17 based on two Gaussian kernel sizes. (The average and 95% confidence interval of three runs are provided.)

| $M$ | Methods | Gaussian kernel size: 3×3 | | | Gaussian kernel size: 5×5 | | |
|---|---|---|---|---|---|---|---|
| | | OverJS | CompJS | Acc | OverJS | CompJS | Acc |
| 10 | DNN (Tan & Le, 2019) | 88.1±0.28 | 81.0±1.74 | 83.3±0.29 | 88.4±0.33 | 80.4±0.78 | 83.2±0.43 |
| | ENN (Sensoy et al., 2018) | 88.0±0.19 | 72.3±0.41 | 84.5±0.12 | 88.0±0.16 | 70.9±1.07 | 84.6±0.01 |
| | E-CNN (Tong et al., 2021) | 30.5±0.67 | 36.8±1.34 | 65.7±0.86 | 30.4±1.34 | 35.8±0.88 | 65.7±0.42 |
| | RAPS (Angelopoulos et al., 2021) | 86.4±0.27 | 61.3±1.56 | 83.3±0.29 | 85.8±0.33 | 60.7±0.89 | 83.2±0.43 |
| | HENN (ours) | **88.8**±0.39 | **96.5**±0.72 | **85.6**±1.24 | **88.7**±0.35 | **96.9**±0.81 | **85.9**±0.33 |
| 15 | DNN (Tan & Le, 2019) | 88.1±0.39 | 84.8±1.62 | 80.2±0.34 | 88.4±0.23 | 84.5±1.08 | 80.6±0.48 |
| | ENN (Sensoy et al., 2018) | 88.0±0.03 | 78.3±0.65 | 82.4±0.36 | 87.8±0.23 | 75.4±2.38 | 84.7±1.60 |
| | E-CNN (Tong et al., 2021) | 31.6±1.45 | 37.3±1.58 | 65.5±0.82 | 33.3±1.21 | 35.1±0.91 | 64.8±1.12 |
| | RAPS (Angelopoulos et al., 2021) | 85.5±0.35 | 66.5±0.72 | 80.2±0.34 | 85.9±0.42 | 67.6±0.62 | 80.6±0.48 |
| | HENN (ours) | **88.8**±0.17 | **96.6**±0.65 | **85.7**±1.27 | **88.9**±0.14 | **97.5**±0.49 | **85.4**±1.78 |

Table 9: Results (%) of BREEDS-nonliving26 based on two different Gaussian kernel sizes. (The average and 95% confidence interval of three runs are provided.)

| $M$ | Methods | Gaussian kernel size: 3×3 | | | Gaussian kernel size: 5×5 | | |
|---|---|---|---|---|---|---|---|
| | | OverJS | CompJS | Acc | OverJS | CompJS | Acc |
| 10 | DNN (Tan & Le, 2019) | 85.6±0.32 | 62.0±0.35 | 82.9±0.33 | 86.0±0.26 | 64.0±1.60 | 83.0±0.12 |
| | ENN (Sensoy et al., 2018) | 85.0±0.49 | 52.9±2.74 | 84.5±0.43 | 85.0±0.48 | 52.8±3.79 | 84.2±0.76 |
| | E-CNN (Tong et al., 2021) | 28.3±0.68 | 35.8±4.23 | 60.6±0.97 | 29.6±0.74 | 37.1±3.93 | 60.8±0.76 |
| | RAPS (Angelopoulos et al., 2021) | 82.7±0.36 | 46.3±1.01 | 82.9±0.33 | 83.4±0.42 | 49.5±1.43 | 83.0±0.12 |
| | HENN (ours) | **86.9**±0.13 | **96.8**±0.57 | **85.4**±0.35 | **87.0**±0.12 | **96.2**±1.70 | **85.3**±0.39 |
| 15 | DNN (Tan & Le, 2019) | 85.6±0.39 | 68.9±0.30 | 81.5±0.16 | 85.5±0.50 | 67.3±3.41 | 81.4±0.55 |
| | ENN (Sensoy et al., 2018) | 85.4±0.26 | 62.6±1.59 | 82.9±0.21 | 85.3±0.08 | 61.9±1.31 | 83.2±0.35 |
| | E-CNN (Tong et al., 2021) | 29.8±1.22 | 35.1±4.41 | 60.1±0.87 | 28.9±0.73 | 35.1±4.67 | 60.3±0.84 |
| | RAPS (Angelopoulos et al., 2021) | 83.8±0.42 | 56.1±0.28 | 81.5±0.16 | 83.7±0.43 | 55.9±0.59 | 81.4±0.55 |
| | HENN (ours) | **86.9**±0.03 | **96.2**±1.14 | **84.1**±0.30 | **86.9**±0.21 | **95.6**±1.09 | **84.8**±0.40 |
| 20 | DNN (Tan & Le, 2019) | 86.7±0.34 | 74.5±0.32 | 80.3±0.15 | 86.5±0.42 | 76.2±0.41 | 79.8±0.23 |
| | ENN (Sensoy et al., 2018) | 85.9±0.43 | 68.3±2.13 | 81.7±0.35 | 86.0±0.49 | 67.9±2.44 | 82.2±0.73 |
| | E-CNN (Tong et al., 2021) | 29.8±0.92 | 35.1±2.43 | 60.5±0.81 | 28.6±0.75 | 36.8±3.46 | 60.9±0.65 |
| | RAPS (Angelopoulos et al., 2021) | 82.4±0.41 | 57.7±0.32 | 80.3±0.15 | 84.1±0.23 | 57.8±0.45 | 79.8±0.23 |
| | HENN (ours) | **87.4**±0.22 | **94.5**±0.46 | **85.5**±0.41 | **87.5**±0.17 | **94.5**±1.00 | **85.3**±0.45 |

Table 10: Results (%) of Gaussian kernel size: 7×7 on Living17 and Nonliving26. (The average and 95% confidence interval of three runs are provided.)

| $M$ | Methods | Living17 | | | Nonliving26 | | |
|---|---|---|---|---|---|---|---|
| | | OverJS | CompJS | Acc | OverJS | CompJS | Acc |
| 10 | DNN (Tan & Le, 2019) | 88.4±0.24 | 79.0±1.05 | 83.3±0.49 | 85.8±0.34 | 63.8±1.48 | 82.9±0.19 |
| | ENN (Sensoy et al., 2018) | 87.9±0.23 | 71.0±0.99 | 84.4±0.23 | 85.3±0.29 | 54.6±1.42 | 84.1±0.33 |
| | E-CNN (Tong et al., 2021) | 30.4±0.98 | 36.7±1.65 | 65.5±1.87 | 28.2±0.74 | 35.5±1.43 | 59.4±2.91 |
| | RAPS (Angelopoulos et al., 2021) | 85.9±0.32 | 60.8±1.72 | 83.3±0.49 | 83.5±0.37 | 53.6±1.73 | 82.9±0.19 |
| | HENN (ours) | **88.7**±0.36 | **96.0**±0.82 | **85.3**±0.28 | **86.8**±0.07 | **95.9**±3.22 | **85.0**±0.49 |
| 15 | DNN (Tan & Le, 2019) | 88.4±0.35 | 83.2±2.31 | 80.2±0.79 | 85.8±0.11 | 70.6±1.20 | 81.2±0.63 |
| | ENN (Sensoy et al., 2018) | 88.1±0.22 | 78.3±0.20 | 82.7±1.00 | 85.4±0.14 | 62.1±0.76 | 83.0±0.57 |
| | E-CNN (Tong et al., 2021) | 30.5±0.47 | 36.6±1.84 | 65.6±2.99 | 28.1±0.59 | 35.6±1.97 | 60.1±2.86 |
| | RAPS (Angelopoulos et al., 2021) | 85.7±0.38 | 66.9±1.33 | 80.2±0.79 | 83.7±0.46 | 56.0±0.84 | 81.2±0.63 |
| | HENN (ours) | **88.7**±0.29 | **97.1**±0.33 | **84.4**±1.71 | **86.9**±0.21 | **94.8**±1.42 | **85.2**±0.55 |

Table 11: Results (%) of Gaussian kernel size: 5×5 on CIFAR100 and tinyImageNet (based on one run).

| $M$ | Methods | CIFAR100 | | | tinyImageNet | | |
|---|---|---|---|---|---|---|---|
| | | OverJS | CompJS | Acc | OverJS | CompJS | Acc |
| 10 | DNN (Tan & Le, 2019) | **86.5** | 65.3 | 83.8 | 83.9 | 46.0 | 83.2 |
| | ENN (Sensoy et al., 2018) | 83.1 | 58.8 | **84.5** | 54.8 | 43.1 | 56.1 |
| | E-CNN (Tong et al., 2021) | 28.5 | 22.4 | 74.2 | 23.4 | 21.3 | 68.2 |
| | RAPS (Angelopoulos et al., 2021) | 80.5 | 49.8 | 83.8 | 72.5 | 43.7 | 83.2 |
| | HENN (ours) | 85.9 | **88.7** | 83.1 | **86.2** | **85.0** | **83.5** |
| 15 | DNN (Tan & Le, 2019) | **86.2** | 69.9 | 82.4 | 83.2 | 50.4 | 82.3 |
| | ENN (Sensoy et al., 2018) | 82.8 | 58.4 | **84.5** | 52.8 | 47.6 | 55.7 |
| | E-CNN (Tong et al., 2021) | 28.7 | 23.4 | 70.2 | 23.3 | 21.2 | 68.5 |
| | RAPS (Angelopoulos et al., 2021) | 80.2 | 52.6 | 82.5 | 75.0 | 43.5 | 82.3 |
| | HENN (ours) | 86.1 | **85.4** | 84.2 | **86.2** | **83.3** | 82.3 |
| 20 | DNN (Tan & Le, 2019) | 86.2 | 73.1 | 80.6 | 83.2 | 53.8 | 81.7 |
| | ENN (Sensoy et al., 2018) | 82.4 | 65.3 | 82.3 | 57.7 | 21.6 | 59.1 |
| | E-CNN (Tong et al., 2021) | 28.6 | 23.6 | 73.5 | 23.4 | 22.5 | 68.2 |
| | RAPS (Angelopoulos et al., 2021) | 78.8 | 55.2 | 80.6 | 75.4 | 40.2 | 81.7 |
| | HENN (ours) | **86.7** | **82.5** | **83.4** | **85.5** | **81.0** | **83.1** |

Table 12: Results (%) of Gaussian kernel size: 7×7 on CIFAR100 and tinyImageNet (based on one run).

| $M$ | Methods | CIFAR100 | | | tinyImageNet | | |
|---|---|---|---|---|---|---|---|
| | | OverJS | CompJS | Acc | OverJS | CompJS | Acc |
| 10 | DNN (Tan & Le, 2019) | 86.2 | 62.4 | 83.8 | 83.7 | 44.8 | 83.3 |
| | ENN (Sensoy et al., 2018) | 82.4 | 30.5 | 84.8 | 46.2 | 43.4 | 83.3 |
| | E-CNN (Tong et al., 2021) | 28.5 | 22.4 | 74.2 | 23.6 | 21.8 | 68.0 |
| | RAPS (Angelopoulos et al., 2021) | 80.0 | 49.3 | 83.8 | 71.5 | 43.9 | 83.3 |
| | HENN (ours) | **87.0** | **82.7** | **85.8** | **84.3** | **86.9** | **83.8** |
| 15 | DNN (Tan & Le, 2019) | 85.7 | 64.2 | 82.5 | 83.6 | 52.1 | 82.5 |
| | ENN (Sensoy et al., 2018) | 82.5 | 39.6 | 83.6 | 48.0 | 42.3 | 82.4 |
| | E-CNN (Tong et al., 2021) | 28.7 | 23.4 | 70.2 | 23.5 | 21.9 | 68.2 |
| | RAPS (Angelopoulos et al., 2021) | 78.3 | 51.6 | 82.5 | 73.8 | 43.5 | 82.5 |
| | HENN (ours) | **86.4** | **79.9** | **84.3** | **84.1** | **83.0** | **83.5** |
| 20 | DNN (Tan & Le, 2019) | 85.3 | 69.8 | 80.5 | 83.5 | 56.4 | 81.7 |
| | ENN (Sensoy et al., 2018) | 81.5 | 44.6 | **81.8** | 43.3 | 41.2 | 81.7 |
| | E-CNN (Tong et al., 2021) | 28.6 | 23.7 | 73.4 | 23.3 | 21.5 | 68.2 |
| | RAPS (Angelopoulos et al., 2021) | 74.4 | 53.2 | 80.5 | 74.1 | 39.3 | 81.7 |
| | HENN (ours) | **85.5** | **81.0** | 80.7 | **83.9** | **81.2** | **83.1** |

## F.3 Seperation of Singleton and Composite Examples

Fig. 4 and 5 show comprehensive ROC curves for CIFAR100 and tinyImageNet based on different $M$s and different Gaussian kernel sizes, which indicates that *vagueness* is the best indicator compared to other different uncertainty measurements.

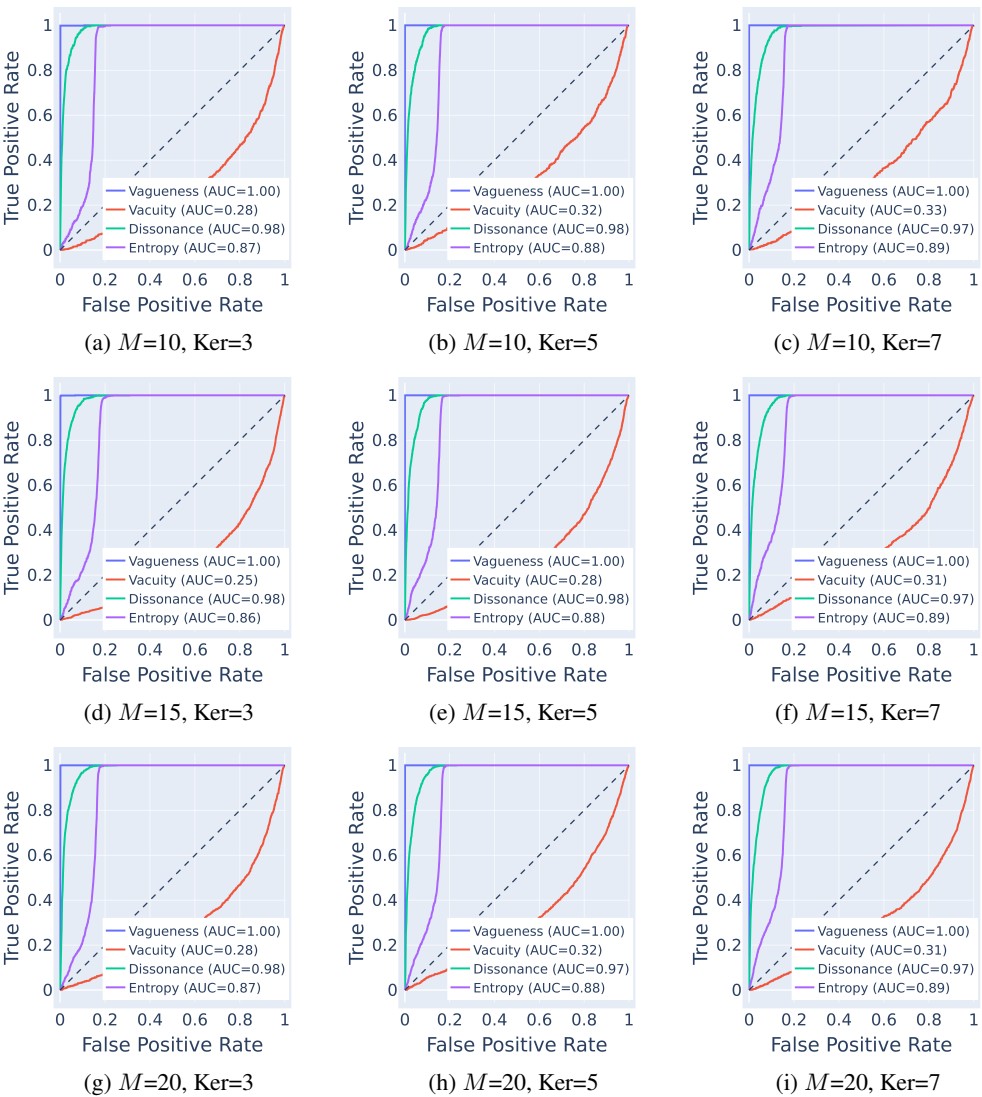

Figure 4: ROC curves of separating composite examples and singleton examples among different measurements: *vagueness* of HENN, *vacuity* of ENN, *dissonance* of ENN, and *entropy* of DNN on CIFAR100 for different numbers of selected composite classes and kernel sizes ("Ker" represents "kernel size").

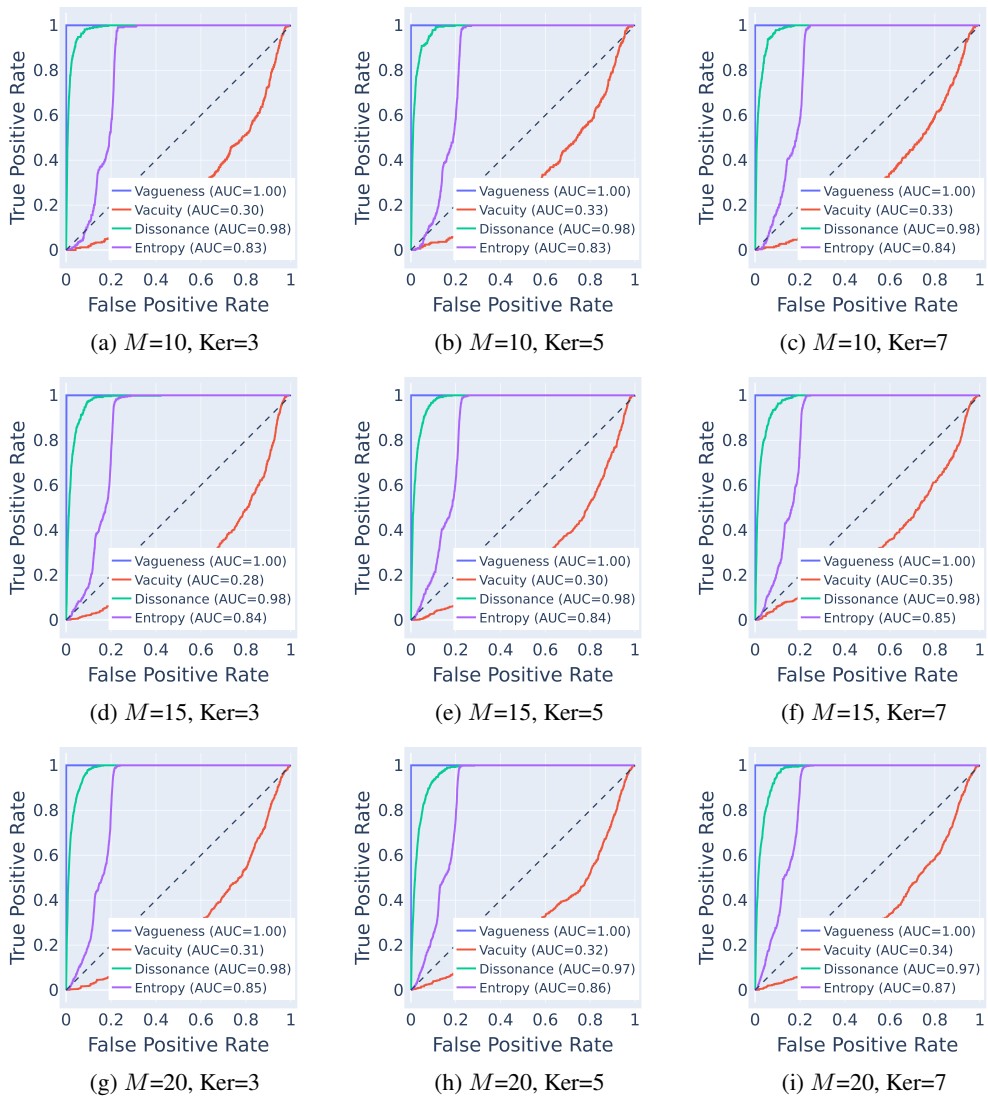

Figure 5: ROC curves of separating composite examples and singleton examples among different measurements: *vagueness* of HENN, *vacurity* of ENN, *dissonance* of ENN, and *entropy* of DNN on tinyImageNet for different numbers of selected composite classes and kernel sizes. ("Ker" represents "kernel size")

## F.4 ABLATION STUDY ON REGULARIZER

To explore the effect of Regularizer for singleton evidence, we try different tradeoff coefficients $\lambda$ for KL regularization in our HENN method. Experiments are conducted on CIFAR100 with a pre-trained EfficientNet-b3 model and a fixed learning rate 1e-5. The results are represented in Table 13. Without this regularization term, the CompJS is 0, which means that the model does not predict composite prediction, and the Acc is 78.5%. The reason is minimizing UPCE loss solely cannot provide sufficient composite evidence output for those composite sets. In this way, the composite examples will have relatively low accuracy compared to singleton ones. If the coefficient $\lambda$ increases, demonstrating a larger preference on flat GDDs instead of UPCE minimizer, the evidence for singleton classes will reduce to be flat as we proved in Proposition 2. Therefore, the CompJS will no longer be zero since the model tend to replace confusing singleton evidence for composite examples. $\lambda = 0.1$ gives us the best performance on both the composite prediction metrics (OverJS, CompJS) as well as singleton prediction accuracy (Acc). This means that HENN can predict composite class labels but also has good singleton class label prediction. This ablation study verifies the importance of the regularization term and shows a fine-tuned tradeoff hyperparameter can provide reliable composite and singleton prediction simultaneously. In addition, the OverallJS remains high across different choices of $\lambda$s, demonstrating the robustness of our method.

Table 13: Effect of Regularizer: Different trade-off coefficient $\lambda$ on CIFAR100 with pretrained EfficientNet-b3 model and 1e-5 learning rate.

| $\lambda$ | OverJS | CompJS | Acc |
|---|---|---|---|
| 0 | 76.4 | 0.0 | 78.5 |
| 0.01 | 83.6 | 76.3 | 85.1 |
| 0.1 | **87.9** | **87.7** | **85.3** |
| 1.0 | 81.8 | 72.0 | 79.7 |

## F.5 ADDITIONAL RESULTS

Table 14: Results (%) of NAbirds based on the pre-trained EfficientNet-b3 backbone. (The average and 95% confidence interval of three runs are provided based on three runs.)

| Methods | OverJS | CompJS | Acc |
|---|---|---|---|
| DNN (Tan & Le, 2019) | 77.38±0.19 | 35.24±3.52 | 78.04±0.27 |
| ENN (Sensoy et al., 2018) | 76.72±0.56 | 37.46±2.39 | 78.45±0.31 |
| HENN (ours) | **80.01**±0.37 | **71.42**±1.43 | **80.14**±0.35 |

### F.5.1 EXPERIMENTS ON FINE-GRAINED DATASET: NABIRDS

We also conduct experiments on one fine-grained dataset: NAbirds (Van Horn et al., 2015). It has 555 different categories of birds and each category has around 50 images for both training and test set. According to the provided class hierarchy information, these 555 subclasses can be divided into 404 groups (superclasses). After filtering out superclasses which has only a single subclass, the same procedure as previous four datasets (TinyImageNet, Living17, Nonliving26, and CIFAR100) is applied to randomly select 10 composite class labels. Tab. 14 shows results based on the fine-grained dataset NAbirds (Van Horn et al., 2015). Consistent with previous experiments on four datasets, HENN outperforms DNN and ENN for a large margin in terms of CompJS. And HENN also performs better in terms of OverJS and Acc.

### F.5.2 REAL-WORLD DATASET WITH COMPOSITE CLASS LABELS

We admit that the datasets with Gaussian blurring are semi-synthetic. From a sizable pool of applicants, we selected 23 students from our department and tasked them with annotating images in the CIFAR10 dataset and one subset of tinyImageNet (renamed as tinyImageNet-20), categorizing each as either a singleton class or a composite set. This effort successfully resulted in a real-world dataset enriched with human-annotated singleton and composite labels. Tab. 15 shows results based

Table 15: Results (%) on CIFAR10 based on two backbones. (The average and 95% confidence interval of three runs are provided based on five runs.).

| | ResNet18 | | | EfficientNet-b3 | | |
|---|---|---|---|---|---|---|
| Methods | OverJS | CompJS | Acc | OverJS | CompJS | Acc |
| DNN (Tan & Le, 2019) | 79.73±0.33 | 40.10±7.06 | 82.17±0.54 | 92.53±0.11 | 53.59±3.15 | 96.49±0.21 |
| ENN (Sensoy et al., 2018) | 67.09±0.75 | 46.80±0.06 | 82.75±0.19 | 77.84±3.86 | 54.83±0.59 | 96.82±0.38 |
| E-CNN (Tong et al., 2021) | 59.68±0.62 | 31.84±0.81 | 66.23±1.47 | 63.65±0.93 | 34.74±2.91 | 68.98±0.72 |
| RAPS (Angelopoulos et al., 2021) | 62.60±0.46 | 33.80±4.86 | 82.17±0.54 | 65.70±0.80 | 39.40±2.29 | 96.49±0.21 |
| HENN (ours) | **80.74**±0.17 | **51.44**±1.02 | **83.03**±0.14 | **93.38**±0.06 | **72.87**±1.25 | **97.52**±0.04 |

Table 16: Results (%) on tinyImageNet-20 based on the ResNet18 backbone. (The average and 95% confidence interval of three runs are provided based on five runs.).

| | ResNet18 | | |
|---|---|---|---|
| Methods | OverJS | CompJS | Acc |
| DNN (Tan & Le, 2019) | 40.03±0.29 | 24.70±1.85 | 42.20±1.24 |
| ENN (Sensoy et al., 2018) | 36.44±1.65 | 22.78±1.48 | 42.45±1.15 |
| HENN (ours) | **42.43**±0.78 | **25.32**±1.87 | **43.93**±1.23 |

(a) CIFAR10 (ResNet18)  (b) CIFAR10 (EfficientNet-b3)  (c) tinyImageNet-20 (ResNet18)

Figure 6: AUC curves of different uncertainty types: Vagueness, Vacuity, Dissonance, and Entropy for two datasets. (a) CIFAR10 based on ResNet18 training from scratch; (b) CIFAR10 fine-tuned on pre-trained EfficientNet-b3; (c) tinyImageNet-20 based on ResNet18 training from scratch.

on real-world dataset CIFAR10 Krizhevsky & Hinton (2009) based on two backbones. HENN outperforms DNN and ENN for a large margin. Fig. 6a and 6b show AUROC curves and scores for different metrics: vagueness, dissonance, vacuity, and entropy. It demonstrates that vagueness is a good indicator to identify whether the image is singleton-labeled or composite-labeled, indicating HENN's advantage.

### F.5.3 ANOTHER DATA CORRUPTION

Table 17: Results (%) of BREEDS-Living-17 based on the pre-trained EfficientNet-b3 backbone. (The average and 95% confidence interval of three runs are provided based on three runs).

| Methods | OverJS | CompJS | Acc |
|---|---|---|---|
| DNN (Tan & Le, 2019) | 87.28±0.23 | 74.61±2.57 | 84.35±0.36 |
| ENN (Sensoy et al., 2018) | 87.46±0.34 | 69.44±3.25 | 85.38±0.28 |
| RAPS (Angelopoulos et al., 2021) | 85.38±0.32 | 62.10±0.26 | 84.35±0.36 |
| HENN (ours) | **88.09**±0.21 | **96.33**±3.54 | **86.12**±0.37 |

Besides the Gaussian blurring we used, the Bicubic transformation was also examined as detailed in Tab.17. The findings from this analysis align consistently with the result of the experiment based on Gaussian blurring.

### F.5.4 CASE STUDY ON HENN TRAINED WITH EXCLUSIVE SINGLETON CLASS DATA

Tab.18 presents the accuracy results from the ENN and HENN methods trained and evaluated on CIFAR100 with only singleton class data (without Gaussian blurring and label replacement) across 5

Table 18: Case Study: HENN Trained with Exclusive Singleton Class Data.

| Methods | ENN | HENN |
|---------|-----|------|
| Acc(%) | $85.82 \pm 1.0$ | $85.81 \pm 2.4$ |

trials each. The mean accuracy and the standard deviation are reported. Under a traditional singleton classification setting, HENN still shows comparable performance in terms of accuracy compared to ENN model. A notable advantage of HENN is its ability to quantify an additional type of uncertainty compared to ENN with minimal performance degradation observed even if the training data consists of exclusive singleton ones.

### F.5.5 CASE STUDY ON EVIDENCE OUTPUT

In this section, we show the effect of the regularization coefficient $\lambda$ by demonstrating its impact on the output evidence throughout a case study. To verify our intuitions for Proposition 2, we set experiments to inspect the non-zero ratio of both singleton evidence and composite evidence among all testing examples. A small positive threshold value of $\gamma = 10^{-4}$ is introduced to determine whether the mean singleton or composite evidence is non-zero while adapting to the computation precision in practice. In other words, for each testing data point, we calculate the predicted evidence $f(\mathbf{x}; \boldsymbol{\theta}) = (\boldsymbol{\alpha}, \mathbf{c})$, and based on the given hyper-domain, it is feasible to get mean evidence in singleton domain $\bar{\boldsymbol{\alpha}} = \frac{1}{|\mathbb{Y}|} \sum_{k=1}^{|\mathbb{Y}|} \alpha_k$, and the composite domain $\bar{\mathbf{c}} = \frac{1}{|\mathscr{C}(\mathbb{Y})|} \sum_{j=1}^{|\mathscr{C}(\mathbb{Y})|} c_j$. Define the indicator function of non-zero singleton prediction as

$$g_{\text{sngl}}(\bar{\boldsymbol{\alpha}}) = \begin{cases} 1, & \text{if } \bar{\boldsymbol{\alpha}} \geq \gamma, \\ 0, & \text{otherwise,} \end{cases} \qquad g_{\text{comp}}(\bar{\mathbf{c}}) = \begin{cases} 1, & \text{if } \bar{\mathbf{c}} \geq \gamma, \\ 0, & \text{otherwise,} \end{cases} \tag{55}$$

and the non-zero ratios are $\text{nz}_{\text{sngl}} = \frac{1}{N_{\text{test}}} \sum_{i=1}^{N_{\text{test}}} g_{\text{sngl}}(\bar{\boldsymbol{\alpha}})^{(i)}, \text{nz}_{\text{comp}} = \frac{1}{N_{\text{test}}} \sum_{i=1}^{N_{\text{test}}} g_{\text{comp}}(\bar{\mathbf{c}})^{(i)}$ by taking the mean of all testing samples. The case study is carried out on CIFAR100 with EfficientNet-b3 backbone with the same setting as in previous sections. By controlling regularization coefficient $\lambda$ at different levels of value, the predicted evidence from HENN is listed in Table 19, indicating that larger regularization can adjust the evidence distribution to be more balanced between singleton and composite parts. Oppositely, a lower regularization coefficient or without any regularization can result in concentrating predictive evidence only on the singleton part.

Table 19: Case Study: Effectiveness of regularization term on evidence distribution.

| $\lambda$ | $\text{nz}_{\text{sngl}}$ | $\text{nz}_{\text{comp}}$ |
|-----------|---------------------------|---------------------------|
| 0.01 | 71.52% | 71.72% |
| $10^{-4}$ | 100.0% | 0.04% |
| $10^{-8}$ | 100.0% | 0.2% |
| 0 | 100.0% | 0.3% |

**Empirical verification of Propositions 1 and Eq.14.** Our case study on CIFAR100 in App. F.5.5 demonstrates observations consistent with our propositions: (1) HENN trained based on UPCE and a training set consisting of only singleton class labels predicts non-zero evidence on composite class labels for $14.4\%$ of the training samples even that the training set does not have evidence of composite class labels to accumulate, and (2) The HENN trained based on UPCE and a training set consisting of only composite class labels predicts non-zero evidence on singleton class labels for $100.0\%$ of the training samples even that the training set does not have evidence of singleton class labels to accumulate. Our proposed regularization can avoid these unexpected behaviors. The UAP for neural networks has been studied (Leshno et al., 1993b) and recently used in the theoretical analyses of ENN-related paper for graph data (Alan Hart et al., 2023).

