# OpenReview forum: "Hyper Evidential Deep Learning to Quantify Composite Classification Uncertainty"
_ICLR.cc/2024/Conference — ICLR 2024 poster_

### Official Review · Reviewer_3vp1 · 2023-10-29

**Soundness:** 3 good
**Presentation:** 3 good
**Contribution:** 3 good
**Rating:** 6
**Confidence:** 3

**Summary:**

The DNN’s uncertainty due to composite set labels (i.e., an example might be labeled as a set of possible classes, but only one class is true) in training data is considered. This work introduces a new type of uncertainty termed vagueness, and propose a framework Hyper-Evidential Neural Network (HENN) to quantify this type of uncertainty. Further, a loss named uncertainty partial cross entropy (UPCE) is proposed. Finally, the proposed method is shown to be effective on four image datasets.

**Strengths:**

1.	A novel type of DNN uncertainty is introduced, and the uncertainty calibration is evaluated with the metric Jaccard Similarity.
2.	An extension of the partial cross, uncertainty partial cross entropy (UPCE) is proposed.

**Weaknesses:**

1.	Although the newly introduced uncertainty concept and problem setup sound interesting, I am not sure the empirical evaluation is realistic and convincing enough. Are all the composite labels used in the empirical evaluations synthetically generated? Could you provide more concrete examples in these synthetic datasets? I am curious how realistic these generated composite labels are. If so, is it possible to run evaluations on real-world composite set labels?

**Questions:**

Please see the comments in the weaknesses.

---

> ### Author Response · Authors · 2023-11-23
> **Response to Reviewer 3vp1**
>
> Thank you for your constructive feedback and for recognizing the novel aspects of our work. We appreciate the opportunity to address your concerns regarding the empirical evaluation of our framework.
>
> > Are all the composite labels used in the empirical evaluations synthetically generated? Could you provide more concrete examples in these synthetic datasets? I am curious how realistic these generated composite labels are.  is it possible to run evaluations on real-world composite set labels?
>
> The composite labels used in our empirical evaluations were indeed synthetically generated. One concrete example could be found in Fig.1(right). A blurring image which is difficult to distinguish between {husk} and {wolf}. We admit that the datasets with Gaussian blurring are semi-synthetic. From a sizable pool of applicants, we selected 23 students from our department and tasked them with annotating images in the CIFAR10 dataset, categorizing each as either a singleton class or a composite set. This effort successfully resulted in a real-world dataset enriched with human-annotated singleton and composite labels. Our method, along with various baseline approaches, was applied to this dataset. The comparative results were in line with those obtained from the synthetic datasets. Additionally, we plan to publicly release the dataset and the labels.  The following table results are also represented in the **Table 15 in the rebuttal revision**. We also show AUROC curves in **Figure 7 in the revision**, which indicates that vagueness is still the best among different uncertainties to distinguish composite examples from singleton examples.
>
> Backbone: ResNet18
>
> | Method | OverJS | CompJS | Acc
> | ---------| ---------| ---------| ---------|
> |DNN|79.73±0.33|40.10±7.06|82.17±0.54|
> |ENN|67.09±0.75|46.80±0.06|82.75±0.19|
> |E-CNN|59.68±0.62|31.84±0.81|66.23±1.47|
> |RAPS|62.60±0.46|33.80±4.86|82.17±0.54|
> |HENN (ours)|**80.74**±0.17| **51.44**±1.02| **83.03**±0.14|
>
>
> Backbone: EfficientNet-b3
> | Method | OverJS | CompJS | Acc
> | ---------| ---------| ---------| ---------|
> |DNN|92.53±0.11| 53.59±3.15| 96.49±0.21|
> |ENN|77.84±3.86| 54.83±0.59| 96.82±0.38|
> |E-CNN|63.65±0.93| 34.74±2.91| 68.98±0.72|
> |RAPS|65.70±0.80 | 39.40±2.29 | 96.49±0.21|
> |HENN (ours)|**93.38**±0.06| **72.87**±1.25| **97.52**±0.04|
>
>
> ### Please kindly let us know if you have any concerns you find not fully addressed. We are more than happy to have a further discussion regarding it. Thank you so much for your time!

---

### Official Review · Reviewer_AQu8 · 2023-10-30

**Soundness:** 3 good
**Presentation:** 3 good
**Contribution:** 2 fair
**Rating:** 6
**Confidence:** 3

**Summary:**

The paper proposes a novel framework for deep learning to deal with composite labels, i.e. labels that couldn't be assigned to a single class due to quality of the input, for example, and were assigned to more than one class. The framework assumes a neural network to output, similarly to the training labels, both singleton classes and composite labels. The paper also proposes a novel uncertainty metric, called vagueness, that is measuring uncertainty related to composite label evidence in training. To train a novel neural network with composite output, the paper proposes a novel loss function.


============

Update after rebuttal: I have read the authors’ response and other reviews. I really appreciate the effort, information and novel results provided by the authors to all reviews. I am happy to see even my suggested small experiment on using singleton labels only implemented.


However, a tiny negative note. I appreciate the authors’ answer and motivation about having composite labels during training, though if possible I would suggest using a slightly more modern reference. Also, the authors haven’t address my concern about why we need for the neural network to output composite labels. Though, it is tiny in comparison to strengths of the paper and none of the other reviewers saw that as a concern, so I am increasing my score.

**Strengths:**

* A very thorough work has been done with the new framework (and basically the new problem setup). Extensive theoretical evaluation, empirical evaluation including ablation study, a good number of baselines considered for comparison.
* In addition to the new framework itself, the novel uncertainty metric is made as a proper contribution on its own with the thorough theoretical and empirical comparison with other uncertainty metrics.
* The paper is mostly well-written and presents the context of prior work. The only drawbacks I can see are obviously due to the lack of space.

Originality: I am not familiar well with the related works, but both the framework and the uncertainty metric appears to be novel.

Quality: Very well done and thought through piece of work. See above.

Clarity: Mostly well-written and easy to follow.

Significance: Empirical evaluation demonstrates that even in terms of single class classification (i.e. output of a NN is a single class) the proposed framework demonstrates improved accuracy in comparison to all considered baselines. Moreover, the proposed uncertainty metric demonstrates significantly superior performance in terms of distinguishing between composite and single labels examples.

**Weaknesses:**

The main weakness that I see in the paper is the lack of motivation how severe is the problem of having composite labels in practice for training and why we need a NN to output us a composite label in practice. The main conceptual motivation that I gathered from the paper is that the new framework allows to estimate uncertainty related to composite labels in training data. However, the argument is not too convincing. It seems that we don't have to produce a composite label in order to estimate the effect of composite labels in the training data (not that I know how to do it without).
Empirical evidence shows us the practical motivation of using the novel framework: it gives us the higher accuracy when we consider a single class prediction. However, again this is shown in the experiments with the composite labels in training, which is not well motivated how often in practice we would have these labels for training.
To this end, I think a small experiment, showing that the proposed framework still works with traditional setup of single labels for training only, would be beneficial.

Originality: I appreciate that the considered problem formulation is different in that a single class is expected to be true for each sample, but maybe lacking due to the quality of the sample, but still I find it is a related area. The area of research devoted to multiclass classification is missing in the related works (e.g., Augustin, A.; Venanzi, M.; Hare, J.; Rogers, A.; and Jennings, N. 2017. Bayesian aggregation of categorical distributions with applications in crowdsourcing. AAAI Press/International Joint Conferences on Artificial Intelligence, but there are lots of others).

Quality: The code is provided which should elevate this weakness, but based on the text not all implementation details are provided sufficiently to reproduce the results. See details below.
Different types of image corruption would be interesting to see in the experiments in addition to the Gaussian blur.

Clarity: A lot of theory is packed in a very small space with lack of illustrative examples. See details below.

Significance: As per the main drawback mentioned above, I am not sure about the significance of the paper due to very specific problem setup, which was not too convincing for me.

Specific comments/suggestions:
1. First two sentence of the text are unclear how to connect to the rest of the text. The problem considered is not related to missing data but rather ambiguous data due to the quality of the input.
2. Missing reference in third paragraph of Section 1.
3. Section 3. Illustrative examples of what this means in practice would be much appreciated. Note that example from Table 1 is not sufficient. For example, what does evidence 24 mean in practice? 24 annotators voted for this category?
4. Figure 1 right is not referred to in the text.
5. UCE (page 6) is not defined.
6. Propositions 1 and 2. How reasonable is the assumption of universal approximation property?
7. Eq. 15 appears without any introduction, connection to the previous text.
8. Preprocessing of the dataset to create composite labels is unclear (including the text in the appendix).
* "Several random subclasses for each selected superclass will be chosen" - for what? How these subclasses are used further?
* Which images are selected to be blurred?
9. How CompJS is computed for baselines not producing composite output is unclear until explanation of what cutoff is in Appendix. Also mentioning of cutoffs in the main text is unclear until this explanation in Appendix (no reference in the main text).
10. It is unclear exactly how superclasses are extracted for Living17 and Nonliving26 datasets.
11. The full list of data augmentations is required for reproducibility.
12. Up until the last paragraph in page 26 in Appendix, it is not clear how the process of duplicating samples with composite labels is working (there is no reference to this paragraph in the earlier text).
13. Section F.4 There are SinglJS, SingleAcc and Acc mentioned in different places. What is the correct one?


Minor:
1. Page 7. The paragraph before Section 5. "As a generalized framework of ENN, The HENN" -> "the"
2. Section F.4 title. "regularier" -> "regulazier"

**Questions:**

My main question to the authors, the answer to which hopefully will clarify any doubts from my side, is the question of motivation of the problem setup. I.e. how important is the problem of having composite labels during training and why do we need to also output composite labels by a NN?

---

> ### Author Response · Authors · 2023-11-23
> **Response to Reviewer AQu8 [1/4]**
>
> We appreciate your thorough review of our paper and the opportunity to address your concerns, particularly regarding the motivation and practical implications of our work.
>
> > The question of motivation of the problem setup. I.e. how important is the problem of having composite labels during training and why do we need to also output composite labels by a NN?
>
> We added **two more examples in the first paragraph of the Introduction section** to motivate the problem setup.
>
> “In various applications, particularly those dependent on data from low-quality sensors or high-quality data with insufficiently distinct features to separate some individual classes, the resulting data often exhibits significant vagueness and ambiguity (Allison, 2001; Ng et al., 2011). For example, in security surveillance, grainy images from store cameras may not provide clear enough resolution to accurately distinguish between different individuals or activities, necessitating the use of composite class labels to address this uncertainty (Allison, 2001). Similarly, in the field of medical imaging, a radiograph displaying features suggestive of multiple possible diagnoses may require composite labels to capture this uncertainty (Allison, 2001) effectively.”
>
> > The area of research devoted to multiclass classification is missing in the related works (e.g., Augustin, A.; Venanzi, M.; Hare, J.; Rogers, A.; and Jennings, N. 2017. Bayesian aggregation of categorical distributions with applications in crowdsourcing. AAAI Press/International Joint Conferences on Artificial Intelligence, but there are lots of others).
>
> The AAAI paper by Augustin et al. considers the problem of aggregating judgments (labels) of proportions from crowdsourcing annotators that are skewed and may provide judgments randomly (i.e., they are spammers). This is different from our work. We are concerned with images collected in challenging environments where weather, poor lighting, smoke, fires, poor focusing, etc., lead to less-than-ideal images. Clean images lead to precise annotations, and degraded images lead to vague annotations.  We expect the classifier to make the best classification effort and provide a composite label when the image is degraded to the point that a human cannot discern a singleton. In short, we are concerned with composite and precise labels for possibly poor-quality images. The classifier needs to make a best-effort prediction for the degraded images as opposed to a bad singleton prediction.
>
> > the experiments with the composite labels in training, which is not well motivated how often in practice we would have these labels for training. To this end, I think a small experiment, showing that the proposed framework still works with traditional setup of single labels for training only, would be beneficial.
>
> We added extra experiments on singleton class only datasets and evaluated the performance on traditional classification tasks with Accuracy for ENN and our HENN. Please refer to the result shown **in Appendix F6.4. Tab.18** presents the accuracy results from the ENN and HENN methods trained and evaluated on CIFAR100 with only singleton class data (without Gaussian blurring and label replacement) across 5 trials each. The mean accuracy and the standard deviation are reported. Under a traditional singleton classification setting, HENN still shows comparable performance in terms of accuracy compared to ENN model. A notable advantage of HENN is its ability to quantify an additional type of uncertainty compared to ENN with minimal performance degradation observed even if the training data consists of exclusive singleton ones.
>
> |Method | ENN | HENN |
> |----------|------|--------|
> |Acc(%)|85.82 ± 1.0| 85.81 ± 2.4|

---

> > ### Author Response · Authors · 2023-11-23
> > **Response to Reviewer AQu8 [2/4]**
> >
> > > Specific Q1: First two sentences of the text are unclear on how to connect to the rest of the text. The problem considered is not related to missing data but rather ambiguous data due to the quality of the input.
> >
> > Thanks for pointing out this. We have changed the first sentences in the Introduction section, which is the same to the first question.
> >
> > > Specific Q2: Missing reference in third paragraph of Section 1.
> >
> > References are added for different uncertainties in the corresponding paragraph:
> >
> > “model uncertainty (mutual information between model parameters and the predicted class probabilities (Depeweg et al., 2018; Malinin & Gales, 2018)), data uncertainty (entropy of the predicted class probabilities (Gal, 2016)), confidence (the largest predicted class probability (Hendrycks & Gimpel, 2017)), vacuity (uncertainty due to lack of evidence (Jøsang, 2016; Shi et al., 2020)), and dissonance (due to conflicting evidence (Zhao et al., 2020)).”
> >
> > > Specific Q3: Section 3. Illustrative examples of what this means in practice would be much appreciated. Note that example from Table 1 is not sufficient. For example, what does evidence 24 mean in practice? 24 annotators voted for this category?
> >
> > In contrast to Bayesian modeling terms, we define ``evidence'' as a measure of the accumulated support from training samples, indicating that the input sample should be categorized into a particular singleton class or composite set. The accumulated support can be interpreted as the weighted aggregated number of training samples that support this class or composite set. Unlike a simple count of samples, evidence is typically weighted. This means that not all samples contribute equally to the evidence. For instance, some samples might be more informative or reliable than others, and the network learns to weigh their contribution to the evidence accordingly. Based on our explanation of the evidence above, 24 means the weighted aggregated number of training samples that support this composite set {2, 3} is 24.
> >
> > The concept of 'evidence' is more intuitively explained in a specific scenario outlined in the theoretical analyses by Bengs et al. (2022) and Shi et al. (2020). In this context, we presume the hyper-Dirichlet distribution is constant across all data points. With this assumption, 'evidence' for a particular singleton class or composite set is quantified by the number of training samples in the training set labeled with the respective singleton or composite set. For example, in medical diagnostics, the presence of 24 pieces of composite evidence could suggest that there are approximately 24 similar cases resulting in diseases {2, 3} based on the current observation. This implies that the cases are identified as having either disease 2 or 3, but without specific information to distinguish between them. Conversely, 3 instances of class 1 evidence indicate that 3 similar cases have been identified as disease 1. In such scenarios, doctors might not have a clear preference between diseases 2 and 3, while maintaining a conflicting opinion between disease 1 and {2, 3} for this observation.
> >
> > **Reference**:
> > - Viktor Bengs, Eyke H¨ullermeier, and Willem Waegeman. Pitfalls of epistemic uncertainty quantification
> > through loss minimisation. In Alice H. Oh, Alekh Agarwal, Danielle Belgrave, and
> > Kyunghyun Cho (eds.), Advances in Neural Information Processing Systems, 2022
> > - Weishi Shi, Xujiang Zhao, Feng Chen, and Qi Yu. Multifaceted uncertainty estimation for labelefficient
> > deep learning. Advances in neural information processing systems, 33:17247–17257,
> > 2020.
> >
> > > Specific Q4: Figure 1 right is not referred to in the text.
> >
> > We added one sentence: “Figure 1 shows examples of high uncertainties in their different types and their corresponding probability density plots for 3-class classification.”
> >
> > > Specific Q5: UCE (page 6) is not defined.
> >
> > We added the definition of UCE on Page 6:
> >
> > We note that if $\mathbf{\tilde{y}}$ is a singleton class label, and we replace GDD with the Dirichlet distribution, then the UPCE loss becomes equivalent to the default UCE loss used in learning ENNs: $\texttt{UCE}({\bf x}, \mathbf{\tilde{y}}) = \mathbb{E}_{{\bf p}\sim \mathtt{Dir}({\bf p}|\bf{\alpha})} [\texttt{CE}({\bf p}, \mathbf{\tilde{y}})]$.

---

> > > ### Author Response · Authors · 2023-11-23
> > > **Response to Reviewer AQu8 [3/4]**
> > >
> > > > Specific Q6: Propositions 1 and 2. How reasonable is the assumption of universal approximation property?
> > >
> > > In essence, the proposed HENN is the GDD extension of evidential deep learning (Ulmer et al., 2023) that is based upon Dirichlet distributions. The propositions demonstrate the need for the KL regularization term in the cost function so that only evidence for the corresponding ground truth class can grow large. While the assumption of UAP in the above propositions may not hold in practice, the analysis does demonstrate how UPCE requires the KL regularization term to moderate the evidence.
> > >
> > > An ablation study is discussed in Section 5.2 to empirically demonstrate the need for the regularization term. Our case study on CIFAR100 in Appendix F.7 demonstrates observations consistent with our propositions: (1) The HENN learned based on UPCE and a training set consisting of only singleton class labels predicts non-zero evidence on composite class labels for $14.4\\%$ of the training samples even that the training set does not have evidence of composite class labels to accumulate, and (2) The HENN learned based on UPCE and a training set consisting of only composite class labels predicts non-zero evidence on singleton class labels for $100.0\\%$ of the training samples even that the training set does not have evidence of singleton class labels to accumulate. Our proposed regularization can avoid these unexpected behaviors.
> > > The UAP for neural networks has been studied in the literature (Leshno et al., 1993b) and used in the theoretical analyses of neural networks (e.g., the recent ENN-related paper for graph data (Alan Hart et al., 2023)).
> > >
> > > **References**:
> > > - Moshe Leshno, Vladimir Ya Lin, Allan Pinkus, and Shimon Schocken. Multilayer feedforward networks with a nonpolynomial activation function can approximate any function. Neural networks, 6(6):861–867, 1993b.
> > > - Russell Alan Hart, Linlin Yu, Yifei Lou, and Feng Chen. Improvements on uncertainty quantification for node classification via distance-based regularization. In Advances in Neural Information Processing Systems (2023), 2023.
> > >
> > > > Specific Q7: Eq. 15 appears without any introduction, connection to the previous text.
> > >
> > > We added a short sentence on Page 7 to indicate that this equation is the regularized UPCE loss function.
> > >
> > > > Specific Q8: Preprocessing of the dataset to create composite labels is unclear (including the text in the appendix).
> > >
> > > "Each dataset has a class hierarchy that divides the original class categories (subclasses) into fewer superclasses. To generate composite labels, we begin by selecting a predetermined number of superclasses, denoted as M. For each chosen superclass, we randomly pick a random number of subclasses. Subsequently, a subset of images belonging to these selected subclasses undergoes Gaussian Blurring (Richard Webster et al., 2018) to create vague images. The resulting blurred images are then assigned new labels that represent a composite of the subclasses, rather than a single subclass. For instance, if two images, classified as subclasses A and B under the same superclass, are selected and blurred, they become indistinct. Consequently, these images are relabeled as the composite set {A, B} to construct the dataset."
> > >
> > > We have updated this part **in section 5.1 and detail in Appendix E**.
> > >
> > > > Specific Q9: How CompJS is computed for baselines not producing composite output is unclear until explanation of what cutoff is in Appendix. Also mentioning of cutoffs in the main text is unclear until this explanation in Appendix (no reference in the main text).
> > >
> > > We add one sentence to Baselines in section 5.1 to introduce the threshold (cutoff) to explain how to produce composite output for DNN and ENN.
> > > “In practice, it is necessary to set a cutoff value of predicted conditional class probabilities to generate set predictions for DNNs and ENNs. (See App. E.3).”
> > >  And we add **the reference to Appendix** for further detail.
> > > We have updated this part in **section 5.1 (Baselines) and detail in Appendix E.3**.

---

> ### Author Response · Authors · 2023-11-23
> **Response to Reviewer AQu8 [4/4]**
>
> > Specific Q10: It is unclear exactly how superclasses are extracted for Living17 and Nonliving26 datasets.
>
> The extraction of superclass and class hierarchy for these two datsaets are explored by Santurkar et al. (2021). We **add one sentence in the Datasets&Preprocessing (Section 5.1)**:
>
> “Their superclasses and hierarchy information have been extracted by Santurkar et al. (2021) based on visual similarities.”
>
> We also introduce **more detail in Appendix E.1** to explain why the Living17 and Nonliving26 are considered:
>
> "Additionally, distinguishing between different classes can often be challenging due to their similar visual features. Unlike WordNet, which organizes its hierarchy based on semantic relationships between words, the class hierarchies of Living17 and Nonliving26 are constructed considering both visual and semantic similarities. These two subsets, part of the broader ImageNet dataset (Deng et al., 2009), consist of images with a resolution of 224×224 pixels. For detailed information about these subsets, please refer to Tables 5 and 6 in the original paper."
>
> > Specific Q11: The full list of data augmentations is required for reproducibility.
>
> Data augmentation has been included **in Appendix E.2** actually. To make this more clear, we add more sentences to explain what data augmentation we used at the end of Appendix E.2:
>
> “We used 2 methods for data augmentation following a typical computer vision setting. First, each image is applied to a random horizontal flip with the flipping probability of 0.5. After that, a random corp is introduced for each image with a size of 32×32 and padding of 4.”
>
> > Specific Q12: Up until the last paragraph in page 26 in Appendix, it is not clear how the process of duplicating samples with composite labels is working (there is no reference to this paragraph in the earlier text).
>
> We add detailed process to explain this in **Appendix E.3 Implementation**:
>
> DNN and ENN are typically designed to process examples with singleton labels and cannot deal with composite class labels during training.  Given that our training dataset includes vague images with composite class labels, we have devised a strategy to enable these baseline models to handle such examples without eliminating training data. This involves duplicating composite examples and assigning them singleton labels derived from their composite set labels, ensuring that all class labels during training remain exclusive. For instance, if there is an image x with a composite label A, B during training, we create two duplicates of x – one labeled as singleton A and the other as singleton B – and use these as inputs for model training.
>
> > Specific Q13: Section F.4 There are SinglJS, SingleAcc and Acc mentioned in different places. What is the correct one?
>
> We united these terms to Acc in F.4. Basically it means the top-1 accuracy for singleton label prediction.
>
>
> > Quality: The code is provided which should elevate this weakness, but based on the text not all implementation details are provided sufficiently to reproduce the results. See details below. Different types of image corruption would be interesting to see in the experiments, in addition to the Gaussian blur.
>
> We have added more detail which is necessary to reproduce the results according to the above questions. In addition, we also wrote a README file in our code repo to explain how to use our code.
>
> For different type of image corruption in addition to Gaussian blur, use another data corruption method: bicubic interpolation, which is popular in the super-resolution research community. For example, in paper [1], the low resolution (LR) input is generated based on the high resolution images. Also, the same method is also used in paper [2]: “The LR counterparts are downsampled using bicubic interpolation”. The first column is LR image after Bicubic interpolation in  the following figure (figure 6 from paper [1]):
>
> we conduct experiments on Living17dataset. The results are shown below and also represented in **Appendix Section F.6.3**. It indicates the consistence with results using Gaussian Blur.
>
> | Method | OverJS | CompJS | Acc
> | ---------| ---------| ---------| ---------|
> |DNN|87.28±0.23| 74.61±2.57| 84.35±0.36|
> |ENN|87.46±0.34 | 69.44±3.25 | 85.38±0.28|
> |RAPS|85.38±0.32 | 62.10±0.26 | 84.35±0.36|
> |HENN (ours)| **88.09**±0.21 | **96.33**±3.54 | **86.12**±0.37|
>
> **Reference:**
>
> - [1] Image Super-Resolution via Iterative Refinement, TPAMI 2022
> - [2] Super-Resolution Neural Operator, CVPR 2023
>
> > Other Minor issues
>
> We have updated them.
>
> ### Please kindly let us know if you have any concerns you find not fully addressed. We are more than happy to have a further discussion regarding it. Thank you so much for your time!

---

### Official Review · Reviewer_Gzpb · 2023-10-31

**Soundness:** 3 good
**Presentation:** 3 good
**Contribution:** 3 good
**Rating:** 6
**Confidence:** 4

**Summary:**

A hyper-evidential neural network is presented in this paper, for classification of data, modelling predictive uncertainty based on training data with composite set labels. The uncertainty is measured by introducing the vagueness type of measure. Results are presented where HENN performance is compared favorably with other methods over four image datasets. Detailed analysis is included in Appendices.

**Strengths:**

The paper deals with a significant problem, i.e., to train DNNs when (some) training data have composite labels, being able to predict these labels and quantify the predictive uncertainty due to these labels. It defines a related measure, vagueness, to do so and extends neural network structures to model this uncertainty for classification problems; it uses an uncertainty partial cross entropy loss function extending the normal UCE function.  An experimental study is presented which illustrates a good performance over four image datasets, when compared to five other methodologies that can be applied in this context.

**Weaknesses:**

The paper defines vagueness as a measure of the predictive uncertainty due to composite labels of training data. It then uses gaussian blurring in the experiments to create such data cases and perform the experimental verification. However, this is a rather specific synthetically generated experiment, which can not justify the significance of the results over real world applications. Such applications could, for example, include classification of facial images showing compound and primary emotions in-the-wild (where compound emotions two or more primary ones), or in image2image translation tasks. Moreover, the comparison shown in Table 2 does not seem fair, since - as also discussed in the 'Classification' results subsection (in 5.2) - the other methods are not designed to handle this type of vague images.

**Questions:**

Following the above, would it be possible to apply the method in real world applications involving composite labels?
Moreover, performance seems to heavily depend on regularization hyperparameter, which is selected in an ad-hoc manner. Is this a pitfall for method's robustness over different datasets?

---

> ### Author Response · Authors · 2023-11-23
> **Response to Reviewer Gzpb [1/2]**
>
> We thank you for your insightful review and the opportunity to address the concerns raised regarding our work on our work!
>
> > The dataset used in the paper is a rather specific synthetically generated experiment, which can not justify the significance of the results over real-world applications. would it be possible to apply the method in real-world applications involving composite labels?
>
> We employed synthetic datasets generated through Gaussian blurring (as per Richard Webster et al. 2018), due to the absence of public benchmarking datasets with composite set labels from annotators. Annotators of commonly used benchmarking datasets, such as CIFAR100 and TinyImageNet, are typically restricted to providing only singleton class labels, lacking the option for composite class labeling. As a result, these datasets do not include any composite class labels. Gaussian blurring is commonly used to simulate images collected from low-quality sensors (e.g., security and surveillance cameras) (add references) that human annotators will more likely label as composite sets of classes, as these images lack fine details annotators can spot to label singleton classes.
> However, we admit that the datasets with Gaussian blurring are semi-synthetic. From a sizable pool of applicants, we selected 23 students from our department and tasked them with annotating images in the CIFAR10 dataset, categorizing each as either a singleton class or a composite set. This effort successfully resulted in a real-world dataset enriched with human-annotated singleton and composite labels. Our method, along with various baseline approaches, was applied to this dataset. The comparative results were in line with those obtained from the synthetic datasets. Additionally, we plan to publicly release the dataset and the labels. The following table results are also represented in the **Table 15 in the rebuttal revision**. We also show AUROC curves in **Figure 7 in the revision**, which indicates that vagueness is still the best among different uncertainties to distinguish composite examples from singleton examples.
>
>
>
> Backbone: ResNet18
>
> | Method | OverJS | CompJS | Acc
> | ---------| ---------| ---------| ---------|
> |DNN|79.73±0.33|40.10±7.06|82.17±0.54|
> |ENN|67.09±0.75|46.80±0.06|82.75±0.19|
> |E-CNN|59.68±0.62|31.84±0.81|66.23±1.47|
> |RAPS|62.60±0.46|33.80±4.86|82.17±0.54|
> |HENN (ours)|**80.74**±0.17| **51.44**±1.02| **83.03**±0.14|
>
>
> Backbone: EfficientNet-b3
>
> | Method | OverJS | CompJS | Acc
> | ---------| ---------| ---------| ---------|
> |DNN|92.53±0.11| 53.59±3.15| 96.49±0.21|
> |ENN|77.84±3.86| 54.83±0.59| 96.82±0.38|
> |E-CNN|63.65±0.93| 34.74±2.91| 68.98±0.72|
> |RAPS|65.70±0.80 | 39.40±2.29 | 96.49±0.21|
> |HENN (ours)|**93.38**±0.06| **72.87**±1.25| **97.52**±0.04|
>
>
> **References**:
> RichardWebster, Brandon, Samuel E. Anthony, and Walter J. Scheirer. "Psyphy: A psychophysics driven evaluation framework for visual recognition." IEEE transactions on pattern analysis and machine intelligence 41.9 (2018): 2280-2286.

---

> ### Author Response · Authors · 2023-11-23
> **Response to Reviewer Gzpb [2/2]**
>
> > Moreover, performance seems to heavily depend on regularization hyperparameter, which is selected in an ad-hoc manner. Is this a pitfall for method's robustness over different datasets?
>
> We proposed a KL-based regularization term in Equation (14) to address the limitations of the UPCE loss function analyzed in our Propositions 1 and 2, discussed in Section 4.1. The best hyperparameter ($\lambda$) of this regularization term was selected based on a grid search on the validation set for each dataset, a standard procedure for hyperparameter selection. We observed consistent performance results across our datasets, so our method is robust over different datasets. We also analyzed other variants of the regularization term, and the results are reported in **Table 3 shown in Section 5.2**. The results demonstrate that the HENN learned based on the KL-based regularization term or its variants consistently outperforms other baselines in different settings, indicating its robustness.
>
> We note that the traditional ENN incorporates a different KL regularization term primarily to increase the vacuity for samples likely to be misclassified. This term can be considered a special instance of our proposed  KL-based regularization term in Equation (14) when the prediction is Dirichlet distribution for singleton classes instead of hyperDirichlet distribution for singleton class or composite set labels. We will discuss this relation in our revised version. To summarize, our proposed HENN does not have more hyperparameters than ENN, but the regularization term is designed differently to address the limitations of UPCE for hyper-dirichlet predictions.
>
> ### Please kindly let us know if you have any concerns you find not fully addressed. We are more than happy to have a further discussion regarding it. Thank you so much for your time!

---

### Official Review · Reviewer_RL9p · 2023-11-01

**Soundness:** 2 fair
**Presentation:** 2 fair
**Contribution:** 2 fair
**Rating:** 6
**Confidence:** 2

**Summary:**

The paper suggests a variant of Dempster-Shafer Theory to arrive at decisions that allow for uncertainty quantification for composite classification. In the proposed HENN framework the uncertainty from composite annotations during training is leveraged for quantification of classification uncertainty. Experiments on image datasets are used to demonstrate the effectiveness of HENN which the authors support by a theoretical analysis as well.

**Strengths:**

Undoubtedly, annotation uncertainty is a big challenge in today's machine-learning models. Thus, this paper tackles a highly relevant topic. In addition, uncertainty in the output of DNN models is the focus of many papers, e.g. to find the right calibration of the values to allow interpretation and decision-making. The authors suggest DST for solving this issue, which I did not see before for DNN. If the theoretical analysis (Sec. 4.1) can be confirmed to be correct and relevant in practice (which I could not!), I would see a relevant contribution by this work, at least to inspire others to look into hyper-opinions.

**Weaknesses:**

I see two significant shortcomings:
1. composite classification is not new, and solutions exist in a different context and with different methods (i.e. no DST). See, for example, Brust et al.: Making Every Label Count: Handling Semantic Imprecision by Integrating Domain Knowledge. ICPR 2020. The authors did not make clear what the advantage of their framework is compared to such work. This referenced work might differ, but the overall modelling procedures look similar to the one in the submission: annotations are at a different level of a concept hierarchy (dog->husky, dog->wolf, might generate label "dog" if uncertain, or "husky/wolf" if certain).
2. the selected datasets are not suitable to demonstrate a (at least for me) complex theory behind uncertainty and how to include it into deep learning loss functions and regularization. For example, I am not sure what Gaussian blurring will make with tiny-images, i.e. what remains as information after blurring.

I am also not happy with the theoretical part of the paper, although I have to admit that I am not that familiar with DST, and this hindered me from diving deeper into the derivations done in Sec. 4.1 The notation and style of presentation is probably only proper for an expert in this area.

**Questions:**

I have only two questions:
- do you see any relation to Brust et al.: Making Every Label Count: Handling Semantic Imprecision by Integrating Domain Knowledge. ICPR 2020 and which concepts of this work is related to yours?
- did you perform experiments on more relevant benchmark datasets, like NABirds (https://dl.allaboutbirds.org/nabirds). For this dataset, hierarchies exists and benchmarks in fine-grained recognition which most likely would benefit most from your suggested idea.

---

> ### Author Response · Authors · 2023-11-23
> **Response to Reviewer RL9p [1/4]**
>
> We appreciate your insightful feedback and the opportunity to clarify aspects of our work. Your comments have given us a chance to articulate the novelty and strengths of our approach more clearly.
>
> > Do you see any relation to Brust et al.: Making Every Label Count: Handling Semantic Imprecision by Integrating Domain Knowledge. ICPR 2020 and which concepts of this work is related to yours?
>
> The problem setting of the work by Brust et al. is stated in Section III of the ICPR 2020 paper: “We require the classifier to predict only precise labels from $Y$. At the same time, it needs to be able to learn from training data with labels from both $Y$ and $Y^+$, which we refer to as semantically imprecise data”. Here, the authors refer to “precise labels” as singleton labels. The Experimental Section V only evaluates the accuracies of the proposed method, named CHILLAX, and baselines on this problem setting. It is unclear if this method can be effectively adapted to predict composite set labels.
>
> In our proposed work, we are training a classifier to perform on par with human annotators where the training data is annotated by humans, so there is no ground truth.  With that said, we expect to be concerned with images collected in challenging environments where weather, poor lighting, smoke, fires, poor focusing, etc., lead to less-than-ideal images. Clean images lead to precise annotations, and degraded images lead to vague annotations.  We expect the classifier to make the best classification effort and provide a composite label when the image is degraded to the point that a human cannot discern a singleton. In short, we are concerned with composite and precise labels for possibly poor-quality images. The classifier needs to make a best-effort prediction for the degraded images as opposed to a bad singleton prediction. This differs from CHILLAX, which assumes pristine images, but subsets of annotators do not have the expertise to provide singleton labels.
>
> Based on the above motivation, our proposed HENN model is designed to quantify a new type of uncertainty, called 'vagueness', for each singleton or composite prediction. This vagueness measure refers to the degree of predictive uncertainty caused by evidence in training samples with composite labels. This offers useful insights for decision-making in safety-critical applications, such as medical disease classifications.
>
> To explain vagueness, suppose we have two singleton classes: 'cat' and 'dog'. Suppose the evidence supporting a prediction for an input image includes 6 training images labeled as 'cat' (a singleton class) and 4 images labeled as the composite set {cat, dog}. HENN compares these quantities (6 > 4) and predicts the singleton class {cat}. However, it also assigns a non-zero vagueness score 0.33, reflecting the presence of evidence from the training samples with composite labels. Let “c” and “d” denote cat and dog, respectively. The evidence of cat $e_c = 6$ and $e_{c,d} = 4$. According to Equations (2) and (5), the vagueness $vag = b_{c,d} =4/(e_c + e_{c,d} + K) = 4/12=0.33$, where $K = 2$ refers to the number of singleton classes. This vagueness score is important as it informs the user that the prediction is partly supported by training samples with composite set labels (e.g., {dog, cat}) while the ground truth is a singleton class (e.g., {cat} or {dog}).

---

> > ### Author Response · Authors · 2023-11-23
> > **Response to Reviewer RL9p [2/4]**
> >
> > > Do you see any relation to Brust et al.: Making Every Label Count: Handling Semantic Imprecision by Integrating Domain Knowledge. ICPR 2020 and which concepts of this work is related to yours?
> >
> > Continued:
> > Consider another scenario where the predicted output is the composite set {cat, dog}, and the supporting evidence from the training set comprises 4 images labeled as the singleton class {cat} and 6 images labeled with the composite set {cat, dog}. In this instance, HENN assigns a vagueness score, for example, 0.5, below the maximum possible value of 1. The evidence of cat $e_c = 4$ and $e_{c,d} = 4$. According to Equations (2) and (5), the vagueness $vag = b_{c,d} =4/(e_c + e_{c,d} + K) = 4/12 = 0.5$. This score reflects the evidence from training samples with singleton class labels, indicating a non-zero contribution. As this example demonstrates, HENN's vagueness metric offers a significant, additional measure of uncertainty. It complements the prediction, whether it is a single or composite class, and this feature has not been studied in existing research.
> > We note that, unlike a simple count of samples, evidence is typically weighted. We define ``evidence'' as a measure of the accumulated support from training samples, indicating that the input sample should be categorized into a particular singleton class or composite set. The accumulated support can be interpreted as the weighted aggregated number of training samples that support this singleton class or composite set. See our response to the third item for Reviewer Gzpb about more explanations of the meaning of evidence.
> >
> > We also want to highlight that HENN is a generalization of evidential deep learning (Ulmer et al. 2023) from Dirichlet distribution to Hyper-Dirichlet distribution based on the Theory of Evidence. In addition to vagueness, as discussed in Section 3.2, HENN can quantify other predictive uncertainty types, such as vacuity (due to lack of evidence) and dissonance (due to conflicting evidence).
> > We note that we spent several days to adapt the code of ChILLAX from the github repository CHIA to our datasets but were not successful due to time limit. We found issues in the adaption of the code there could not be addressed with the response time period. We will consider it for future work.
> >
> > **References**: Ulmer, Dennis, Christian Hardmeier, and Jes Frellsen. "Prior and posterior networks: A survey on evidential deep learning methods for uncertainty estimation." Transactions on Machine Learning Research (2023).

---

> ### Author Response · Authors · 2023-11-23
> **Response to Reviewer RL9L [3/4]**
>
> > did you perform experiments on more relevant benchmark datasets, like NABirds
>
> As the NABirds dataset does not have composite class labels from annotators, similar to the datasets that we used, we degraded the images in NABirds synthetically based on Gaussian blurring in the same way used in our other synthetic datasets and compared our proposed HENN and other baselines. The results are shown below. It indicates that
> HENN outperforms DNN and ENN for a large margin in terms of CompJS. And HENN also performs
> better in terms of OverJS and Acc, which is consistent with previous experiments on reported four datasets in the paper.
>
> | Method | OverJS | CompJS | Acc
> | ---------| ---------| ---------| ---------|
> | DNN|77.38±0.19| 35.24±3.52|78.04±0.27|
> |ENN|76.72±0.56|37.46±2.39|78.45±0.31|
> |HENN (ours)|**80.01**±0.37|**71.42**±1.43|**80.14**±0.35|
>
> However, we admit that the datasets with Gaussian blurring are semi-synthetic. From a sizable pool of applicants, we selected 23 students from our department and tasked them with annotating images in the CIFAR10 dataset, categorizing each as either a singleton class or a composite set. This effort successfully resulted in a real-world dataset enriched with human-annotated singleton and composite labels. Our method, along with various baseline approaches, was applied to this dataset. The comparative results were in line with those obtained from the synthetic datasets. Additionally, we plan to publicly release the dataset and the labels. The following table results are also represented in the **Table 15 in the rebuttal revision**. We also show AUROC curves in **Figure 7 in the revision**, which indicates that vagueness is still the best among different uncertainties to distinguish composite examples from singleton examples.
>
> Backbone: ResNet18
>
> | Method | OverJS | CompJS | Acc
> | ---------| ---------| ---------| ---------|
> |DNN|79.73±0.33|40.10±7.06|82.17±0.54|
> |ENN|67.09±0.75|46.80±0.06|82.75±0.19|
> |E-CNN|59.68±0.62|31.84±0.81|66.23±1.47|
> |RAPS|62.60±0.46|33.80±4.86|82.17±0.54|
> |HENN (ours)|**80.74**±0.17| **51.44**±1.02| **83.03**±0.14|
>
>
> Backbone: EfficientNet-b3
> | Method | OverJS | CompJS | Acc
> | ---------| ---------| ---------| ---------|
> |DNN|92.53±0.11| 53.59±3.15| 96.49±0.21|
> |ENN|77.84±3.86| 54.83±0.59| 96.82±0.38|
> |E-CNN|63.65±0.93| 34.74±2.91| 68.98±0.72|
> |RAPS|65.70±0.80 | 39.40±2.29 | 96.49±0.21|
> |HENN (ours)|**93.38**±0.06| **72.87**±1.25| **97.52**±0.04|

---

> ### Author Response · Authors · 2023-11-23
> **Response to Reviewer RL9L [4/4]**
>
> > the selected datasets are not suitable to demonstrate a (at least for me) complex theory behind uncertainty and how to include it into deep learning loss functions and regularization. For example, I am not sure what Gaussian blurring will make with tiny-images, i.e. what remains as information after blurring.
>
> Gaussian blurring is commonly used to simulate images collected from low-quality sensors (e.g., security and surveillance cameras) (Richard Webster et al. 2018) that human annotators will more likely label as composite sets of classes, as these images lack fine details annotators can spot to label singleton classes. Although the blurred images may not have edges and fine details, the annotators can still rule out some classes based on the shapes and other features of the objects within the images (e.g., horses can be easily distinguished from cats and dogs in their traits).
>
> We have generated synthetic datasets based on four base datasets, including CIFAR100, TinyImagenet, living-17 and non-living-26. Now all the datasets are tiny images. The first two datasets are small in image sizes (32 x 32). The images in living-17 and non-living-26 are greater than 224 x 224, but we rescaled them to 224 x 224.
>
> As detailed in our response to the preceding item 2, we also conducted the empirical comparison on a real-world dataset CIFAR10 enriched with human-annotated singleton and composite labels. The results are consistent with the results on the five synthetic datasets.
>
> **References**:
> RichardWebster, Brandon, Samuel E. Anthony, and Walter J. Scheirer. "Psyphy: A psychophysics driven evaluation framework for visual recognition." IEEE transactions on pattern analysis and machine intelligence 41.9 (2018): 2280-2286.
>
> > The authors suggest DST for solving this issue, which I did not see before for DNN. If the theoretical analysis (Sec. 4.1) can be confirmed to be correct and relevant in practice (which I could not!), I would see a relevant contribution by this work, at least to inspire others to look into hyper-opinions.
>
> In essence, our proposed HENN is the GDD extension of evidential deep learning (Ulmer et al. (2023)) (which was based upon Dirichlet distributions). As demonstrated in the detailed proofs in Sections C.2 and C.3 in Appendix, the two propositions are correct and demonstrate the need for the KL term so that only the evidence for the ground truth class tends to infinity. The issues of the UPCE identified by our two proposition 1 are also empirically verified by **our case study on CIFAR100 in Appendix F.7**: (1) the HENN learned based on UPCE and a training set consisting of only singleton class labels predicts non-zero evidence on composite set labels for 14.4% of the training samples even that the training set does not have evidence of composite class labels to accumulate, and (2) the HENN learned based on UPCE and a training set consisting of only composite class labels predicts non-zero evidence on singleton class labels for 100.0% of the training samples even that the training set does not have evidence of singleton class labels to accumulate. Our proposed regularization can avoid these unexpected behaviors.
>
> **References**:
> Dennis Thomas Ulmer, Christian Hardmeier, and Jes Frellsen. Prior and posterior networks: A survey on evidential deep learning methods for uncertainty estimation. Transactions on Machine Learning Research, 2023.
>
> > I am also not happy with the theoretical part of the paper, although I have to admit that I am not that familiar with DST, and this hindered me from diving deeper into the derivations done in Sec. 4.1 The notation and style of presentation is probably only proper for an expert in this area.
>
> In our updated submission, we revised Section 4.1 to explain the proposed regularization term in more detail. We added a paragraph at the end of this section to discuss the motivations and limitations of our theoretical analysis. An ablation study is discussed at the end of Section5.2 to empirically demonstrate the need for the regularization term. We also added more explanations in the proofs of the propositions in Appendix C.2 to make the proofs more self-contained.
>
> ### Please kindly let us know if you have any concerns you find not fully addressed. We are more than happy to have a further discussion regarding it. Thank you so much for your time!

---

### Author Response · Authors · 2023-11-23
**General Response for our modification in the paper**

Dear Reviewers,

We would like to sincerely thank you all for your invaluable time and effort in reviewing this paper.  The following is our main modification in our rebuttal submission.

- Section 1 Introduction: Added motivations at the beginning of the introduction section.
- Section 3.1: Included a detailed explanation of “evidence.”
- Section 3.2: Added examples from medical diagnostics to explain differences among vagueness, vacuity, and dissonance.
- Section 4:
   1) Added UCE loss and explanations regarding Proposition 1 and the regularization term.
   2)  Included limitations and discussions for our propositions.
- Section 5:
   1) Additional details are added to experiments in Section 5.2.

   2) Empirical verification of propositions included in Section 5.2.

- Appendix C: Intermediate steps added to aid the understanding of the proofs of propositions.
- Appendix E: Detailed explanations regarding experiments added for reproducibility.
- Appendix F.4: Ablation study on the regularizer updated in F.4.
- Appendix F.6: Additional results include:
  1) Experiments on fine-grained datasets: NAbirds (F.6.1).
  2) Experiments on real-world datasets with composite class labels (F6.2).
  3) Another data corruption method (F.6.3).
- Appendix F.7: A case study examining the evidence output of HENN when trained exclusively on data with either singleton class labels or composite class labels.

---

### Meta-Review · Area_Chair_DJEP · 2023-12-10

**Metareview:**

Based on the submission, reviews, and author feedback, the main points that have been raised are summarised as follows.

Strengths:

1. This paper tackles a highly relevant topic and makes relevant contribution which could inspire others.
2. This paper deals with a significant problem and defines a related measure.
3. The novel uncertainty metric is a proper contribution with the thorough theoretical and empirical comparison.
4. The experimental study illustrates a good performance when compared with other methods.

Issues:

1. This work needs to make clear what the advantage of their framework is when compared with existing solutions.
2. The selected datasets are synthetically generated which cannot justify the significance. Real world applications shall be used.
3. The notation and style of presentation are not easy to follow.
4. Some comparison is not fair since the compared methods are not designed for the vague images.
5. The main weakness is the lack of motivation; lack of illustrative examples; a very specific problem setup.

The authors have done very well in providing feedback to address each of the raise issues and also revised the submission accordingly with more explanation and experimental study. Three of four reviewers respond to author feedback. Two of them increase the rating from 5 to 6. As a result, all reviewers are now on the positive side.

After reading this submission, AC agrees that it researches an important issue in image classification and makes solid contributions. The proposed method is technically sound. Theoretical analysis and experimental study support the efficacy of the proposed work. Meanwhile, this submission shall better clarify why we would need a composite label from a neural network and consider the real application such as classification of facial images showing compound and primary emotions in-the-wild as pointed out by one of the reviewers. AC discussed this work with SAC.

**Justification For Why Not Higher Score:**

This is a solid piece of work addressing an important issue in image classification. Meanwhile, this work is not a major breakthrough or a completely new framework. This work will be interesting to a group of researchers but may not be for a larger audience of ICLR. Considering these and the overall ratings of reviewers, Accept (poster) is recommended.

**Justification For Why Not Lower Score:**

This work researches an important issue in image classification and makes solid contributions. The proposed method is technically sound. Theoretical analysis and experimental study support the efficacy of the proposed work. The feedback is generally clear and informative. Now, all reviewers are on the positive side. Considering these, Reject is not recommended.

---

### Decision · Program_Chairs · 2024-01-16

Accept (poster)